# Dissolved Nitric Oxide in the Lower Elbe Estuary and the Hamburg Port Area

Riel Carlo O. Ingeniero[1], Gesa Schulz[2,3], Hermann W. Bange[1]

[1]Marine Biogeochemistry Research Division, GEOMAR Helmholtz Centre for Ocean Research Kiel, Wischhofstr. 1-3, Kiel, 24148 Germany

[2] Institute of Geology, Center for Earth System Research and Sustainability (CEN), Universität Hamburg, Hamburg, 20146, Germany

[3]Institute of Carbon Cycles, Helmholtz-Zentrum Hereon, Geesthacht, 21502, Germany

*Correspondence to*: Riel Carlo O. Ingeniero (ringeniero@geomar.de)

**Abstract.** Nitric oxide (NO) is an intermediate of various microbial nitrogen cycle processes and the open ocean and coastal areas are generally a source of NO in the atmosphere. However, our knowledge about its distribution and the main production processes in coastal areas and estuaries is rudimentary at best. To this end, dissolved NO concentrations were measured for the first time in surface waters along the lower Elbe Estuary and Hamburg Port area in July 2021. The discrete surface water samples were analyzed using a chemiluminescence NO analyzer connected to a stripping unit. The NO concentrations ranged from below the limit of detection (9.1 pM) to 17.7 pM, averaging at 12.5 pM, and were supersaturated in the surface layer of both the lower Elbe Estuary and the Hamburg Port area, indicating that the study site was a source of NO to the atmosphere during the study period. On the basis of a comprehensive comparison of NO concentrations with parallel nutrient, oxygen, and nitrous oxide concentration measurements, we conclude that the observed distribution of dissolved NO was most likely resulting from nitrification. In the Hamburg Port area, however, nitrifier denitrification and/or denitrification might affect the NO distribution as well.

## 1 Introduction

Nitric oxide (NO) is an atmospheric trace gas that is rapidly oxidized to atmospheric nitrogen dioxide ($NO_2$). $NO_x$ (= NO + $NO_2$) is a significant contributor to photochemical smog (Haagen-Smit and Fox, 1954), a cause of acid rain (Likens et al., 1979; Fanning, 1989), and affects tropospheric ozone ($O_3$) (Haagen-Smit and Fox, 1954). Atmospheric $NO_x$ has an atmospheric lifetime ranging from hours to days (IPCC, 2021). Because its atmospheric reactions yield $O_3$, methane, and nitrate aerosols, it is an indirect greenhouse gas with an overall negative efficient radiative forcing (IPCC, 2021).

The major sources of atmospheric $NO_x$ are emissions from fossil fuel combustion and soils (Jaeglé et al., 2005). Until now, little is known about the distribution as well as the production and consumption processes of NO in the marine environment. Two known primary sources of NO in the ocean are NO photolysis from nitrite and NO production from phytoplankton, macroalgae, and the microbial nitrogen cycle. Bange et al. (2024) noted that the consumption mechanisms of NO in the marine environment are still unresolved.

Zafiriou et al. (1980) measured the dissolved NO concentration in seawater for the first time in the central equatorial Pacific Ocean. They noted that the ocean could be a source of NO to the atmosphere due to its photochemical production from dissolved nitrite ($NO_2^-$). NO is also an important intermediate of microbial nitrogen cycle processes such as denitrification, nitrification, and anammox (Schreiber et al., 2012; Kuypers et al., 2018). Moreover, NO was identified as a signal molecule on a cellular level in many marine organisms and between bacteria and algae (see Abada et al., 2023).

The determination of dissolved NO concentration is challenging because of its reactivity, which results in a very short lifetime in (sea)water (Lancaster, 1997), ranging from 3 to 100 s (Zafiriou and McFarland, 1981; Olasehinde et al., 2010). Nevertheless, measurements of dissolved NO in aquatic environments such as open and coastal oceans and rivers have received increasing attention during the last decade. Examples of recent NO measurement campaigns include those in the Kurose River in Japan (Anifowose et al., 2015), the Seto Inland Sea in Japan (Olasehinde et al., 2010), the tropical Northwestern Pacific Ocean (Tian et al., 2019), the oxygen minimum zone off the coast of Peru (Lutterbeck and Bange, 2015; Lutterbeck et al., 2018), and the coastal seas off Qingdao (Tian et al., 2021).

These studies performed at different periods have indicated that both open and coastal seas are a source of atmospheric NO with fluxes ranging from 0.70 (Anifowose and Sakugawa, 2017) to as high as $45.00 \times 10^{-17}$ mol cm$^{-2}$ s$^{-1}$ (Gong et al., 2023). Global estimates for oceanic NO emissions are still lacking, and studies on the temporal (i.e., diurnal, seasonal, interannual) and spatial variability of NO emissions are not available. To address these gaps, expanded measurements of NO distribution in the open ocean and coastal waters are essential to enhance our understanding and provide a more accurate assessment of sea-to-air flux densities.

A recent paper by Gong et al. (2023) argued that DIN plays an important role in NO distribution– a high level of dissolved inorganic nitrogen (DIN) establishes the necessary conditions for NO production. Other studies (e.g., Olasehinde et al., 2010; Anifowose et al., 2015; Anifowose and Sakugawa, 2017; Ayeni et al., 2021) also observed a positive correlation between NO concentrations or photoproduction rates with dissolved $NO_2^-$ concentrations. To our understanding, dissolved NO measurements and the magnitude of flux density in estuaries, which have relatively high DIN concentrations (Howarth et al., 2011), have not yet been reported.

This paper presents the first measurement of dissolved NO concentrations in the lower Elbe Estuary and Hamburg Port basins during a ship campaign in July 2021. The overarching objectives of our study were (i) to determine the distribution of dissolved NO along the salinity gradient, (ii) to estimate the flux density of NO across the water/atmosphere interface, and (iii) to identify the potential production pathways and controlling factors on NO distribution in the lower Elbe Estuary and Hamburg Port area.

## 2 Methods

### 2.1 Study site

Originating from the Karkonosze Mountains in the northern region of the Czech Republic, the Elbe River basin is the fourth largest catchment area (148,268 km$^2$) in Central Europe (Amann et al., 2012) with average long-term freshwater runoff of

about 720 m$^3$ s$^{-1}$ (Kerner, 2007). In this study, the given stream distance (i.e., Elbe-km) refers to the distance from the point where the Elbe passes the border from the Czech Republic to Germany.

Its estuary stretches about 140 km from the weir in Geesthacht to the coastal city of Cuxhaven in Lower Saxony, Germany. It is considered the most significant riverine nitrogen source in the German Bight of the North Sea (Dähnke et al., 2008). It is a turbid and well-mixed estuarine system with a maximum turbidity zone near Brunsbüttel at Elbe-km 698 (Burchard et al., 2017; Kappenberg and Grabemann, 2001). It has semi-diurnal tidal ranges of 2 to 4 meters, and its wind conditions are dominated by westerly winds (Hein et al., 2021). Generally, the Elbe Estuary is deepened and dredged to maintain a water depth of 15 to 20 m to grant access for large container ships into the Port of Hamburg (Kerner, 2007).

Its water residence time is estimated to range between 3 to 22 days (Geerts et al., 2012), with longer residence times during summer when the river discharge is low (Hein et al., 2014). The dissolved NO concentration in the surface water of the Elbe Estuary was measured at various sampling points from the mouth of the estuary to the Hamburg Port area, as shown in Fig. 1. Sampling was performed upstream against the outgoing tide to prevent tidal effects on our measurements.

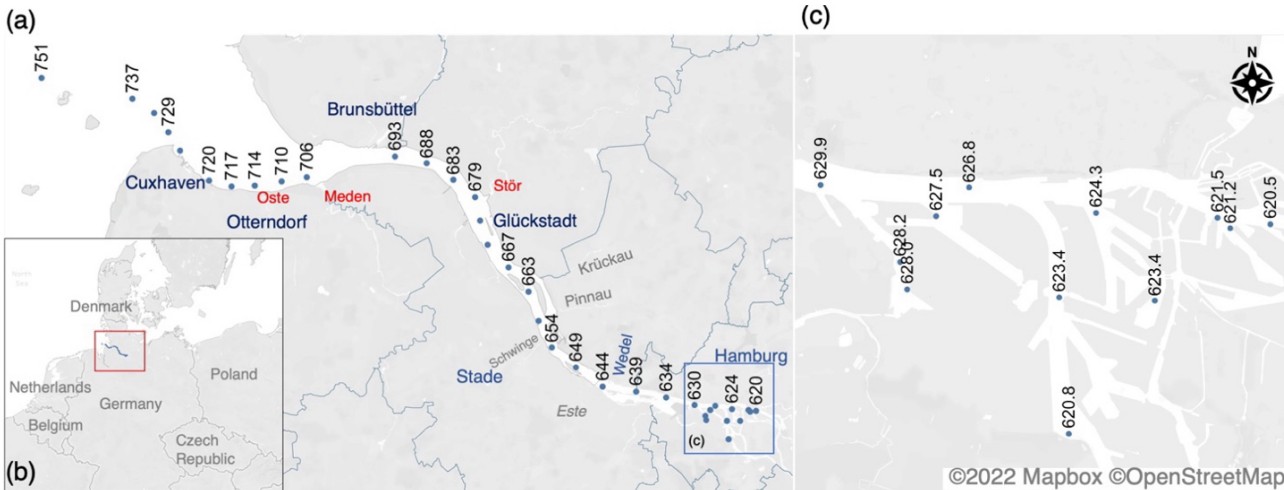

Figure 1: a) Map of the sampling locations with corresponding stream kilometers and b) the relative position of the sampling area in Northern Europe. c) A higher-resolution map of the sampling points in the Hamburg Port area from Elbe-km 620 to 630. Note that in this study, we used the following salinity zoning: the brackish-coastal zone downstream of Elbe-km 690, the limnic zone from Elbe-km 630 to 690, and the Hamburg Port area located from Elbe-km 620 to 630. We indicated the tributaries Oste, Meden, and Stör in the map. Background map: ©OpenStreetMap 2022. Distributed under the Open Data Commons Open Database License (ODbL) v1.0.2.

## 2.2 Sampling

Surface water was sampled on board the RV *Ludwig Prandtl* during a campaign from 27 to 29 July 2021, using a FerryBox flow-through system (Petersen et al., 2014). The system drew water from approximately 1.2 m below the water surface through a membrane pump. The FerryBox continuously measured in situ biogeochemical parameters such as dissolved oxygen (O$_2$), pH, salinity, and water temperature. The sensors in the FerryBox system were routinely calibrated or compared to established standard methods. For instance, the optode measurements were compared with the Winkler titration method, leading to an O$_2$

(in µM) correction of $[1.12 \times O_{2(optode\ measurement)}] + 13.41$ ($R^2 = 0.97$), and the salinity measurements with the Optimare Precision Salinometer (Bremerhaven, Germany).

Discrete water samples were collected for analysis of nutrients, chlorophyll a, and dissolved NO every 20 minutes from the FerryBox system bypass using field sampling collection, preservation, and storage methods described in detail in earlier related publications (Schulz et al., 2022; Norbisrath et al., 2022).

Nitrous oxide ($N_2O$) concentrations were measured continuously from a bypass of the FerryBox system using laser-based off-axis integrated cavity output (OA_ICOS) absorption spectroscopy (Model 914-0022, Los Gatos Res. Inc., San Jose, CA, USA) coupled with a (sea)water/gas equilibrator. The method had a relative average standard error of 1.28%; details are described in Schulz et al. (2023) and Brase et al. (2017). Furthermore, wind speeds at 10 m height were measured onboard with a MaxiMet GMX600 (Gill Instruments, Ltd., Saltash, UK) weather station.

## 2.3 Measurement of dissolved NO

Because of the short lifetime of NO, triplicate NO samples were measured within 20 minutes of sampling following the method described by Lutterbeck and Bange (2015). During this campaign, we used a portable NO calibration source (2BTech Model 714 NO2/NO/O3 Calibration Source™) to calibrate the NO detector (Birks et al., 2020). The resulting gas output from the calibrator covered the detection range of the NO detector from 0 to 1000 ppb NO. A schematic diagram of the minor update to the components in the analytical method described in Lutterbeck and Bange (2015) is shown in Fig. S1.

NO signal outputs by the NO detector were recorded using PuTTY 0.78, a free and open-source client application for Windows. To determine NO mole fractions, the Rieman integrals of the signal peaks were calculated using the MATLAB (2022b) trapezoidal numerical integration function trapz. After applying a linear calibration curve of aqueous NO standard solutions prepared according to Lutterbeck and Bange (2015), the final concentrations of dissolved NO ($C_w$ in mol $L^{-1}$) were computed with Eq. (1):

$$C_w = x'_{sw} \times P \times K_H \tag{1}$$

where x' is the measured mole fraction of NO from the water sample, P is the atmospheric pressure, and $K_H$ is the Henry's law constant for NO ($1.9 \times 10^{-3}$ mol $L^{-1}$ atm$^{-1}$) (see Zacharia and Deen, 2005).

The instrument limit of detection (LOD) was computed as 3 times the standard deviation ($\sigma$) of the blank or zero calibration point. During this campaign, the instrument limit of detection was 9.1 pM, while the average relative standard error was approximately $\pm\ 26\ \%$. NO concentrations < LOD were omitted in further calculations.

## 2.4 Estimation of NO flux density and saturation ratios.

The flux of NO at the water–air boundary ($F_{NO}$, in mol cm$^{-2}$ s$^{-1}$) was estimated using Eq. (2) from Anifowose and Sakugawa (2017):

$$F_{NO} = k_w \times (C_w - K_H\,p_{NO}) \times 10^{-1} \tag{2}$$

where $k_w$ is the liquid phase transfer velocity (m s$^{-1}$), and $p_{NO}$ is the partial NO pressure (atm) in the overlying atmosphere. The value of $k_w$ expressed in m s$^{-1}$ was determined according to Borges et al. (2004) (see also Brase et al., 2017):

$$k_w = (360000)^{-1} \times (4.045 + 2.58\,U) \times \left(\frac{S_c}{600}\right)^{-0.5} \tag{3}$$

$$S_c = \frac{v_{sw}}{D} \tag{4}$$

where $U$ is the wind speed at 10 m height (m s$^{-1}$), $S_c$ is the Schmidt number, $v_{sw}$ is the kinematic viscosity of surface water, and $D$ is the diffusion coefficient of NO in water. $U$ measured onboard ranged from 1.76 to 8.86 m s$^{-1}$, with a mean ± standard deviation (SD) wind speed at the sampling stations of $5.78 \pm 2.12$ m s$^{-1}$. The kinematic viscosity ($v_{sw}$, in m$^2$ s$^{-1}$) was calculated using the following equation:

$$v_{sw} = \frac{\mu_{sw}}{\rho} \tag{5}$$

where the dynamic viscosity of surface water ($\mu_{sw}$, in kg m$^{-1}$ s$^{-1}$), a function of temperature ($T$) and salinity ($S$), was estimated using Eqs. (6–9) from Sharqawy et al. (2010), while density ($\rho$, in kg m$^{-3}$) was determined using a MATLAB (2022b) function from Ruiz-Martinez (2021) derived from Gill (1983).

$$\mu_{sw} = \mu_w\,(1 + AS + BS^2) \tag{6}$$

$$A = 1.541 + (1.998\,x\,10^{-2}\,T) - (9.52\,x\,10^{-5}\,T^2) \tag{7}$$

$$B = 7.974 - (7.561\,x\,10^{-2}\,T) + (4.724\,x\,10^{-4}\,T^2) \tag{8}$$

$$\mu_w = 4.2844 \times 10^{-5} + (0.157\,(T + 64.993)^2 - 91.296)^{-1} \tag{9}$$

The NO diffusion coefficient ($D$, $\times\,10^{-9}$ m$^2$ s$^{-1}$) was calculated according to Wise and Houghton (1968):

$$D = 0.9419\,e^{0.0447T} \tag{10}$$

Furthermore, the NO saturation ratio ($NO_{sat}$) was calculated based on the measured NO concentration in surface water ($NO_{sw}$) and the NO concentration ($NO_{eq}$) in equilibrium with the mole fraction of NO ($x'_{NO}$) in the overlying atmosphere:

$$NO_{sat}\,\% = 100 \times \frac{C_w}{C_{eq}} \tag{11}$$

$$C_{eq} = p_{NO} \times K_H = x'_{NO} \times P \times K_H \tag{12}$$

where $P$ is the total ambient pressure set to 1 atm. In this study, it is important to note that the flux density is a rough approximation since the atmospheric NO mole fraction ($x'_{NO}$) was not measured on board but estimated from the air monitoring data available from https://luft.hamburg.de/ (last accessed on 2 May 2023). The mean hourly atmospheric NO concentrations (mole fractions) measured at seven air monitoring stations in the Hamburg Port area during the study period (see Fig. S2) ranged from 2.00 µg m$^{-3}$ (1.60 ppb) to 8.25 µg m$^{-3}$ (6.60 ppb), with a mean ± SD concentration of $4.30 \pm 1.76$ µg m$^{-3}$ (3.40 ±

1.41 ppb). The mean atmospheric NO mole fraction of 3.40 ± 1.41 ppb was used to estimate $C_{eq}$ with Henry's law constant (Eq. 12).

## 2.5 Measurement of ancillary biogeochemical parameters

Dissolved inorganic nitrogen concentrations (nitrate ($NO_3^-$), nitrite ($NO_2^-$), and ammonium ($NH_4^+$)) were measured spectrophotometrically (Dafner, 2015) using air-segmented flow analysis techniques (SEAL AutoAnalyzer 3, SEAL Analytical, Germany). The total dissolved inorganic nitrogen (DIN) was calculated from the sum of $NO_3^-$, $NO_2^-$, and $NH_4^+$ concentrations. The limits of detection were as follows: 0.05 µmol $L^{-1}$ for $NO_3^-$, 0.05 µmol $L^{-1}$ for $NO_2^-$, and 0.07 µmol $L^{-1}$ for $NH_4^+$.

Chlorophyll a was extracted in 90% acetone overnight, measured photometrically (UV-Vis Spectrophotometer DR-6000, Hach Lange GmbH, Germany), and calculated using the parametrization by Jeffrey and Humphrey (1975).

## 2.6 Data analysis

Calculations for apparent oxygen utilization (AOU), $N_2O$ saturation, $N_2O$ sea(water)-to-air flux density, and excess $N_2O$ ($\Delta N_2O$) are discussed in Schulz et al. (2023). The data in this study were visualized using Origin 10.5.117 and MATLAB

(2022b). Statistical analyses, including mean, median, standard deviation (SD), and Pearson's correlation (R), were performed with MATLAB (2022b). Results with p-values < 0.05 were considered statistically significant at a 95 % confidence level.

## 3. Results

### 3.1 Biogeochemical setting along the estuary

Figure 2 presents a scatter plot of various near-surface biogeochemical parameters measured from the North Sea to the Hamburg Port area during the campaign (see also Figure S3). As seen in Fig. 2, we can generally observe the mixing of warmer, less oxygenated, more acidic, and nutrient-rich waters of the Elbe Estuary with North Sea waters.

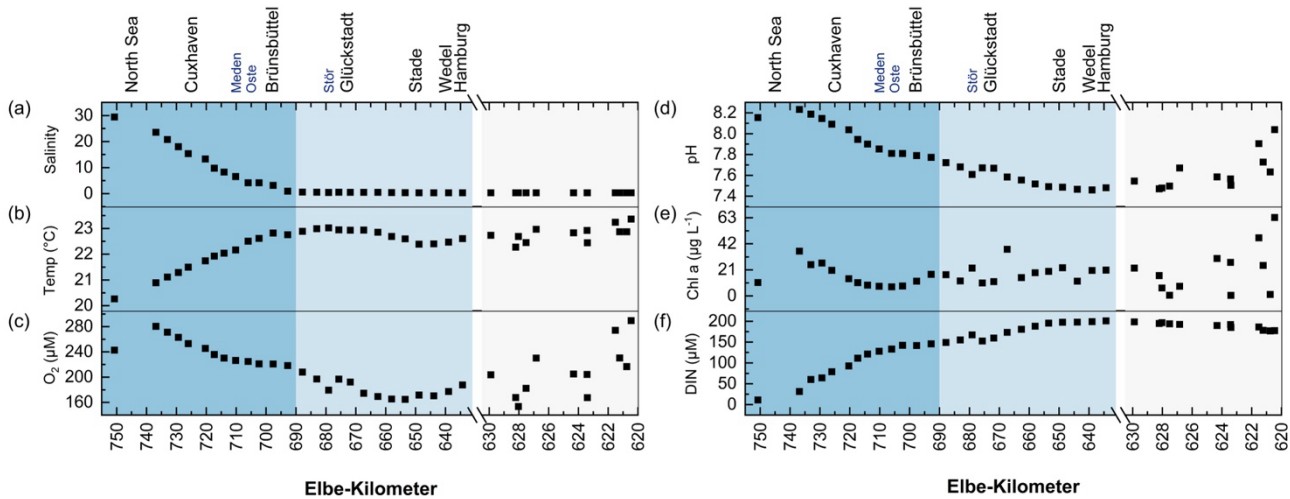

**Figure 2: Distribution of various surface biogeochemical parameters along the Elbe Estuary from the North Sea to Hamburg Port area– (a) salinity, (b) temperature (°C), (c) $O_2$ concentration (µM), (d) pH, (e) chlorophyll a concentration (µg $L^{-1}$), and (f) dissolved inorganic nitrogen (µM). Note that we divided the graph into three distinct salinity zones: the brackish-coastal zone downstream of Elbe-km 690, the limnic zone from Elbe-km 630 to 690, and the Hamburg Port area located from Elbe-km 620 to 630. The zones can be distinguished through a contrast of blue, indicating a decrease in salinity. We included the relative position of selected localities and tributaries along the Elbe Estuary, Meden, Oste, and Stör.**

Surface salinity (Fig. 2a) ranged from 0.31 (Hamburg Port area) to 29.43 (North Sea, Elbe-km 751). A pronounced salinity difference of 6.0 was observed between the North Sea's surface waters (Elbe-km 751) and the estuary's mouth near Cuxhaven (Elbe-km 737). From Elbe-km 737, salinity decreased linearly (0.54 $km^{-1}$, p=2.97 × $10^{-6}$) until Elbe-km 693 (near Brunsbüttel). From Elbe-km 693, salinity gradually declined to about 0.31, approaching the Hamburg Port area. Based on these salinity values, we divided the study site into three distinct zones: the brackish-coastal zone downstream of Elbe-km 690, the limnic zone from Elbe-km 630 to Elbe-km 690, and the Hamburg Port area from Elbe-km 620 to 630.

The surface temperature (Fig. 2b) steadily increased upstream due to the warmer outflow water from the Elbe River in summer. From 20.26 °C in the North Sea (Elbe-km 751), the temperature increased to 23.03 °C near Glückstadt (Elbe-km 679). Upstream of Elbe-km 679, the water temperature ranged from 22.27 °C to 23.37 °C, with the highest surface water temperature of 23.37 °C recorded in the Hamburg Port area at Elbe-km-620.46. This site also had the highest chlorophyll a and $O_2$ concentrations of 63.3 µg $L^{-1}$ and 289.6 µM (saturation: 109.2 %), respectively.

The $O_2$ concentration (Fig. 2c) in the North Sea was 242.8 µM (102.3 % saturation) and thus slightly lower than the $O_2$ concentration of 280.6 µM (115.6 % saturation) at Elbe-km 737. Upstream of Elbe-km 737, the $O_2$ concentration decreased to 179.5 µM (67.3 % saturation) near Glückstadt. From Glückstadt to near Wedel, the $O_2$ concentration generally declined, ranging from 170.5 µM (63.1 % saturation) to 179.5 µM (67.3 % saturation). In the Hamburg Port area, the $O_2$ concentration is highly dynamic, ranging from 153.5 µM (57.1 % saturation) to 289.6 µM (109.2 % saturation), following the pH and

chlorophyll a concentration trend. The minimum $O_2$ concentrations in the campaign were also measured in this location, near Elbe-km 628 and Elbe-km 623.

The trend in pH (Fig. 2d) was analogous to the trend in $O_2$ concentrations (Fig. 2c), likely due to the influence of primary production. At Elbe-km 751 in the North Sea, the measured pH value was 8.15, which is also slightly lower than that at Elbe-km 737 (pH 8.23). From Elbe-km 737, the pH decreased to the lowest measured pH during the entire field campaign (pH 7.46) near Wedel.

Furthermore, the chlorophyll a concentrations ranged from 0.46 µg $L^{-1}$ to 46.9 µg $L^{-1}$ (Fig. 2e). Notably, the minimum and maximum chlorophyll a concentrations were measured in the Hamburg Port area at Elbe km 623 and 622, respectively. No distinguishable spatial pattern on chlorophyll a concentration was observed, except that a distinct peak in chlorophyll a concentration coincided with the maximum suspended particulate matter (SPM) concentration (not shown) of 412.5 mg $L^{-1}$ at Elbe-km 667.4 near Glückstadt.

Overall, the DIN concentrations (Fig. 2f) increased from the mouth of the estuary upstream, with the highest concentrations (201 µM) recorded just before the Hamburg Port area (see also Fig. S3). Further details on the concentration of the DIN substrates are presented in the next section.

In the supplementary material, we provided a table presenting the summary statistics (Table S1) and box plots (Figure S3) of the biogeochemical parameters measured in this study.

### 3.2 Distribution of N₂O and dissolved inorganic nitrogen (DIN) concentrations

Figure 3 shows the distribution of $N_2O$ and DIN components in the study area, while Fig. 4 presents box plots of the data in each salinity zone. The $N_2O$ concentrations (Fig. 3a) ranged from 9.1 nM (Elbe-km 737) to 38.0 nM (Elbe-km 628.04), with a mean ± SD concentration of 18.0 ± 6.5 nM and a median concentration of 16.6 nM (see also Schulz et al., 2023).

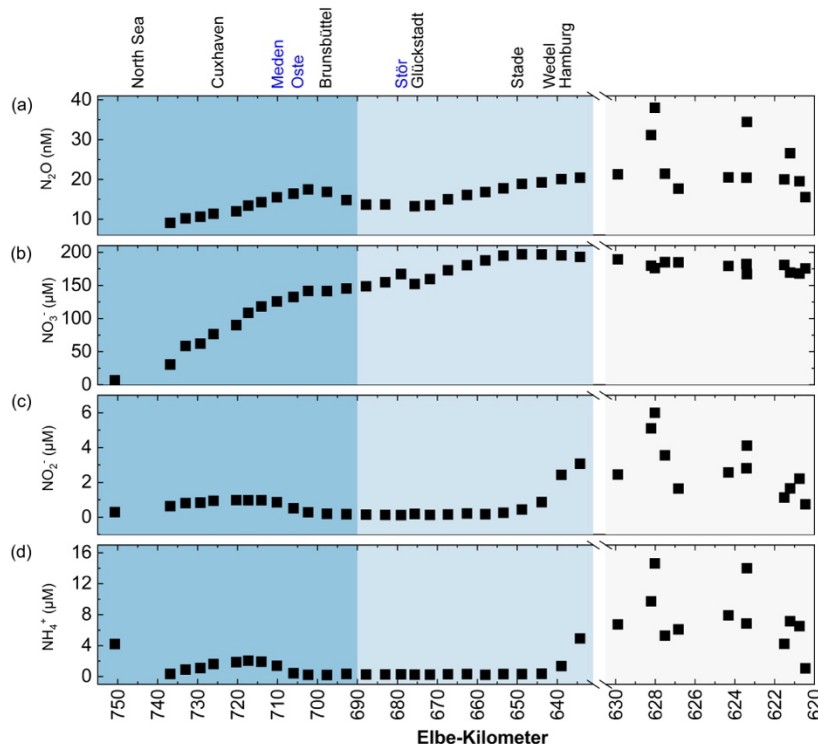

**Figure 3: Scatter plot of concentrations of (a) N₂O, (b) NO₃⁻, (c) NO₂⁻, and (d) NH₄⁺ along the Elbe Estuary. We included the relative position of selected localities and tributaries along the Elbe Estuary, Meden, Oste, and Stör.**

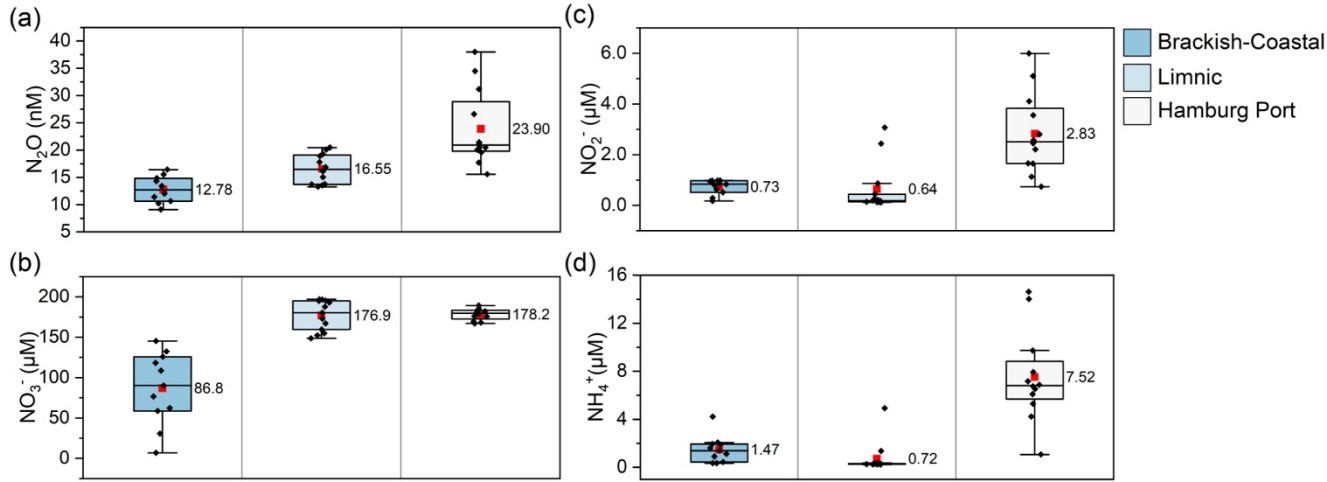

**Figure 4: Box plot of concentration of (a) N₂O, (b) NO₃⁻, (c) NO₂⁻, and (d) NH₄⁺ in each salinity zone along the Elbe Estuary. The boxes represent the interquartile range (IQR), with the lower and upper whiskers extending to the 25th and 75th percentiles, respectively. The median line is shown in the center of each box, and the mean is represented by a red square.**

Enhanced N₂O concentrations (mean: 23.9 ± 7.1 nM) were measured in the Hamburg Port area (Fig. 4a). Notable higher $N_2O$ concentrations were observed in the Hamburg Port area at Elbe-km 628.04, 628.21, and 623.40 (Fig. 3a) where the minimum $O_2$ concentrations were measured (Fig. 2c). At these locations, dissolved $N_2O$ concentrations exceeded 30 nM. The $N_2O$ were supersaturated at 440 %, 361 %, and 401 %, respectively, with corresponding fluxes of 131, 116, and 133 µmol m⁻² d⁻¹ (data are taken from Schulz et al., 2023).

The DIN (Fig. 2f) generally increased from the mouth of the estuary upstream until it reached around 200 µM in the Hamburg Port area. The trend was driven by its primary component, $NO_3^-$, which reached its maximum of 196.89 µM in the limnic zone at Elbe-km 649 before entering the Hamburg Port area (Fig. 3b). Upstream of this point, the $NO_3^-$ concentrations were lower. $NO_2^-$ and $NH_4^+$ concentrations closely followed similar trends (Figs. 3c and 3d), with slightly higher mean concentrations in the brackish–coastal zone than in the limnic zone (Figs. 4c and 4d). An increase in $NO_2^-$ and $NH_4^+$ concentrations was also observed downstream of the maximum turbidity zone (Dähnke et al., 2022) at the confluence of River Oste and Meden. $NO_2^-$ and $NH_4^+$ concentrations increased from Elbe-km 650 to the Hamburg Port area, where significant variability in their concentrations was observed. The spikes in $NO_2^-$ and $NH_4^+$ concentration coincided with those in $N_2O$ concentration at the Hamburg Port basins at Elbe-km 623.40, 628.04, and 628.21 (Fig. 3).

### 3.3 Dissolved NO concentrations, saturation ratios, and flux densities

NO concentrations in surface water of the Elbe Estuary (from Elbe-km 737 to Elbe-km 620, n=35) ranged from < LOD to 17.7 pM, with a mean ± SD concentration of 12.5 ± 1.9 pM and a median concentration of 12.1 pM (Fig 5a). Near the mouth of the estuary, the NO concentrations of five samples were below the detection limit. Concentrations started to increase slightly above the detection limit at the outflow of the River Meden near Otterndorf at Elbe-km 710 and 714. The measured NO concentration remained steady at around 12.0–13.0 pM in the limnic zone of the estuary, with a slightly enhanced concentration at Elbe-km 676 (13.7 pM), a sampling site in the port of Glückstadt.

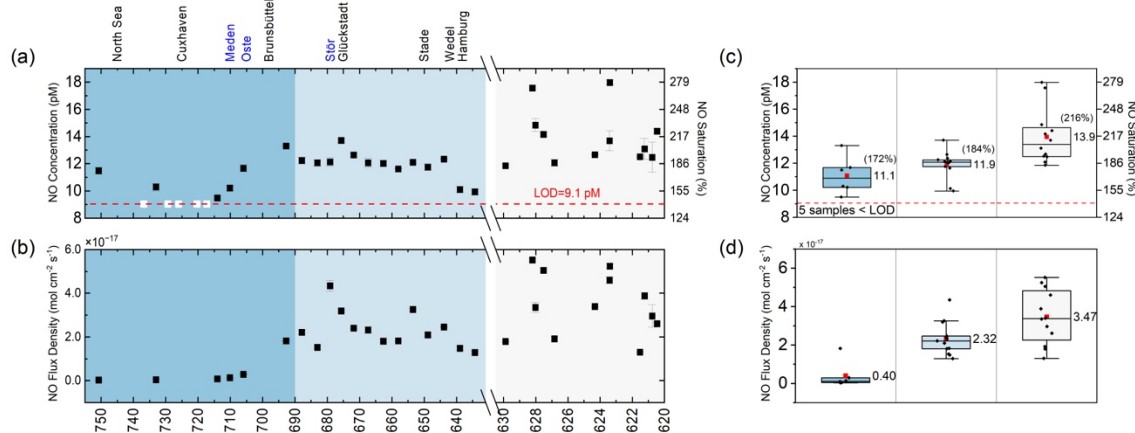

**Figure 5: Scatterplot of (a) dissolved NO concentration (pM) and saturation (%) and (b) sea–to–air flux density (mol cm⁻² s⁻¹) estimates along the Elbe Estuary. Boxplot of (c) dissolved NO concentration (pM) and saturation (%) and (d) sea–to–air flux density**

**estimates along the Elbe Estuary. Note that the mean atmospheric NO concentration of 4.30 μg m$^{-3}$ at seven background air monitoring stations in Hamburg located proximate to the Elbe Estuary was used to estimate the flux density. We included the relative position of selected localities and tributaries along the Elbe Estuary, Meden, Oste, and Stör. The error bars indicate the standard standard deviation of the replicates for a and b and the samples for c and d.**

Further upstream, high NO concentrations in the Hamburg Port area were observed, with peaks of 17.7 pM at Elbe-km 623.40 and 17.6 pM at Elbe-km 628.21. Generally, average NO concentrations increased as salinity decreased (Fig. 5c), with a mean ± SD concentration of 11.1 ± 1.4 pM in the brackish-coastal zone, 11.9 ± 1.1 pM in the limnic zone, and 13.9 ± 2.0 pM in the

250 Hamburg Port area (Fig. 5c).

The NO saturation values (excluding < LOD) ranged from 147 to 274 %, with mean saturation values of 172 %, 184 %, and 216 % in the brackish-coastal zone, limnic zone, and the Hamburg Port area, respectively (Fig. 5c). The overall mean ± SD and median NO saturations in the surface layer of the Elbe Estuary were 194 ± 29 % and 189 %, respectively.

Moreover, the NO flux density (excluding < LOD) ranged from $3.1 \times 10^{-19}$ to $5.5 \times 10^{-17}$ mol cm$^{-2}$ s$^{-1}$, with overall mean ±

SD and median flux densities of 2.4 $(\pm 1.5) \times 10^{-17}$ mol cm$^{-2}$ s$^{-1}$ and $1.58 \times 10^{-17}$ mol cm$^{-2}$ s$^{-1}$, respectively (Fig. 5b). The mean NO flux densities also generally increased as salinity decreased (Fig. 5d).

## 4. Discussion

### 4.1 NO concentrations, saturations, and flux densities

Figure 6 shows a compilation of average dissolved NO concentrations and estimated flux densities from previous studies. The

260 measured concentrations from the previous studies were highly variable and ranged between 5.9 and 260 pM. Previous studies suggest that the ocean could potentially act as a significant source of NO to the atmosphere. Early studies by Zafiriou and McFarland (1981) already reported supersaturation of up to a factor of $10^4$ in the central equatorial Pacific Ocean. More recent studies performed since 2010 reported supersaturation from 100 to 10,000%. Up to now, the majority of the literature reports that NO production causes supersaturation in the surface water and positive sea-to-air flux density, indicating emissions as a

265 major sink; however, regional exceptions, such as one measurement in the Shandong Peninsula (Gong et al., 2023), indicate that generalizations should be made cautiously. The values greatly vary temporally and spatially across the studied sites, and no spatiotemporal patterns or variability have been established yet.

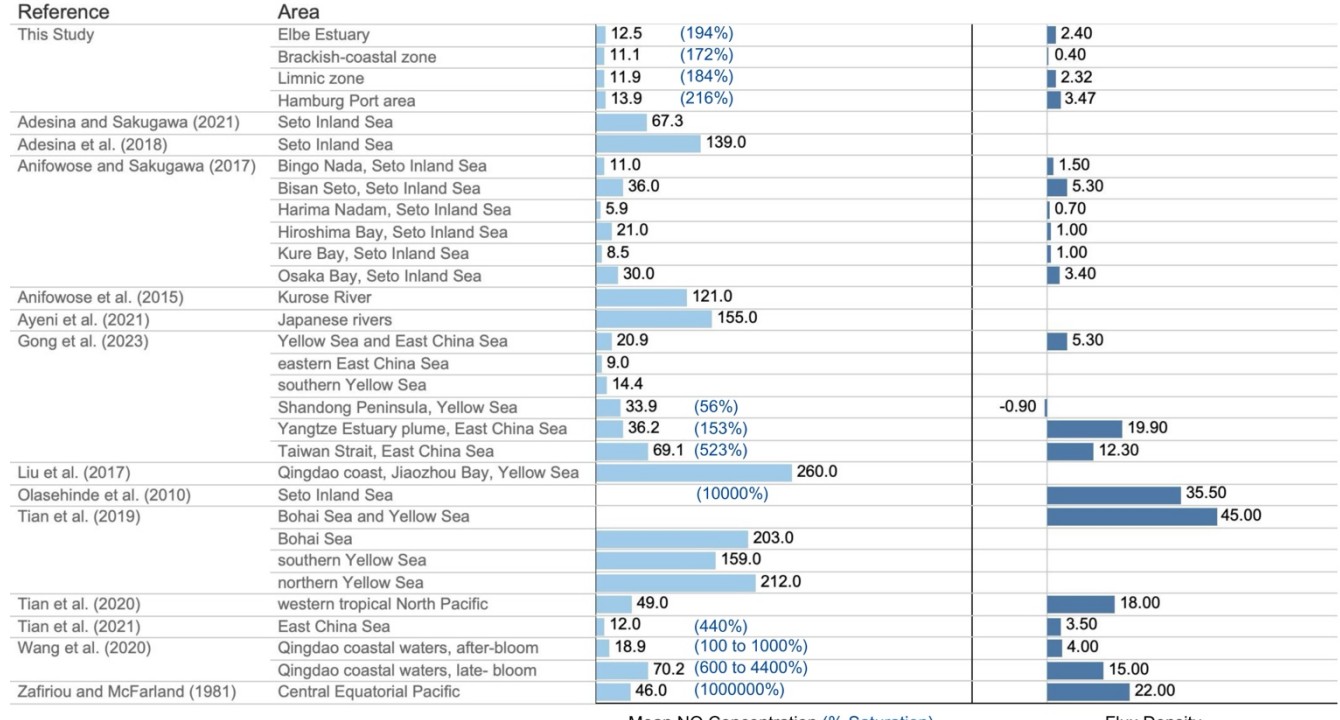

| Reference | Area | Mean NO Concentration (% Saturation) | Flux Density |
|---|---|---|---|
| This Study | Elbe Estuary | 12.5 (194%) | 2.40 |
| | Brackish-coastal zone | 11.1 (172%) | 0.40 |
| | Limnic zone | 11.9 (184%) | 2.32 |
| | Hamburg Port area | 13.9 (216%) | 3.47 |
| Adesina and Sakugawa (2021) | Seto Inland Sea | 67.3 | |
| Adesina et al. (2018) | Seto Inland Sea | 139.0 | |
| Anifowose and Sakugawa (2017) | Bingo Nada, Seto Inland Sea | 11.0 | 1.50 |
| | Bisan Seto, Seto Inland Sea | 36.0 | 5.30 |
| | Harima Nadam, Seto Inland Sea | 5.9 | 0.70 |
| | Hiroshima Bay, Seto Inland Sea | 21.0 | 1.00 |
| | Kure Bay, Seto Inland Sea | 8.5 | 1.00 |
| | Osaka Bay, Seto Inland Sea | 30.0 | 3.40 |
| Anifowose et al. (2015) | Kurose River | 121.0 | |
| Ayeni et al. (2021) | Japanese rivers | 155.0 | |
| Gong et al. (2023) | Yellow Sea and East China Sea | 20.9 | 5.30 |
| | eastern East China Sea | 9.0 | |
| | southern Yellow Sea | 14.4 | |
| | Shandong Peninsula, Yellow Sea | 33.9 (56%) | -0.90 |
| | Yangtze Estuary plume, East China Sea | 36.2 (153%) | 19.90 |
| | Taiwan Strait, East China Sea | 69.1 (523%) | 12.30 |
| Liu et al. (2017) | Qingdao coast, Jiaozhou Bay, Yellow Sea | 260.0 | |
| Olasehinde et al. (2010) | Seto Inland Sea | (10000%) | 35.50 |
| Tian et al. (2019) | Bohai Sea and Yellow Sea | | 45.00 |
| | Bohai Sea | 203.0 | |
| | southern Yellow Sea | 159.0 | |
| | northern Yellow Sea | 212.0 | |
| Tian et al. (2020) | western tropical North Pacific | 49.0 | 18.00 |
| Tian et al. (2021) | East China Sea | 12.0 (440%) | 3.50 |
| Wang et al. (2020) | Qingdao coastal waters, after-bloom | 18.9 (100 to 1000%) | 4.00 |
| | Qingdao coastal waters, late-bloom | 70.2 (600 to 4400%) | 15.00 |
| Zafiriou and McFarland (1981) | Central Equatorial Pacific | 46.0 (1000000%) | 22.00 |

**Figure 6: Comparison of the mean dissolved NO concentration (pM), saturation values (%), and estimated sea–to–air flux density (× 10⁻¹⁷ mol cm⁻² s⁻¹) from previous studies and the present study.**

In our study, the mean NO concentration was $12.5 \pm 1.9$ pM, which is at the lower end of the range of previously published measurements from the marine environment. Despite these low concentrations, NO was supersaturated in the surface layer of the Elbe Estuary (see Section 3.3), indicating that the Elbe Estuary, particularly the Hamburg Port area, was a source of NO to the atmosphere during the study period. The mean estimated flux density in this study is close to the average flux density from the Kurose River reported by Anifowose and Sakugawa (2017) but at the lower end of flux densities reported from shelf and open ocean waters.

Our result is consistent with previous studies reporting strong supersaturation of NO in surface seawater (Fig. 6). Olasehinde et al. (2010) and Adesina and Sakugawa (2021) have demonstrated that even at extremely low NO concentration, NO production in surface seawater is sufficient to cause strong supersaturation. Both studies, citing Jacobi and Hilker (2007), discussed that because of the low solubility of NO in (sea)water or aqueous solution, NO can be transferred to the gas phase. The observed variations in NO concentrations and flux densities along the salinity gradient indicate potential changes in NO production pathways and controlling factors that influence NO distribution in the entire Elbe Estuary and at each salinity zone, which we discuss in the following sections. The correlation analysis between NO and various biogeochemical parameters is presented in Table S3.

## 4.2 Influence of salinity and freshwater input on NO concentrations

Along the salinity gradient, the average NO concentrations tended to increase from the North Sea towards the Elbe River (Section 3.3). Although this negative correlation is not statistically significant (Table S3), the negative slope suggests a potential inverse relationship, in which a decrease in salinity appears to coincide with an increase in NO concentration, which could potentially be attributed to DIN input from the Elbe River. This finding has been observed in prior studies by Gong et al. (2023), Adesina et al. (2021), and Tian et al. (2019), which have identified a similar negative trend between dissolved NO concentrations and salinity.

Specifically, Gong et al. (2023) found that salinity had a significant negative correlation with dissolved NO concentration ($r=-0.44$, $p<0.05$, $n=13,326$), while Adesina et al. (2021) reported an even stronger correlation for steady-state NO concentrations versus salinity ($r=-0.83$, $p<0.01$). Furthermore, Adesina et al. (2018) reported a negative correlation between salinity and NO photochemical generation rates ($r = -0.504$, $p > 0.05$). Meanwhile, Tian et al. (2019) observed an inverse relationship between salinity and surface dissolved NO concentrations for stations affected by the outflow of the Yellow River in the southern Bohai Sea and ascribed it to high DIN input.

In the Kurose River, Ayeni et al. (2021) examined the spatial variability of NO concentrations and identified a gradient from low concentrations in the upstream sections to higher concentrations in the downstream sections, the latter being notably influenced by anthropogenic activities. Moreover, Gong et al. (2023) discussed that NO distribution is likely influenced by freshwater inputs due to the ready availability of precursor DIN substrates. Nonetheless, similar to what we observed, Gong et al. (2023) noted that salinity is insufficient to explain the uneven distribution of NO at their study site, indicating that other parameters influence NO concentrations along the Elbe Estuary.

## 4.3 Influence of dissolved inorganic nitrogen on NO concentrations

There are two widely known sources of NO in surface seawater: $NO_2^-$ photolysis and biological production. It is well reported that $NO_2^-$ is the primary source of NO in seawater through photolysis (Treinin and Hayon, 1970; Zafiriou et al., 1980; Anifowose et al., 2015) as shown in R1 below:

$$NO_2^- + H_2O + hv \rightarrow NO + OH + NO^- \text{ where } hv = 295 \leq \lambda \leq 410 \text{ nm} \qquad (R1)$$

However, the mechanism of how nitrogen-containing nutrients ($NO_3^-$, $NO_2^-$, $NH_4^+$) and their cycling affect the dissolved NO distribution in aquatic environments remains unresolved. Gong et al. (2023) argued that nitrogen-containing nutrients may serve as the substrate or intermediate for photochemical and microbial NO production; thus, high concentrations of $NO_3^-$, $NO_2^-$, and $NH_4^+$ ensure that the necessary conditions for NO production are met.

In this study, the $NO_3^-$, $NO_2^-$, and $NH_4^+$ concentrations were higher than those found in previous studies in the river and coastal areas (e.g., Ayeni et al., 2021; Gong et al., 2023; Liu et al., 2017); however, the elevated nitrogenous nutrient concentrations did not correspond to a higher average dissolved NO concentration in the Elbe Estuary. One of the plausible reasons is the high turbidity in the water column, which may influence NO photoproduction rates. The Elbe Estuary is considered "heavily

modified" in the European Water Framework Directive due to river modifications brought by shipping, harbor operation, and flood defense (Taupp and Wetzel, 2013; Netzband et al., 2002). Annually, 15–20 million $m^3$ of sediment has to be dredged in the tidal Elbe to maintain the depth required for navigation, causing high turbidity in the water column (Taupp and Wetzel, 2013; Heininger et al., 2015). Future research is recommended to determine whether turbidity and suspended particulate matter influence NO production in estuarine and coastal systems.

Our observation challenges the assumption that higher concentrations of nitrogen nutrients automatically lead to increased dissolved NO concentration, indicating that other factors may play significant roles in regulating NO concentrations in coastal and estuarine environments.

Some studies (e.g., Olasehinde et al., 2010; Anifowose et al., 2015; Anifowose and Sakugawa, 2017; Ayeni et al., 2021; Gong et al., 2023) observed positive correlations between NO concentrations or photoproduction rates and $NO_2^-$ concentrations. In contrast, other studies (Tian et al., 2020, 2021) did not observe any relationship between surface NO distribution and concentrations of $NO_3^-$, $NO_2^-$, and $NH_4^+$.

While Gong et al. (2023) argued that nitrogen-containing nutrients may serve as the substrate or intermediate for photochemical and microbial NO production, they also noted areas in their study site (i.e., the nearshore region of the Shandong Peninsula) where the DIN concentrations were significantly lower yet had high surface dissolved NO concentration, attributing it to the uptake of atmospheric NO into the surface layer. Likewise, Ayeni et al. (2021) also noted that some rivers in Japan with higher $NO_2^-$ concentrations had lower rates of photoproduction of NO and vice versa, attributing these imbalances to nitrogen cycling processes (nitrification, denitrification, and anammox), which could produce or consume NO, or the photochemical transformation of organic nitrogen from dissolved organic matter producing $NO_2^-$ to form NO in areas with low $NO_2^-$.

**Table 1: Pearson correlation coefficient (R) between NO and some nitrogen parameters at each salinity zone. Significant correlations are denoted in bold font. Note that the superscripts after the R values indicate a significant correlation at ***$p$ < 0.001, **$p$ < 0.01, and *$p$ < 0.05.**

|  | Overall | Coastal-Brackish | Limnic | Hamburg Port |
|---|---|---|---|---|
| DIN ($\mu M$) | 0.3264 | 0.1630 | **–0.6576*** | 0.0677 |
| $NO_3^-$ ($\mu M$) | 0.2552 | 0.1715 | **–0.5998*** | –0.3726 |
| $NO_2^-$ ($\mu M$) | **0.5425**** | **–0.9419**** | **–0.8380***** | **0.6570*** |
| $NH_4^+$ ($\mu M$) | **0.6051***** | –0.2267 | **–0.7323**** | 0.5558 |
| $NO_2^- + NH_4^+$ ($\mu M$) | **0.6005***** | –0.4396 | **–0.8011***** | **0.6060*** |
| $NO_2^-/O_2$ ratio | **0.5692***** | **–0.9212**** | **–0.8404***** | **0.6711*** |
| $N_2O$ (nM) | **0.6609***** | 0.3164 | **–0.7527**** | **0.6940*** |
| $N_2O/NO_2^-$ ratio | –0.1196 | **0.9422*** | **0.6064*** | –0.2644 |
| $N_2O/NH_4^+$ ratio | –0.1833 | **0.9589**** | **0.7569**** | 0.0195 |
| $N_2O/(NO_2^- + NH_4^+)$ ratio | –0.1550 | **0.9881**** | **0.7014*** | –0.0409 |

Shown in Table 1 is a correlation analysis between NO and the DIN substrates. Using the entire data set (i.e., "Overall"), we observed a significant positive correlation between NO and $NO_2^-$, $NH_4^+$, the sum of $NO_2^-$ and $NH_4^+$, $N_2O$, and $NO_2^-/O_2$ ratio.

Drawing solely from these findings, one might infer that elevated concentrations of $NO_2^-$ and $NH_4^+$ invariably lead to an increase in NO concentration. However, this is not a consistent observation across all salinity zones. We found that there is a significant negative relationship ($p < 0.05$) between the concentrations of NO and DIN, $NO_3^-$, $NO_2^-$, $NH_4^+$, sum of $NO_2^-$ and $NH_4^+$, and $N_2O$ in the limnic zone, and between NO and $NO_2^-$ in the coastal-brackish zone; whereas there is a significant positive correlation ($p < 0.05$) between concentrations of NO and $NO_2^-$, $N_2O$, and sum of $NO_2^-$ and $NH_4^+$ in the Hamburg Port area (Table 1). In the next section, we will try to explain the different processes that may have contributed to this trend.

## 4.4 Biological production of NO

We explored the possibility of NO production from phytoplankton (e.g., Wang et al., 2020; Kim et al., 2006) as NO may be generally consumed or produced by phytoplankton while they bloom and/or in response to environmental stress and pollution (Burlacot et al., 2020; Estevez and Puntarulo, 2005; Mallick et al., 2002; Zhang et al., 2006). However, we noted that chlorophyll a concentrations (a proxy for phytoplankton biomass) were minimal ($< 6.50$ µg $L^{-1}$) at areas where the NO concentration peaked (i.e., Elbe 623.40 and 628.04), suggesting that phytoplankton bloom may not be a major factor contributing to higher dissolved NO concentration in our study site. Similar to the findings of Tian et al. (2021), the NO distribution in the Elbe Estuary is not directly related to the distribution of chlorophyll a ($p > 0.05$, Table S3).

As discussed earlier, Ayeni et al. (2021) suggested that nitrogen cycling processes (nitrification, denitrification, and anammox) may influence NO distribution in river systems. These processes are simplified in R2 to R4 (see Kuypers et al., 2018). In the following sections, we investigated how these various nitrogen cycling processes may have influenced NO concentrations in the Elbe estuary.

$$\text{Nitrification: } NH_4^+ \xrightarrow{\text{ammonium monooxygenase}} NH_2OH \xrightarrow{\text{hydroxylamine oxidation}} \mathbf{NO} \longrightarrow NO_2^- \xrightarrow{\text{nitrite oxidoreductase}} 2NO_3^- \quad (R2)$$

with $N_2O$ branching upward ($\uparrow$) from $\mathbf{NO}$.

$$\text{Denitrification: } NO_3^- \xrightarrow{\text{nitrate reductase}} NO_2^- \xrightarrow{\text{nitrite reductase}} \mathbf{NO} \xrightarrow{\text{nitric oxide reductase}} N_2O \xrightarrow{\text{nitrous oxide reductase}} N_2 \quad (R3)$$

$$\text{Anammox: } NO_2^- + NH_4^+ \xrightarrow{\mathbf{NO},\ \text{hydrazine synthase}} N_2H_4 \xrightarrow{\text{hydrazine dehydrogenase}} N_2 \quad (R4)$$

### 4.4.1 Nitrification in the Elbe Estuary

Recently, Gong et al. (2023) noted nitrification as the major contributor to microbial production of NO in the coastal seas of the Yellow and East China Seas. Nitrification is an aerobic two-step process in which $NH_4^+$ is oxidized to nitrate (R2). In this process, $N_2O$ is assumed to be a byproduct and NO is reported to be an obligate intermediate of bacterial nitrification produced by hydroxylamine oxidoreductase (Caranto and Lancaster, 2017).

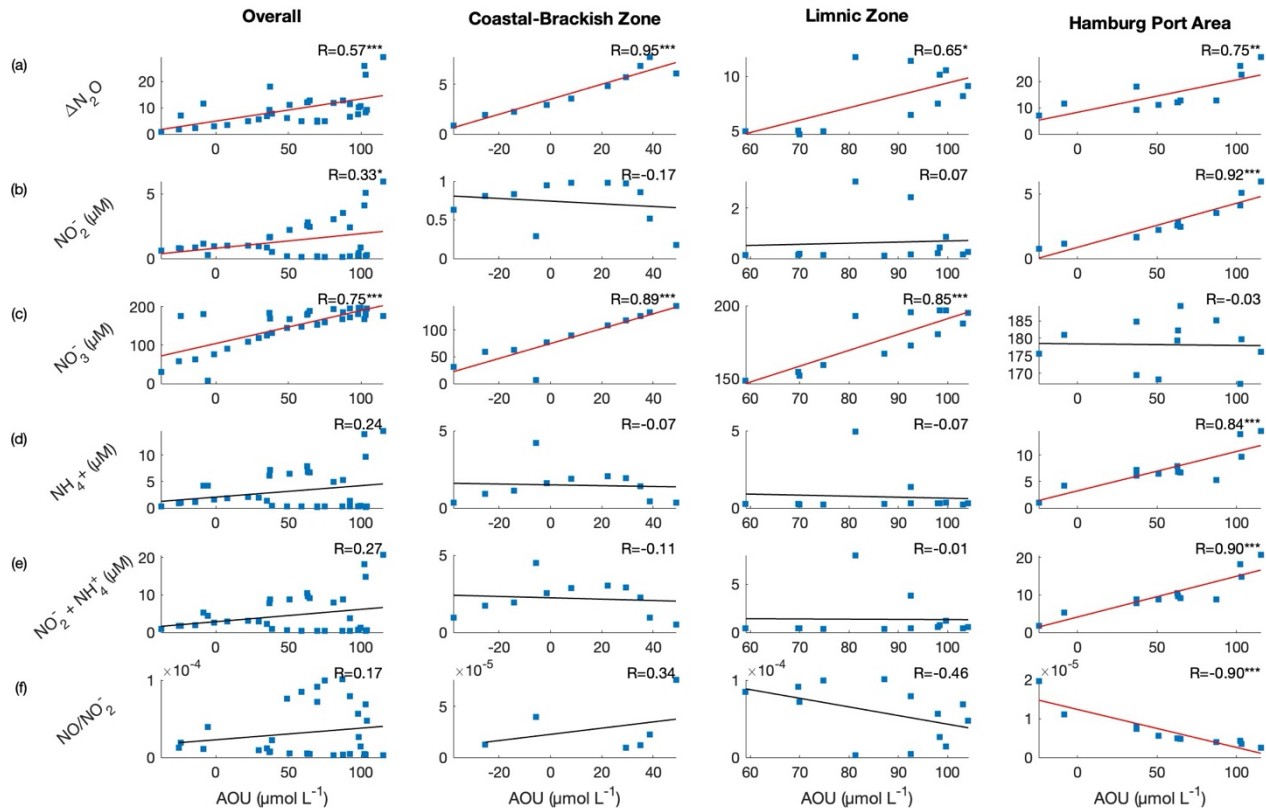

**Figure 7: Scatter plots between AOU and $\Delta N_2O$, $NO_2^-$ (μM), $NO_3^-$(μM), $NH_4^+$(μM), sum of $NO_2^- + NH_4^+$ (μM), and $NO/NO_2^-$ ratio. Note that the superscripts on the R-value indicate a significant correlation at $^{***}p < 0.001$, $^{**}p < 0.01$, and $^{*}p < 0.05$. Significant correlations are denoted with a red regression line. We observed significant correlations between AOU and $NO_3^-$ in the coastal-brackish and limnic zones, indicating nitrification as a dominant process.**

In the Elbe Estuary, nitrification is reported as the main production pathway of $N_2O$ (Brase et al., 2017; Schulz et al., 2023). As shown in Fig. 7, AOU had a significant positive linear relationship with $\Delta N_2O$ in the entire stretch of the Elbe Estuary— in the brackish-coastal zone (R=0.95, p < 0.001), limnic zone (R=0.65, p < 0.05), and in the Hamburg Port area (R=0.75, p < 0.01). Additionally, a significant positive linear correlation was noted between $N_2O$ and $NO_3^-$ (see Walter et al., 2006) in the brackish coastal zone (R=0.96, p<0.001) and in the limnic zone (R=0.94, p<0.001). Previous studies (Yoshinari, 1976; Yoshida et al., 1989; Nevison et al., 2003; Walter et al., 2006) established that a significant positive linear relationship between AOU and $\Delta N_2O$, as well as $N_2O$ and $NO_3^-$, indicates $N_2O$ production from nitrification; Brase et al. (2017) and Schulz et al. (2023) also observed these linear correlations in the Elbe Estuary.

If $N_2O$ levels rise with $NO_3^-$ levels, it could indicate active nitrification (see Schulz et al., 2023), during which NO is produced as an intermediate (Caranto and Lancaster, 2017). The nitrification process can be observed in the brackish-coastal and limnic zones of the Elbe Estuary; we also observed significant inverse correlations in these two zones – the significant inverse

relationship between NO vs $NO_2^-$ and NO vs $NO_2^-/O_2$ (Figs. 8c and 8f). We speculate this could be partly attributed to $NH_4^+$

limitation (see Fig. 4) and high $O_2$ concentration. The well-oxygenated conditions in the coastal-brackish and limnic zones facilitate the aerobic process of nitrification. In the nitrification process (R2), $NH_4^+$ serves as the initial substrate. We observed in the limnic zone that the NO concentration was higher when $NH_4^+$ was low (Fig. 8d); $NH_4^+$ is consumed as it is oxidized to NO. As NO underwent further oxidation to produce $NO_2^-$, NO concentration decreased while $NO_2^-$ increased. This plausibly explains the negative correlation between NO vs. $NO_2^-$ and $NO_2^-/O_2$ ratio that we observed in our study. Furthermore, we

observed that NO concentration can also be explained by the ratio of $N_2O$, the byproduct of nitrification, and $NH_4^+$, the initial reactant of nitrification, and $NO_2^-$, the oxidized product of NO, as shown in Figs. 8h and i.

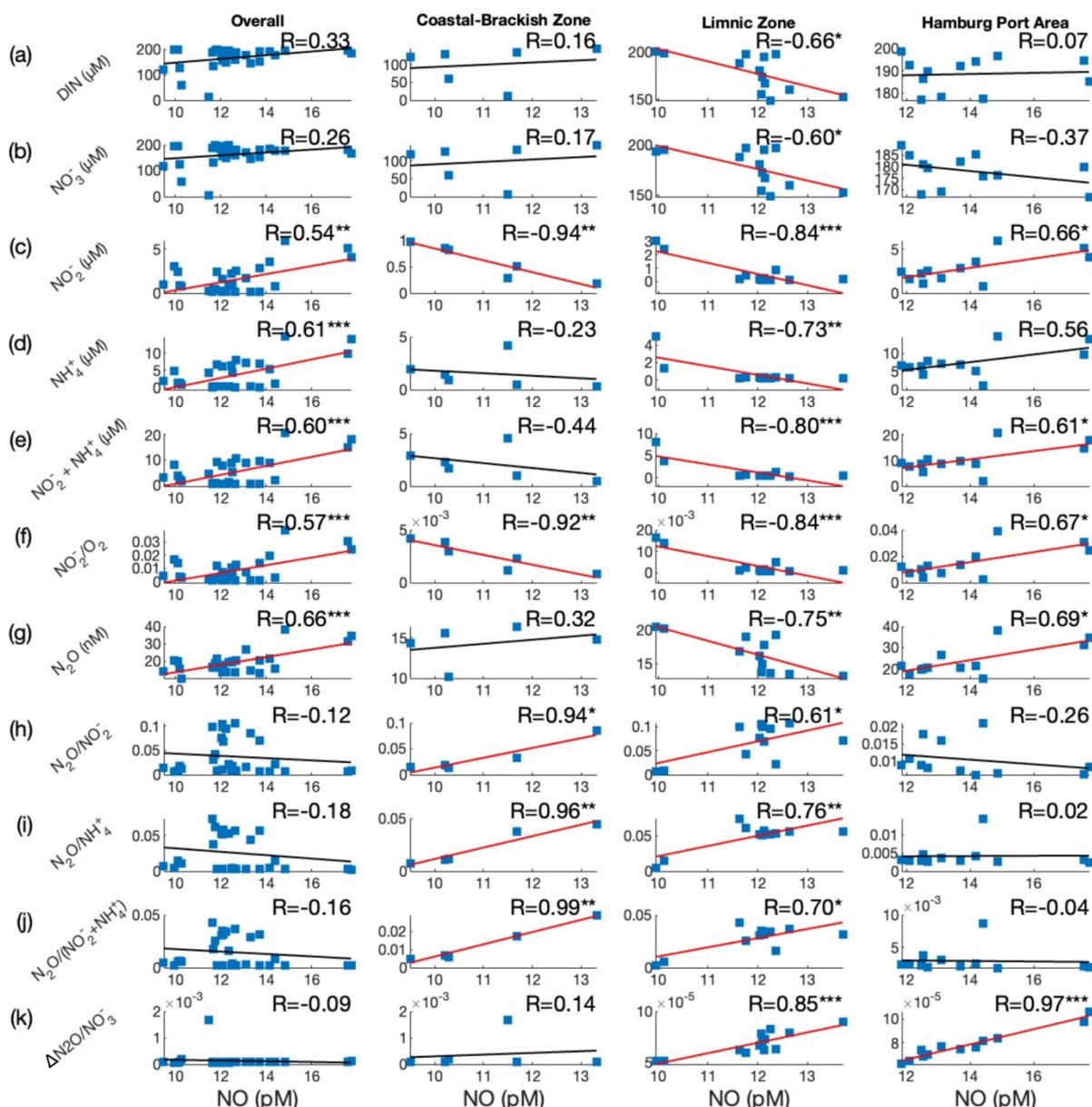

**Figure 8: Scatter plots between NO and DIN, NO₃⁻(μM), NO₂⁻ (μM), NH₄⁺(μM), sum of NO₂⁻ + NH₄⁺ (μM), NO/NO₂⁻ ratio, N₂O, N₂O/NO₂⁻, N₂O/NH₄⁺, and N₂O/ NO₂⁻ + NH₄⁺, and ΔN₂O/NO₃⁻. Note that the superscripts on the R-value indicate a significant correlation at ***p < 0.001, **p < 0.01, and *p < 0.05. Significant correlations are denoted with a red regression line.**

### 4.4.2 Nitrogen transformation processes in the Hamburg Port area

In contrast to the brackish-coastal and limnic zones, where $NH_4^+$ and $NO_2^-$ concentrations are low (see Section 3.2), the concentrations of $NH_4^+$ and $NO_2^-$ (>1 μM) in the Hamburg Port area were elevated (Figs. 4c and 4d). $NH_4^+$ and $NO_2^-$ may

primarily come from anthropogenic sources like wastewater discharge, agricultural runoff, industrial effluents, and remineralization of suspended particulate matter or organic matter (Wolfstein and Kies, 1999; Kerner, 2000). Sanders et al. (2018) reported that the phytoplankton growth in the upstream section of the Elbe Estuary, which produces degradable organic matter, plays an important role in the remineralization and nitrification processes in the upper part of the Elbe Estuary (i.e., Hamburg Port area). The study noted that remineralization provides the substrate $NH_4^+$ necessary for nitrification. If $NH_4^+$ is not limited or has a continuous supply in the nitrification reaction (R2), one would expect a direct relationship between NO and $NO_2^-$. Indeed, we noted a significant correlation between the concentrations of NO vs. $NO_2^-$ and NO vs. $NO_2^-/O_2$ (Table 1, see also Fig. 8c and Fig. 8f). Aside from correlation with $NO_2^-$ and NO vs. $NO_2^-/O_2$, we also noted a significant correlation between NO and the sum of $NO_2^-$ and $NH_4^+$, $N_2O$, and $\Delta N_2O/NO_3^-$. This means that the distribution of NO concentration in the Hamburg Port area can be partly explained by processes involving these factors.

Despite the strong correlation between AOU and $\Delta N_2O$ in the Hamburg Port Area, we observed no significant correlation between AOU and $NO_3^-$. We think that this lack of correlation between AOU vs $NO_3^-$ may be brought by other nitrogen transformation processes that influence $NO_3^-$ concentration or that affect $NO_2^-$ oxidation, such as nitrifier-denitrification, denitrification (R3), anammox (R4), and/or primary production. Previous studies reported that the Hamburg Port area is a hotspot for $N_2O$ production, attributed to nitrification and nitrifier-denitrification processes (Brase et al., 2017). Prior studies confirmed the highest denitrification rates in the sediments (Deek et al., 2013) and the highest nitrification rates in the water column at this section of the Elbe Estuary (Sanders et al., 2018). During this study, we didn't have the tools to distinguish the exact process involved. However, future studies are recommended to utilize dual stable isotope techniques and molecular or genetic tools to detect marker genes specific to nitrogen-cycling microorganisms.

### 4.5. Influence of dissolved oxygen on NO

Dissolved oxygen, which was mainly influenced by primary productivity and respiration (see Figs. 2c–e), played a significant role in the distribution of nitrogen compounds. In this study, we noted significant negative correlations ($p < 0.0001$) between $O_2$ and $NO_2^-$, $NH_4^+$, and $N_2O$ (Fig. S6). Moreover, distinct peaks of $NO_2^-$ (> 4 µM) and $NH_4^+$ (>9.5 µM) were measured at the sampling sites in the Hamburg Port area at Elbe-km 628.04, 628.21, and 623.40, with the lowest $O_2$ concentrations (<150 µM) (Fig. 3). In this sampling locations, relatively higher concentrations of NO (>14 pM) and $N_2O$ (>30 µM) were also measured. At these sampling stations, the $N_2O$ and NO saturations were exceedingly high, reaching values over 360% and 270%, respectively. These high NO and $N_2O$ saturations are notable, as they suggest a significant level of production.

The measured $O_2$ concentrations in the Hamburg Port area could inhibit anaerobic nitrogen processes such as denitrification or anammox in the water column. Nevertheless, oxygen-limited conditions often found in sediments (Schroeder et al., 1991; Deek et al., 2013) and within biofilms or anoxic microsites on suspended particles (Liu et al., 2013; Xia et al., 2017) in estuaries provide suitable microenvironments for anaerobic processes to occur. For instance, nitrifying bacteria may switch from nitrification to nitrifier-denitrification (see Schulz et al., 2022; Dai et al., 2008) in these environments. We noted that both

NO and $N_2O$ concentrations started to increase downstream of the maximum turbidity zone near the confluence of River Meden and Stör.

Moreover, previous studies (Schroeder, 1997; Sanders et al., 2018; van Beusekom et al., 2021; Brase et al., 2017) indicated that low $O_2$ conditions may develop in the Hamburg Port area due to its geomorphological features and high nutrient content, particularly from runoff and remineralization process (Kerner and Spitzy, 2001). During eutrophication, increased nutrient availability stimulates algal growth, leading to $O_2$ depletion at night or daybreak, as algae consume $O_2$ through respiration. As the algal blooms eventually die off and decompose (Goosen et al., 1995), microbial processes like nitrifier-denitrification and denitrification thrive under low $O_2$ conditions, potentially releasing NO and $N_2O$. These biological processes are important in shaping the biogeochemical profile of the estuary, with photosynthesis contributing to peaks in $O_2$ and chlorophyll a during daylight hours and respiration leading to $O_2$ depletion and potentially creating favorable conditions for $N_2O$ and NO production during nighttime or in less oxygenated microenvironments such as suspended sediments or particulate matter (Schulz et al., 2022). Future studies on the influence of primary productivity and respiration on $O_2$ conditions and the NO production or consumption processes in estuaries are recommended.

Further investigation is necessary to elucidate the potential seasonal variability in NO concentrations within the Elbe Estuary. Seasonal fluctuations significantly influence $NO_2^-$ and $NH_4^+$ loading (Malinowski et al., 2020), particularly during the spring and summer when agricultural practices intensify, and nitrogen-based fertilizers are used extensively (Pastuszak et al., 2018). These factors contribute to increased $NO_2^-$ and $NH_4^+$ concentrations in the estuary, primarily due to heightened surface runoff and leaching from agricultural areas. Additionally, in the Hamburg Port area, the decomposition of phytoplankton coming from the upstream Elbe River may contribute to increased $NO_2^-$ and $NH_4^+$ concentrations due to remineralization. Sanders et al. (2018) reported that nitrification varied seasonally and was linked to the remineralization of this organic matter. This could subsequently lead to $O_2$ depletion, especially during the warmer summer months, coinciding with increased microbial activity that could intensify nitrogen transformation processes (Schulz et al., 2023; Sanders et al., 2018). No comprehensive study to date has examined the seasonal dynamics of NO concentrations in such contexts.

## 5. Conclusion

Our study provides the first measurement of dissolved NO in the Elbe Estuary, shedding light on the potential sources and processes driving NO production in this area. We observed variations in NO concentrations and flux densities along the salinity gradient, with elevated levels in the Hamburg Port area. During this campaign, the surface water of the lower Elbe Estuary and Hamburg Port area was supersaturated. During the time of sampling, the Elbe Estuary was a source of NO to the atmosphere with a mean flux density of ($\pm$ SD) of 2.40 ($\pm$1.54) $\times$ $10^{-17}$ mol cm$^{-2}$ s$^{-1}$. Notably, the Hamburg Port area showed a higher mean flux density of 3.47 ($\pm$1.43) $\times$ $10^{-17}$ mol cm$^{-2}$ s$^{-1}$. Furthermore, areas with higher $NO_2^-$ and $NH_4^+$ concentrations, which primarily come from anthropogenic sources like wastewater discharge, agricultural runoff, and industrial effluents, exhibited elevated dissolved NO concentrations.

Excess nutrients may also result in algal blooms that can lower $O_2$ concentration and favor processes that produce NO and $N_2O$. Moreover, despite nitrogen-containing nutrient concentrations in the Elbe Estuary being higher than those reported in previous studies in other regions, the NO concentrations remained low. This observation prompts further investigation into how nitrogen transformation processes could influence NO distribution in the Elbe Estuary. It is recommended that future research adopt a more comprehensive approach, incorporating both higher temporal resolution (covering diurnal and seasonal cycles) and spatially diverse sampling strategies, including stable isotope techniques, measurement of process rates through isotope tracers, and use of molecular or genetic tools to detect marker genes specific to nitrogen-cycling microorganisms. This combined approach will enable a more nuanced understanding of the dynamic interplay between various controlling factors influencing NO concentration in the coastal and estuarine environments.

## Acknowledgment

We would like to thank the captain and crew of the R/V Ludwig Prandtl for their assistance and support during the fieldwork. Likewise, we would also like to express our gratitude to our colleagues from the Helmholtz-Zentrum Hereon, who participated in the sampling and invited our research group to join this field campaign. We also thank Yoana Voynova and her working group for providing the FerryBox data. We would also like to thank Dr. Annette Kock, who assisted in the logistics of this field campaign, and Ms. Estela Monteiro for providing helpful comments on the earlier version of the manuscript. Riel Carlo O. Ingeniero was supported by DAAD Research Grants – Doctoral Programmes in Germany (57440921). Lastly, we gratefully acknowledge the anonymous reviewers for their insightful comments and suggestions, which significantly enhanced the quality of our paper and prepared it for publication.

## Data Availability Statement

The data sets generated and/or analyzed in this study can be accessed by contacting the corresponding author and will be made publicly available at coastMap Geoportal (www.coastmap.org), which connects to PANGAEA, with future DOI availability anticipated.

## Authors Contribution

RI conceptualized and designed the study, carried out field sampling and analysis, and authored the manuscript. GS managed the research cruise, performed laboratory analyses of supplementary biogeochemical parameters, and contributed a critical review of the manuscript. HB assisted with study conceptualization, supervision, and a critical review of the manuscript.

## Competing Interest

At least one of the (co-)authors is a member of the editorial board of Biogeosciences.

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
