# Peer review of "Dissolved Nitric Oxide in the Lower Elbe Estuary and the Hamburg Port Area"

_EGUsphere, 2023_

## Author Comment (AC1)

**Response to Reviewer 1**

*The authors present a recent effort of NO measurement in the Lower Elbe Estuary and the Hamburg Port Area, filling research blanks of this trace gas in coastal areas and estuaries. This manuscript is well-organized, with nice figures. It does provide an important picture of estuarine NO, an active trace gas difficult to measure, showing the distribution, flux, and potential production mechanisms of NO in the study region.*

*However, I have two major concerns here (also see specific comments below):*

1. *This paper is a good case study, but, as a manuscript expected to be published in bg, the text is lacking in the laying out of the scientific issues as well as extrapolation. For example, in the introduction there is a lack of elicitation of the gaps for the current research, and in the discussion, there is a lack of implications of the conclusions for other research in the field (i.e., what is the new knowledge compared to other published NO studies).*

2. *The whole discussion section and the present of implications is still weak, e.g., the main conclusions are mainly drawn through correlations, but without sufficient explanation and logic connection between correlation and their conclusion. This problem is particularly evident in section 4.4.*

*In the present version, I think there are still some gaps away from the publication level, and a major revision would be recommended.*

We thank the Reviewer for dedicating her/his/their time and effort to offer constructive feedback, which is instrumental in enhancing our manuscript. We acknowledge the reviewer's feedback to expound on the gaps in research on nitric oxide in the marine environment in our Introduction section. We will revise our manuscript to mention these gaps in our paper.

We acknowledge the reviewer's point that the discussion section requires strengthening, particularly in establishing a robust causal link between the observed correlations and our discussion/conclusions. Our approach was to interpret the available data in order to explain the patterns of NO distribution in the Elbe Estuary. In our revised manuscript, we will enhance the clarity of the manuscript, ensuring that the role of NO as an intermediate in the nitrogen cycle is comprehensively explained and clearly articulated. We will address points raised by the reviewers to enhance the discussion section.

The reviewer noted that we have an insufficient explanation of the correlation analysis on nitrogen nutrients, NO, $N_2O$, excess $N_2O$ ($\Delta N_2O$), and apparent oxygen utilization (AOU). We recognize that relying on correlation alone may not adequately illustrate the complexities of NO cycling. We will provide references to substantiate the use of the said ratios in our discussion. For example, it has been well-established that a linear relationship between $\Delta N_2O$ and AOU indicates the occurrence of $N_2O$ production from nitrification (Yoshinari, 1976; Nevison et al., 2003; Schulz et al., 2023).

To assist readers unfamiliar with the complexities of the role of NO in the nitrogen cycle, we will include a schematic diagram. By providing these, the reviewer and readers will better appreciate the various correlations we reported between the different dissolved inorganic nitrogen substrates and NO and $N_2O$. Despite these limitations, we view this study as an initial step in laying the groundwork for future research.

We have addressed the concerns highlighted by the reviewer and detailed the changes we intend to implement in the revised manuscript to address the reviewer's critiques. Reviewer comments are presented in bold italics, while our responses are in plain font.

**Specific comments:**

**Introduction**

*Lines 36-39 This is just a list of past study areas, and the authors should have devoted some space to specifying the major scientific conclusions and advances made by these studies in the marine environment NO.*

*Line 40 What is the research gap of NO? Where might the behavior of estuarine NO differ from that of the study areas described above, or what is the scientifical importance of studying estuarine NO? These should be briefly described in the Intro section.*

We appreciate the reviewer's constructive comments on the Introduction section of our manuscript. We recognize the importance of providing a clear scientific context and the specific research gaps our study addresses. Our paper indeed presents a novel case study on the measurement of dissolved NO concentration on the interface between the riverine environment and coastal seas in a well-studied estuarine system in Europe—the Elbe Estuary.

In the Introduction section, we briefly enumerated the areas where dissolved NO concentrations were already measured. It supports our argument that limited studies are done on NO in the marine environment. We will modify the paragraph from lines 34 to 39 and add a text that provides context on the importance of measuring NO concentration and estimating sea-to-air flux densities:

"The determination of dissolved NO concentration in surface water is challenging because of its reactivity, which results in a very short lifetime in (sea)water (Lancaster, 1997) ranging from 3 s to 100 s (Zafiriou and McFarland, 1981; Olasehinde et al., 2010). Nevertheless, measurements of dissolved NO in aquatic environments such as open and coastal oceans and rivers have received increasing attention during the last decade. Examples of recent NO measurement campaigns include those in the Kurose River in Japan (Anifowose et al., 2015), the Seto Inland Sea in Japan (Olasehinde et al., 2010), the tropical Northwestern Pacific Ocean (Tian et al., 2019b), the oxygen minimum zone off the coast of Peru (Lutterbeck and Bange, 2015; Lutterbeck et al., 2018), and the coastal seas off Qingdao (Tian et al., 2021, 2024). The majority of the studies performed at different time frames have indicated that both open and coastal seas are a source of atmospheric NO with fluxes ranging from 0.70 (Anifowose and Sakugawa, 2017) to as high as $45.00 \times 10^{-17}$ mol cm$^{-2}$ s$^{-1}$ (Tian et al., 2019a).

Global estimates for oceanic NO emissions and their temporal (i.e., diurnal, seasonal) and spatial variability are still lacking. To address these gaps, expanded measurements of NO distribution in the open ocean and coastal waters are essential to enhance our understanding and provide a more accurate assessment of sea-to-air flux densities.

This paper presents the first measurements of dissolved NO concentrations in the lower Elbe Estuary and Hamburg Port basins during a ship campaign in July 2021. The overarching objectives of our study were (i) to determine the distribution of dissolved NO along the salinity gradient, (ii) to estimate the flux density of NO across the water/atmosphere interface, and (iii) to identify the potential production pathways and controlling factors on NO distribution in the lower Elbe Estuary and Hamburg Port area."

**Method**

***Line 51 Define Elbe-km here.***

The definition of Elbe-km will be moved from line 62 (Figure caption) to the main text. For better coherence, we will move the definition after the sentence "Originating from the Karkonosze Mountains in the northern region of the Czech Republic, the Elbe River basin is the fourth largest catchment area (148,268 km$^2$) in Central Europe (Amann et al., 2012) with average long-term freshwater runoff of about 720 m$^3$ s$^{-1}$ (Kerner, 2007)."

***Line 79 Method uncertainty and detect limit should be presented here.***

We will include the uncertainty (i.e., average standard error of 1.28%) and refer to the previous publications for the detection limit of the methods (Schulz et al., 2023; Brase et al., 2017).

***Line 83 The text here says that triplicate NO samples are measured. But I don't see the error bars in the figures. Uncertainty of NO flux density estimate also needs to be added.***

We will revise the Figures and add the error bars in the NO concentration distribution and estimated NO flux density.

***Line 128-129 and Fig. S2. I was surprised by the range of data in the figure, which, given the error bars, can range from -5 to 15 µg/m$^3$. I'm a bit curious whether this range of error is primarily from (a) limitations of the detection method, (b) spatial heterogeneity, or (c) temporal variability. If it's from (a), the authors' averaging method may be reasonable, and if it's from (b) and (c), how large are the potential calculation errors? It looks like it might have (up to) an order of magnitude impact on the flux calculations.***

We acknowledge the Reviewer's concerns regarding the precision of our NO flux calculations. For the same reason, we have clearly stated and emphasized in the manuscript that the calculated NO flux represents a rough estimate. Ideally, measuring atmospheric NO concentrations directly onboard the research vessel would enhance accuracy, as *in situ* measurements reduce potential errors in calculating flux.

Nonetheless, due to the lack of necessary additional onboard instrumentation (i.e., NO analyzer dedicated to atmospheric measurement), we have followed a methodology similar to that used by Tian et al. (2019a), published in Biogeosciences. They also used the average atmospheric NO concentrations (2.13 ppb) in their study area for estimating flux density in the Bohai Sea. While their study just noted personal communication as the source of the average atmospheric NO concentration, we provided the source of our data (i.e., atmospheric NO measurement by the Hamburg Institute for Hygiene and Environment).

The atmospheric NO concentration was measured using the chemiluminescence method and follows the DIN (Deutsches Institut für Normung e.V.) EN 1411 standard. The DIN is the German national organization for standardization and is the German ISO member body. Calibration and quality assurance on measurement data are discussed on their website (https://luft.hamburg.de/allgemeine-informationen/kalibrierung-und-qualitaetssicherung-598742). In summary, they ensure the following:

- Use of Suitability-Tested Devices: Only devices that have passed suitability tests are employed.
- Regular Checks and Calibrations: Gas measuring devices are checked every 25 hours, and manual calibrations are performed quarterly or post-repair, using traceable standards to monitor and adjust for deviations and long-term drift.

- Traceability: Calibration standards are biennially compared with national and European reference laboratories to ensure alignment with European standards.
- Participation in Round Robin Tests: Annual nationwide and regional tests are conducted to synchronize standards and test instruments across federal states.
- Regular Maintenance: Comprehensive maintenance schedules are followed at all measuring stations in compliance with EN standards, with more extensive tests being less frequent but more intensive.
- Validation of Measurement Data: Data is manually reviewed daily, monthly, and annually to confirm its plausibility based on technical, meteorological, and empirical factors.

To improve the accuracy of our study, we selected all seven background monitoring stations located near the Hamburg Port Area. These designated monitoring stations measure background concentration levels of air pollutants and are typically far enough from emission point sources. We think that all the seven background stations near the Elbe Estuary reflect the ambient atmospheric NO concentrations over the Elbe Estuary. Moreover, to further minimize error, we specifically selected data from the period coinciding with our study. We did not include nighttime atmospheric NO measurements, typically lower due to reduced vehicular and industrial emissions at night. We used the average NO value at the seven background monitoring stations to provide a conservative estimate of the atmospheric NO concentration in the Elbe Estuary during the study period. If we look at the average values at each time point, it is near the average concentration of 4.3 $\mu$g m$^{-3}$ that we used to calculate the flux density. Notably, measurements outside the typical rush hours are close to this average concentration value. Here is the statistic of the hourly NO measurement ($\mu$g m$^{-3}$):

**Minimum: 2.00**
**Maximum: 8.25**
**Standard deviation: 1.76**
**Median: 3.86**

We should have been clearer in the Figure S2 caption that the error bars or whiskers in the scatter plot represent the standard deviation of the values measured at the "background" monitoring stations for each time point and not the minimum and maximum NO concentration values typical for box and whisker plots. We will edit the Figure caption to indicate that the error bars represent the standard deviation.

Atmospheric NO concentration may vary spatially and temporally as NO$_x$ can be emitted from vehicles and ships. You would notice that high variability at each time point is more pronounced from around 6:00 to 8:00 AM, which may be attributed to the morning rush hour.

**Section 4.1**

*The discussion in this section was a bit weak. I really like the summary of NO in Figure 6, but there wasn't much discussion of it in the main text. For example, why is it that estuaries with higher nutrient instead have lower NO concentrations than open ocean/nearshore? This study' NO is already supersaturated but still on the lower end of all the studies, what is causing the high concentrations (potentially supersaturated several times over) on the other sites?*

*Will some of the correlation patterns in this work appear in whole compile data set? How important are estuarine/oceanic NO emissions relative to terrestrial/human systems based on currently available data? Etc… This may require more work to sort out, but I believe it may expand the scientific value of this paper to be more than just like a case study.*

Thank you for your feedback. We acknowledge the need to improve the discussion and provide a more comprehensive analysis of our results. We will resolve this in our revised manuscript.

The question of why the Elbe Estuary, with relatively higher nutrients, has a lower NO concentration is still unresolved and requires further investigation. Nevertheless, this observation is an important finding of our study. This observation challenges the assumption that higher concentrations of nitrogen nutrients automatically lead to increased dissolved NO concentration. In our manuscript, subsequent to our discussion highlighting the Elbe Estuary's relatively higher nutrient levels yet lower NO concentrations than other study sites, we delve into the conflicting findings concerning the relationship between NO distribution and nitrogen-containing nutrients (refer to lines 280-291).

We also reported the hypothesis of Ayeni et al. (2021) regarding these conflicting relationships: "…Likewise, Ayeni et al. (2021) also noted that some rivers in Japan with higher $NO_2^-$ concentrations had lower rates of photoproduction of NO and vice versa, attributing these imbalances to nitrogen cycling processes (nitrification, denitrification, and anammox), which could produce or consume NO, or the photochemical transformation of organic nitrogen from dissolved organic matter producing $NO_2^-$ to form NO in areas with low $NO_2^-$."

The high reactivity of nitric oxide (NO) as a radical initiates various consumption mechanisms which may influence its concentration in the Elbe Estuary. Zafiriou et al. (1979) reported that there is no evidence of interaction between NO and metals under marine conditions, though NO is known to react with metals yielding nitrosyl (M-NO) or iso-nitrosyl (M-ON) metal complexes (Ford and Lorkovic, 2002; Richter-Addo et al., 2002). We are not certain whether reaction with transition metals is a sink in the marine environment, particularly in coastal and estuarine environment as this has not been explored. Additionally, organisms (algae, phytoplanktons) can both consume and produce NO. A recent paper by (Bange et al., 2024) noted that the consumption processes for NO in the sea(water) are still unresolved.

While current literature suggests that coastal areas could potentially act as significant sources of emission to the atmosphere, this may vary temporally and spatially across the studied sites. Up to now, the majority of the literature reports positive sea-to-air flux, indicating emissions as a major sink; however, regional exceptions, such as one measurement in the Shandong Peninsula (Gong et al., 2023), indicate that generalizations should be made cautiously.

Regarding the Reviewers' comment on the importance of estuarine/oceanic NO emissions relative to terrestrial/anthropogenic emissions, we will briefly discuss these in our revised manuscript. We will add a few examples from highly cited papers on NO emissions from the terrestrial environment, including but not limited to Bouwman et al. (2002a), Bouwman et al. (2002b), Stehfest and Bouwman (2006), Williams et al. (1992). As the reviewers pointed out, this would enhance the discussion on the importance of NO emission from coastal areas as a source to the atmosphere.

**Section 4.2**

***Because salinity is also an indicator of mixing, the negative correlation with salinity noted here is likely to represent "mixing" for NO (i.e., mixing affects both NO and salinity), not "salinity and freshwater input influencing NO concentrations" (i.e., salinity/freshwater itself influences NO).***

We appreciate the reviewer's comment regarding the role of mixing in the observed negative correlation between salinity and nitric oxide (NO) concentrations. We recognize that mixing indeed plays an important role in the distribution of biogeochemical parameters in the Elbe Estuary. Indeed, Dähnke et al. (2008) noted that conservative mixing behavior could be observed in the Elbe Estuary irrespective of the season.

However, our intention in this section is to emphasize the significance of riverine/freshwater inputs as a primary source of higher NO concentrations. We supported our argument with two studies: one documenting relatively higher surface dissolved NO in the southern Bohai Sea due to the Yellow River's outflow ascribing it to high DIN input (Gong et al., 2023), and another study (Ayeni et al., 2021) noting a NO concentration gradient in the Kurose River, with downstream sections influenced by anthropogenic activities.

**Section 4.4**

***The source/sink of NO is so complex that I would suggest that the authors include a suitable concept fig in an attachment or in the main text to allow more readers to easy follow the processes you describe.***

We understand the importance of ensuring clarity for readers and a broader audience. We will provide a schematic diagram or illustration providing known sources and sink processes of NO, particularly as an intermediate in the nitrogen cycle. We will revise our manuscript so readers can easily follow the complex nitrogen cycle processes discussed and other known sources and sinks.

**Section 4.4.1**

***Lines 322-323 Why this statement make sense? Nitrification only contribute minor part of AOU. Some explanations or references are needed.***

We thank the reviewer for this comment and the opportunity to clarify this statement. Indeed, nitrification only contributes a minor part to the AOU. We understand the previous text could be enhanced for clarity, and as such, we will edit the text to ensure that readers understand the text better (edit this to be more precise).

However, it is established that a significant linear correlation between excess $N_2O$ ($\Delta N_2O$) and AOU indicates the occurrence of nitrification. We will revise the text and provide references to support this argument.

Previous: In this study, we noted that nitrification occurs in the entire stretch of the study site based on the plots of AOU and excess $N_2O$ (Fig. 7a), as AOU correlates significantly with the $\Delta N_2O$ in all three salinity zones.

Edited: Based on the plot of AOU and excess $N_2O$ ($\Delta N_2O$), $N_2O$ production can be observed in the entire stretch of the study site (Fig. 7a). We also observed that AOU has a significant positive linear relationship with $\Delta N_2O$ in the entire stretch of the Elbe Estuary— in the brackish coastal zone (R=0.95, p < 0.001), limnic zone (R=0.65, p < 0.05), and in the Hamburg Port area (R=0.75, p < 0.01). Previous reports (Schulz et al., 2023; Brase et al., 2017; Nevison et al., 2003; Walter et al., 2006) indicated that a significant positive linear relationship between AOU and $\Delta N_2O$ usually indicates $N_2O$ production from nitrification.

***Lines 324-325 I can't follow these sentences. Many ratios (e.g., $N_2O/NH_4^+$, $NO_2^-/O_2$ ...) appear in the correlation diagram. What do these ratios represent? Some background should be provided.***

We understand the need to enhance clarity for readers. We will revise the text for readers to understand these ratios. We will provide a diagram of the nitrogen cycle for readers to understand how these ratios might be related to the different nitrogen cycling processes. For instance, by providing a diagram similar to that of Figure 1 in Caranto and Lancaster (2017) showing the nitrification process, it would be easier to pinpoint that $N_2O$ is a product, $NH_4^+$ is a reactant, and $NO_2^-$ and $NO_3^-$ can be oxidized from NO.

***Lines 326-327 How "a significant positive linear relationship exists between N₂O and NO₃⁻" is linked to "These findings point to NO production via nitrification"? I can't find the logic connection.***

We will revise the text to clarify the link between $N_2O$ and $NO_3^-$ in nitrification. To establish a logical connection between these statements, it's important to understand the following:

- Nitrous oxide ($N_2O$) is a known byproduct of nitrification and an intermediate of the denitrification processes.
- Nitrification is a microbial process where ammonia ($NH_3$) is oxidized to nitrate ($NO_3^-$), and it can also lead to the production of nitrite ($NO_2^-$).
- During nitrification, obligatory intermediates (Caranto and Lancaster, 2017), nitric oxide (NO) and hydroxylamine ($NH_2OH$) can be produced. NO can further yield $N_2O$.

The significant positive linear relationship between $N_2O$ and $NO_3^-$, may suggest that as the concentration of nitrate increases, so does the concentration of $N_2O$. If this relationship is found to be significant within the context of the study, it is possible that the processes leading to the production of $NO_3^-$ (like nitrification) are also associated with the production of $N_2O$ (see Schulz et al., 2023). Hence, if $N_2O$ levels are rising with $NO_3^-$ levels, it could be indicative of active nitrification, during which NO is produced as an intermediate. The logic is that if $N_2O$ is increasing with $NO_3^-$ and we know that $N_2O$ can be a byproduct of nitrification (which also produces $NO_3^-$) then an increase in both could point to nitrification as the source process, and thus, the production of NO as part of that process.

***Line 331 What "observed trends" refer to?***

We will revise the manuscript to enhance clarity for the reader.

***Line 334 Authors discuss here that nitrification is the SINK of NO. I am a little confused because the whole section discusses about nitrification as SOURCE of NO.***

Note that while NO can be produced in the nitrification process as an obligatory intermediate (Caranto and Lancaster, 2017), it can also be consumed in further oxidation steps. Shown below is **Figure 1** from Caranto et al. (2017):

[Figure]

[Figure]

**Figure 1.** Schematic diagram comparing the prevailing view on the nitrification process and the model proposed by Caranto and Lancaster (2017) that shows nitric oxide is an additional obligate intermediate in the nitrification process (From "Nitric oxide is an obligate bacterial nitrification intermediate produced by hydroxylamine oxidoreductase," by J.D. Caranto and K.M. Lancaster, 2017).

In the nitrification process, ammonia ($NH_4^+$) undergoes oxidation to form hydroxylamine ($NH_2OH$), which can further yield NO and then yield $N_2O$, $NO_2^-$, or $NO_3^-$. Another Reviewer agreed with the idea of the $NH_4^+$ limitation in the coastal/brackish and limnic zones leading to the observed significant inverse relationship between NO and $NO_2^-$ and NO vs $NO_2^-/O_2$ ratio. If $NH_4^+$ is not limited or has a continuous supply in the reaction, one would see a direct relationship between NO and $NO_2^-$. When $NH_4^+$ is limited, NO will be consumed in the process, decreasing its concentration while increasing the product $NO_2^-$, $NO_3^-$, or $N_2O$.

***Section 4.4.2***

***This entire section suffers from a problem like that of section 4.4.1, in that a large amount of the text simply suggests the correlation without explaining it, making the logical chain of support for the author's argument incomplete. For example, almost all of the text in lines 350-365.***

We understand the need to enhance the manuscript by providing a thorough explanation of the nitrogen cycle processes. We will revise the manuscript accordingly for clarity.

***Other notes:***

***Table S2: Why NO flux density (mol $m^{-2}$ $s^{-1}$) have a different unit with $N_2O$ flux density (μmol $L^{-1}$ $d^{-1}$)? It also differs from unit in the main text and figure 5 and 6.***

Thank you for the attention to detail. We apologize for the oversight. We have corrected the unit of NO flux density to mol $cm^{-2}$ $s^{-1}$ in Table S2, and the unit of $N_2O$ flux density to μmol $m^{-2}$ $d^{-1}$. These are standard flux density units established in prior publications. For easier comparability and consistency with previous publications, we have used the units μmol $m^{-2}$ $d^{-1}$ for $N_2O$ flux density and mol $cm^{-2}$ $s^{-1}$ for NO flux density.

***Why don't you add NO to the correlation plots of the main text and attachments? I don't see NO in Figure 7 and Figures S4-S6? And if space permits, I suggest you place Fig. S4 (after adding NO) and Fig. S7 into main text.***

We will follow the suggestion of the Reviewer to add correlation plots of NO to the main text. We removed NO in Figure 7 and Figures S4-S6 because we have made separate correlation plots of NO vs other parameters. However, we can add this to the Figures if it would enhance clarity.

**References:**

Amann, T., Weiss, A., and Hartmann, J.: Carbon dynamics in the freshwater part of the Elbe estuary, Germany: Implications of improving water quality, Estuar Coast Shelf Sci, 107, 112–121, https://doi.org/10.1016/j.ecss.2012.05.012, 2012.

Anifowose, A. and Sakugawa, H.: Determination of daytime flux of nitric oxide radical (NO•) at an inland sea–atmospheric boundary in Japan, Journal of Aquatic Pollution and Toxicology, 1, 10, 2017.

Anifowose, A. J., Takeda, K., and Sakugawa, H.: Photoformation rate, steady-state concentration and lifetime of nitric oxide radical (NO) in a eutrophic river in Higashi-Hiroshima, Japan, Chemosphere, 119, 302–309, https://doi.org/10.1016/j.chemosphere.2014.06.063, 2015.

Ayeni, T. T., Jadoon, W. A., Adesina, A. O., Sunday, M. O., Anifowose, A. J., Takeda, K., and Sakugawa, H.: Measurements, sources and sinks of photoformed reactive oxygen species in Japanese rivers, Geochem J, 55, 89–102, https://doi.org/10.2343/geochemj.2.0620, 2021.

Bange, H. W., Mongwe, P., Shutler, J. D., Arévalo-Martínez, D. L., Bianchi, D., Lauvset, S. K., Liu, C., Löscher, C. R., Martins, H., Rosentreter, J. A., Schmale, O., Steinhoff, T., Upstill-Goddard, R. C., Wanninkhof, R., Wilson, S. T., and Xie, H.: Advances in understanding of air–sea exchange and cycling of greenhouse gases in the upper ocean, Elem Sci Anth, 12, https://doi.org/10.1525/elementa.2023.00044, 2024.

Bouwman, A., Boumans, L., and Batjes, N.: Modeling global annual N2O and NO emissions from fertilized fields, Global Biogeochem Cy, 16, 28-1-28–9, https://doi.org/10.1029/2001gb001812, 2002a.

Bouwman, A. F., Boumans, L. J. M., and Batjes, N. H.: Emissions of N2O and NO from fertilized fields: Summary of available measurement data, Glob. Biogeochem. Cycles, 16, 6-1-6–13, https://doi.org/10.1029/2001gb001811, 2002b.

Brase, L., Bange, H. W., Lendt, R., Sanders, T., and Dähnke, K.: High Resolution Measurements of Nitrous Oxide (N2O) in the Elbe Estuary, Frontiers Mar Sci, 4, 162, https://doi.org/10.3389/fmars.2017.00162, 2017.

Caranto, J. D. and Lancaster, K. M.: Nitric oxide is an obligate bacterial nitrification intermediate produced by hydroxylamine oxidoreductase, Proc National Acad Sci, 114, 8217–8222, https://doi.org/10.1073/pnas.1704504114, 2017.

Dähnke, K., Bahlmann, E., and Emeis, K.: A nitrate sink in estuaries? An assessment by means of stable nitrate isotopes in the Elbe estuary, Limnol. Oceanogr., 53, 1504–1511, https://doi.org/10.4319/lo.2008.53.4.1504, 2008.

Ford, P. C. and Lorkovic, I. M.: Mechanistic Aspects of the Reactions of Nitric Oxide with Transition-Metal Complexes, Chem Rev, 102, 993–1018, https://doi.org/10.1021/cr0000271, 2002.

Gong, J.-C., Jin, H., Li, B.-H., Tian, Y., Liu, C.-Y., Li, P.-F., Liu, Q., Ingeniero, R. C. O., and Yang, G.-P.: Emissions of Nitric Oxide from Photochemical and Microbial Processes in Coastal Waters of the Yellow and East China Seas, Environmental Science & Technology, 57, 4039–4049, https://doi.org/10.1021/acs.est.2c08978, 2023.

Kerner, M.: Effects of deepening the Elbe Estuary on sediment regime and water quality, Estuar Coast Shelf Sci, 75, 492–500, https://doi.org/10.1016/j.ecss.2007.05.033, 2007.

Lancaster, J. R.: A Tutorial on the Diffusibility and Reactivity of Free Nitric Oxide, Nato Sci S A Lif Sci, 1, 18–30, https://doi.org/10.1006/niox.1996.0112, 1997.

Lutterbeck, H. E. and Bange, H. W.: An improved method for the determination of dissolved nitric oxide (NO) in seawater samples, Ocean Sci, 11, 937–946, https://doi.org/10.5194/os-11-937-2015, 2015.

Lutterbeck, H. E., Arévalo-Martínez, D. L., Löscher, C. R., and Bange, H. W.: Nitric oxide (NO) in the oxygen minimum zone off Peru, Deep Sea Res Part Ii Top Stud Oceanogr, 156, 148–154, https://doi.org/10.1016/j.dsr2.2017.12.023, 2018.

Nevison, C., Butler, J. H., and Elkins, J. W.: Global distribution of N2O and the ΔN2O-AOU yield in the subsurface ocean, Glob. Biogeochem. Cycles, 17, n/a-n/a, https://doi.org/10.1029/2003gb002068, 2003.

Olasehinde, E. F., Takeda, K., and Sakugawa, H.: Photochemical Production and Consumption Mechanisms of Nitric Oxide in Seawater, Environ Sci Technol, 44, 8403–8408, https://doi.org/10.1021/es101426x, 2010.

Richter-Addo, G. B., Legzdins, P., and Burstyn, J.: Introduction: nitric oxide chemistry., Chem. Rev., 102, 857–60, https://doi.org/10.1021/cr010188k, 2002.

Schulz, G., Sanders, T., Voynova, Y. G., Bange, H. W., and Dähnke, K.: Seasonal variability of nitrous oxide concentrations and emissions in a temperate estuary, Biogeosciences, 20, 3229–3247, https://doi.org/10.5194/bg-20-3229-2023, 2023.

Tian, Y., Xue, C., Liu, C.-Y., Yang, G.-P., Li, P.-F., Feng, W.-H., and Bange, H. W.: Nitric oxide (NO) in the Bohai Sea and the Yellow Sea, Biogeosciences, 16, 4485–4496, https://doi.org/10.5194/bg-16-4485-2019, 2019a.

Tian, Y., Yang, G.-P., Liu, C.-Y., Li, P.-F., Chen, H.-T., and Bange, H. W.: Photoproduction of nitric oxide in seawater, Ocean Sci, 16, 135–148, https://doi.org/10.5194/os-16-135-2020, 2019b.

Tian, Y., Wang, K.-K., Yang, G.-P., Li, P.-F., Liu, C.-Y., Ingeniero, R. C. O., and Bange, H. W.: Continuous Chemiluminescence Measurements of Dissolved Nitric Oxide (NO) and Nitrogen Dioxide (NO 2 ) in the Ocean Surface Layer of the East China Sea, Environ Sci Technol, https://doi.org/10.1021/acs.est.0c06799, 2021.

Tian, Y., Jian, H.-M., Liu, C.-Y., Gong, J.-C., Li, P.-F., and Yang, G.-P.: Distribution and influencing factors of atmospheric nitrogen oxides (NOx) over the east coast of China in spring: Indication of the sea as a sink of the atmospheric NOx, Mar. Pollut. Bull., 200, 116095, https://doi.org/10.1016/j.marpolbul.2024.116095, 2024.

Walter, S., Bange, H. W., Breitenbach, U., and Wallace, D. W. R.: Nitrous oxide in the North Atlantic Ocean, Biogeosciences, 3, 607–619, https://doi.org/10.5194/bg-3-607-2006, 2006.

Williams, E. J., Hutchinson, G. L., and Fehsenfeld, F. C.: NOx And N2O Emissions From Soil, Glob. Biogeochem. Cycles, 6, 351–388, https://doi.org/10.1029/92gb02124, 1992.

Yoshinari, T.: Nitrous oxide in the sea, Mar. Chem., 4, 189–202, https://doi.org/10.1016/0304-4203(76)90007-4, 1976.

Zafiriou, O. C. and McFarland, M.: Nitric oxide from nitrite photolysis in the central equatorial Pacific, J Geophys Res, 86, 3173, https://doi.org/10.1029/jc086ic04p03173, 1981.

Zafiriou, O. C. and True, M. B.: Nitrite photolysis in seawater by sunlight, Mar Chem, 8, 9–32, https://doi.org/10.1016/0304-4203(79)90029-x, 1979.

---

## Author Comment (AC2)

**Response to Reviewer 3**

*Summary*

*The manuscript titled "Dissolved Nitric Oxide in the Lower Elbe Estuary" by Ingeniero et al. quantified the fluxes of nitric oxide (NO) in relation to other nitrogen cycle parameters in the Elbe River Estuary and Hamburg Port Area. Using a clever chemiluminescent detection method and flow-through sampling system, the authors measured dissolved NO concentrations in surface waters alongside temperature, salinity, pH, and dissolved oxygen ($O_2$). The authors made concurrent measurements of nitrate, nitrite, and ammonium with an autoanalyzer and nitrous oxide ($N_2O$) with laser spectroscopy. The authors found that NO was supersaturated in the surface layer of both study areas, so they were both a source of NO to the atmosphere. Based on the concurrent [$O_2$] and dissolved inorganic nitrogen measurements, the authors conclude that this NO is likely produced via biological processes (nitrification, denitrification, and nitrifier-denitrification), as opposed to the photolysis of nitrite.*

*General Appraisal*

*In this paper, the authors present the first-ever measurements of NO in the Elbe River system. NO measurements in the literature are scarce because its short lifetime makes analysis difficult, so this paper represents a substantial contribution to our understanding of NO in the marine environment. Furthermore, the authors measure significant NO supersaturation and fluxes in the surface waters of much of the Elbe River, which is important because NO is a contributor to smog, acid rain, and ozone.*

*The major strengths of this paper are the presentation of novel, high-resolution NO measurements and the clear relationships that emerge between NO and other inorganic nitrogen species, [$O_2$], pH, and chlorophyll. The authors present a clean, concise interpretation of these results and the paper is generally straightforward and easy to read.*

*The major weakness of this paper is that the discussion of temporal variability (day/night and seasonal variations) is not linked to the clear boom-and-bust cycle seen in the Hamburg Port area. The authors have locations with peaks of chlorophyll and [$O_2$], and other locations with oxygen and pH minima and $N_2O$ and NO maxima. This implies to me that there are some locations where you captured net production and others where they captured net respiration, which draws down [$O_2$] and creates an ideal environment for $N_2O$ and NO production in sediments or particles. The authors allude to this in the conclusions, but how would day-night temporal variation at each site affects the data? Would blooms in some locations propagate downstream and create pockets of high respiration further downstream? The authors have a paragraph in the conclusions about potential temporal effects, and my suggestion would be to move this paragraph into the discussion and link it more clearly to their results.*

*The paper is generally well-written. There are only a few grammatical errors and clumsy sentences that I note in the technical corrections.*

*My primary concern is about the conclusion (and I believe this is only stated in the abstract) that nitrifier-denitrification is the primary source of NO in the Hamburg Port area. While I agree that the lack of correlation between nitrate and apparent oxygen utilization (AOU) in the Hamburg Port area may indicate the presence of denitrification or nitrifier-denitrification, I don't think you can rule out one or the other. In other words, all you can conclude from this data is that there is a process other than nitrite oxidation that is consuming nitrite. Likewise, if you invoke denitrification and/or nitrifier-*

*denitrification in sediments or particles, I don't think you can rule out the presence of anammox. In fact, instead of ruling out anammox based on water column [O2], you should list it as a potential alternative source of NO. The strong correlations between NO, nitrite, and ammonium may indeed be a sign of anammox as a source of NO in the Hamburg Port area. Also, while denitrification and/or nitrifier-denitrification may be present in this zone, the water column [O2] suggests that the primary source of NO would still be nitrification, and this is supported by the strong correlations in this zone between NO, nitrite, and ammonium.*

We appreciate your recognition of the novel contributions our work makes to the field – the first-ever measurements of NO in the Elbe River system and the identification of significant NO supersaturation and fluxes in surface waters. Your acknowledgment of the clarity and readability of the paper is encouraging.

Your critique concerning the discussion of temporal variability and its connection to the observed biogeochemical cycles within the Hamburg Port area is well-founded. We will incorporate your suggestion in our revised manuscript.

Regarding the primary sources of NO, we acknowledge the Reviewer's concerns about the conclusiveness of nitrifier-denitrification as the dominant process in the Hamburg Port area. We agree that the current data does not definitively rule out other processes, such as anammox, and that it is prudent to consider such alternative NO consumption processes. We will revise the manuscript to reflect a more balanced view of potential NO sinks or sources, acknowledging the strong correlations observed and how they may implicate various nitrogen-transforming processes, including anammox.

Thank you for the helpful and very detailed comments. The detailed suggestions will be thoroughly implemented to enhance the manuscript's technical quality. We have addressed the concerns highlighted by the reviewer and detailed the changes we intend to implement in the revised manuscript to address the reviewer's critiques. Reviewer comments are presented in bold italics, while our responses are in plain font. We look forward to submitting a comprehensive revised manuscript that addresses the points you've raised.

**Specific comments**

*Line 14: Is the same chemiluminescent optode spot system often used for $O_2$ (Frey et al., 2023)?*

No. The Luminescence Measuring Oxygen Sensors used by Frey et al. are different from our detection method: We used a chemiluminescent method for NOx which is typically used for atmospheric monitoring of NOx. Lutterbeck and Bange (2015) describe the method in detail. In our earlier drafts of the manuscript, we cited the method paper by Lutterbeck and Bange (2015) in the Abstract for clarity. However, adhering to standard writing practices, we omitted this citation from the Abstract in the final draft when we submitted the paper to Biogeosciences. This paper, if published, would be the first application of the method in a coastal and estuarine environment. We will edit the text as follows:

"The discrete surface water samples were analyzed using a chemiluminescence NO analyzer connected to a stripping unit."

**Line 15: Why not write pM instead of 10^-12 mol/L? You do so later in the manuscript.**

Thank you for your comment. For consistency, we will follow your suggestion to use pM instead of $10^{-12}$ mol L$^{-1}$.

**Line 20: Based on your discussion, this could be nitrifier-denitrification or denitrification. I don't think you can rule out one or the other based on your data.**

We agree with this comment. While we cannot rule out which exact nitrogen cycling processes could be present, we think that nitrifier-denitrification or denitrification influences the NO distribution in the Hamburg Port Area. We will be including denitrification in the text.

**Line 34: What is the lifetime of NO in seawater/water?**

The lifetime of nitric oxide (NO) in seawater or water is relatively short due to its high reactivity. In aquatic environments, NO can rapidly react with oxygen, metals, and organic compounds. The exact lifetime can vary depending on several factors, including temperature, pH, and the presence of reactants, but it is typically on the order of a few seconds to a few minutes (i.e. 3 – 100 s) (Zafiriou and McFarland, 1981; Olasehinde et al., 2010). We will provide these values in the revised manuscript.

**Line 72: The way this equation is written is confusing. Are you multiplying the corrected $O_2$ by 1.12? Or the uncorrected? What are the units of the intercept? Also, does the intercept of 13.41 mean that the detection limit of the oxygen optodes is 13.41 (units?)?**

We will edit the $O_2$ correction equation. The revised equation will be stated as: $[1.12 \times O_{2(optode\ measurement)}]$ + 13.41 ($R^2 = 0.97$). The unit is μM.

**Line 83: Give us some numbers for what this lifetime is**

Please see our response above (-> Line 34). We will add these values in the revised manuscript.

**Line 84: So the calibrator is just an NO source, right?**

Yes, this is right. It is a portable calibration source that operates using a compact nitrous oxide ($N_2O$) cartridge, producing gas output that is traceable to the US National Institute of Standards and Technology (NIST) standards, as detailed in the study by Birks et al. (2020).

**Line 90: Why do you need the calibrator in addition to the aqueous NO standard solutions?**

The calibrator is used to adjust the NO analyzer, ensuring its responses are accurate and reliable. This step is fundamental because it directly affects the instrument's precision and accuracy, ensuring that its readings are consistent with "true" NO concentrations. Calibration with the calibrator involves adjusting the instrument's response to known concentrations of NO gas. This process ensures that the instrument's detection and measurement systems are properly aligned with the actual concentrations, correcting for any drift, sensor degradation, or other factors that might affect accuracy over time. Meanwhile, the aqueous NO standard solution is used for method calibration.

***Line 94: This calculation is to convert the mole fraction you measure in the headspace to the dissolved NO concentration, right? Is there a reason to assume that the headspace is at a pressure of 1 atm? I would assume it would be slightly over pressurized... how would that affect your measurements?***

We use the stripping method detailed in Lutterbeck and Bange (2015). Furthermore, the NO analyzer operates with atmospheric pressure input and will display an error if it exceeds a certain pressure threshold. A needle valve was also installed to reduce pressure variations.

*Line 97: Here you use pM. I would stick to this throughout the text.*

Thank you for pointing out the inconsistency. We have revised the manuscript to ensure that 'pM' is consistently used throughout the manuscript.

*Lines 102-103: In eqn. (2) you assume the barometric/atmospheric pressure is 1 atm. Is this a reasonable assumption at this time of year, in this part of the world?*

The average air pressure in Hamburg during this time is at 1009 hPa, or when converted to atmosphere, is 0.9958 atm which is close to 1 atm.

See https://meteostat.net/en/place/de/hamburg?s=10147&t=2021-07-27/2021-07-29 (last accessed 1 March 2024), which uses weather data from NOAA.

*Line 125: Same comment as above with setting atmospheric pressure to 1 atm.*

Please see our response above for Lines 102-103.

*Lines 129-130: How was this mean value calculated? Mean of all hourly measurements at all monitoring stations over the study period? Given the short lifetime of NO, doesn't it make sense to calculate a mean $c_{EQ}$ on a day-by-day or even shorter basis - or do all of the stations look like figure S2, where the hourly concentrations are all within error of the average?*

You are correct. This is the mean of the average hourly measurement at all monitoring stations over the study period. We excluded nighttime values as NO concentrations are rather low in the evening due to low emissions from vehicles. We think this is a conservative estimate of the NO concentration in the Elbe Estuary.

*Lines 172-174: Is the variability of [$O_2$] because of changes in productivity?*

The variability in [$O_2$] levels can indeed be a result of changes in productivity. Note that the measurements were taken during the daytime when net productivity should be higher. During photosynthesis, phytoplankton consume $CO_2$ and release $O_2$, which increases the [$O_2$] in the water. The higher the phytoplankton productivity, the more $O_2$ is produced. Additionally, photosynthesis affects pH levels. As phytoplankton consume $CO_2$, they can reduce the amount of $CO_2$ in the water, which can cause the water to become less acidic (increase in pH level).

*Lines 184-185: Report a number for the maximum concentration to give a sense of scale - 200 µM is a lot!*

To improve specificity, we will edit the sentence and provide the value of the concentration.

*Lines 200-201: It looks like the peaks in $N_2O$ correspond to the minima in [$O_2$] - if that's the case, worth pointing out here.*

Yes, this is correct. We will edit the sentence to emphasize $N_2O$ production in minimum dissolved $O_2$ concentration (See also Fig. S6 f).

***Line 225: You should also mention that the peaks in NO in the Hamburg Port area correspond to the peaks in $N_2O$, $NO_2^-$, and $NH_4^+$!***

This is correct for two peaks in the Hamburg Port Area but not in the maximum NO concentration measured.

***Line 232: I would recommend converting these flux values to fM: 0.31-55 fmol $cm^{-2}$ $s^{-1}$.***

We agree that it would have been better to use the shorter name. However, for the sake of inter-comparability with previous research, we decided to use scientific notation in reporting the flux values.

***Line 238: How do your measurements compare to previous measurements in terms of saturation? If 147-274% saturated is at the low end of marine NO measurements, I'm curious what these higher concentrations correspond to. This would imply that the ocean could be a major source of NO to the atmosphere!***

We will update the discussion to include saturation values of previous studies (if these are available).

***Lines 251-269: I would avoid interpreting a relationship that is not statistically significant. This section is mostly literature review anyways.***

While the relationship in our findings is not statistically significant in linear correlation analysis, the general trend remains that at lower salinity values (with higher DIN) NO values are also relatively higher. In this section, we want to emphasize the importance of DIN input from freshwater, particularly ammonium and nitrite on the NO distribution.

***Lines 276-278: This is a really important finding: you have much higher $NO_3^-$, $NO_2^-$, and $NH_4^+$ than previous studies in other rivere and coastal areas, but not higher NO. What is unique to the Elbe river compared to the other rivers cited here?***

The question of why the Elbe Estuary, with relatively higher nutrients, has a lower NO concentration is still unresolved and requires further investigation. Nevertheless, we think that this observation is an important finding of our study. This observation challenges the assumption that higher concentrations of nitrogen nutrients automatically lead to increased dissolved NO concentration. In our manuscript, subsequent to our discussion that highlighted the Elbe Estuary's relatively higher nutrient levels yet lower NO concentrations compared to other study sites, we delve into the conflicting findings concerning the relationship between NO distribution and nitrogen-containing nutrients (refer to lines 280-291).

We also reported the hypothesis of Ayeni et al. (2021) regarding these conflicting relationships: "…Likewise, Ayeni et al. (2021) also noted that some rivers in Japan with higher NO2− concentrations had lower rates of photoproduction of NO and vice versa, attributing these imbalances to nitrogen cycling processes (nitrification, denitrification, and anammox), which could produce or consume NO, or the photochemical transformation of organic nitrogen from dissolved organic matter producing $NO_2^-$ to form NO in areas with low NO2−."

The high reactivity of nitric oxide (NO) as a radical initiates various consumption mechanisms which may influence its concentration in the Elbe Estuary. Zafiriou et al. (1979) reported that there is no evidence of interaction between NO and metals under marine conditions, though NO is known to react with metals yielding nitrosyl (M-NO) or iso-nitrosyl (M-ON) metal complexes (Ford and Lorkovic, 2002; Richter-Addo et al., 2002). Additionally, biological productivity can both consume and, produce NO, further

contributing to its dynamic cycle in the environment. A recent paper by (Bange et al., 2024) noted that the consumption processes for NO in the sea(water) are still unresolved.

*Lines 291-292: What about the $DN_2O/NO_3^-$ ratio?*

Thank you for your helpful comment. There is a strong correlation between NO and $DN_2O/NO_3^-$ ratio in the Hamburg Port area ($R^2$=0.95, $p<0.001$) and limnic zone ($R^2$=0.72, $p<0.001$). We will provide the $DN_2O/NO_3^-$ ratio plot in our revised Figure.

*Lines 299-302: So the overall trend (which is positive) is driven by the Hamburg Port area, and the overall trend masks the negative relationships in the limnic and coastal-brackish zones. This is a good example of an ecological fallacy.*

Indeed, this is accurate, which underlines the rationale for incorporating this information into the discussion. It is often essential to focus on finer details as opposed to the broader context, as this approach can reveal features influenced by biogeochemical processes within particular ecological zones that might otherwise be overlooked.

*Lines 307-308: Add citation: Burlacot et al. (2020).*

We will add this important paper discussing algal photosynthesis utilizing NO and producing $N_2O$ in our citation.

*Lines 308-310: It's worth pointing out that the Chl. peaks occurred right before the NO peaks.*

We will revise the manuscript to note this relevant observation.

**Lines 313-315: What about the anaerobic process (anammox, denitrification) in the river sediments?**

We understand that anammox and denitrification processes could occur in the river sediments.We will briefly provide references regarding these anaerobic processes in the sediments (Schroeder et al., 1991; Deek et al., 2013). There are also previous studies that measured NO in sediments (Sørensen, 1978; Schreiber et al., 2014).

However, we are not certain whether NO released from sedimentary processes significantly impacts NO in the sampled water in the Hamburg Port Area. The overall water depth in the Hamburg Port Area was >15 m, so NO released from the sediments is unlikely to make it to the surface layer where we took the samples (because it has a short lifetime in seawater).

*Line 323: You could also look at the relationship of $DN_2O/NO_3^-$ vs. [O2] or $DN_2O/AOU$ vs. O2 (Nevison et al., 2003).*

Thank you for your helpful comment. In a revised manuscript we will look at this relationship to support our discussion on the nitrification process in the Elbe Estuary. Shown is the result of our regression analysis. We noted a significant positive correlation between $DN_2O/NO_3^-$ vs. [O2] in the limnic zone but a negative correlation (not significant) in the Hamburg Port Area. Meanwhile, the $DN_2O/AOU$ ratio vs [O2] is negatively correlated in the overall plot ($R^2$=-0.4031, $p$=0.02) but there is no significant linear

relationship in the coastal brackish zone and limnic zone. There is a significant negative correlation between $DN_2O/AOU$ vs $[O_2]$ ratio in the Hamburg Port Area ($R^2$=-0.61928, p=0.03).

```
{'Overall'              }    {'dN2ONitrateRatio'}     0.23955      0.19431
{'Coastal-Brackish Zone'}    {'dN2ONitrateRatio'}     0.23418      0.65515
{'Limnic Zone'          }    {'dN2ONitrateRatio'}     0.66273      0.013565
{'Hamburg Port Area'    }    {'dN2ONitrateRatio'}    -0.47012      0.12301
{'Overall'              }    {'dN2OAOURatio'    }    -0.40311      0.018083
{'Coastal-Brackish Zone'}    {'dN2OAOURatio'    }    -0.27444      0.44287
{'Limnic Zone'          }    {'dN2OAOURatio'    }    -0.16077      0.61768
{'Hamburg Port Area'    }    {'dN2OAOURatio'    }    -0.61928      0.031764
```

**Line 325: In this context, I would actually call $NO_2^-$ a product of nitrification, not a precursor, because $NH_4^+$ oxidation to $NO_2^-$ produces $N_2O$ and NO as a byproduct; $NO_2^-$ oxidation does not.**

You are correct, $NO_2^-$ is a product of nitrification. We will amend the text to accurately reflect nitrite as a product in the nitrification process. Thank you for bringing this to our attention, ensuring the precision of the scientific content of our manuscript.

**Line 329: The limnic zone correlations in Figure S7 look like they're being driven by two points at either extreme of NO, while the rest of the points cluster in the middle. I would avoid over-interpreting these plots.**

Thank you for your pointing this out. We agree that we have to avoid overinterpretation of the result. However, it is also important to note the significant correlation that exists at p < 0.001.

**Lines 351-352: Elaborate here upon why the lack of a significant relationship between $NO_3^-$ and AOU indicates the presence of denitrification or nitrifier-denitrification.**

We will revise the text to note that there could be other processes aside from nitrification (not just denitrification/nitrifier denitrification) that affected the $NO_3^-$ and AOU relationship in the Hamburg Port Area. This may also include high respiration/remineralization rates and mixing with water from the port basins, which might impact the correlation.

**Line 363-365: If you imply that denitrification could be occurring in the sediments even though the water column oxygen concentrations are too high, I don't think you can rule out anammox based on water column oxygen concentrations.**

We agree to this, and we will not exclude the possibility of anammox process without other evidence to rule it out. Genetic analysis of nitrogen cycle marker genes could have been helpful in our data analysis.

**Line 367: ...or anoxic microsites within particles.**

Thank you for your helpful suggestion. We will add this phrase to the revised manuscript and cite relevant publications (Liu et al., 2013; Xia et al., 2017).

**Lines 370-371: I'm really interested in this apparent boom and bust cycle in your data. You have locations with peaks of chlorophyll and oxygen, and other locations with oxygen and pH minima and $N_2O$ maxima. This implies to me that there are some locations where you captured net production and others where you captured net respiration, which draws down $O_2$ and creates an ideal environment for**

*N$_2$O and NO production in sediments or particles. You allude to this in the conclusions, but how do you think day-night temporal variation in each of your sites affects your data? Would blooms in some locations propagate downstream and create pockets of high respiration further downstream?*

Thank you for highlighting these aspects of our dataset. The observed fluctuations in chlorophyll and oxygen concentrations, along with the corresponding variations in pH and N$_2$O levels across different locations, indeed suggest episodic events of net production and respiration. These biological processes are fundamental in shaping the biogeochemical profile of the estuary, with photosynthesis contributing to peaks in oxygen and chlorophyll during daylight hours, and respiration leading to oxygen depletion and potentially creating favorable conditions for N$_2$O and NO production during nighttime or in less oxygenated microenvironments such as suspended sediments or particulate matter (Schulz et al., 2022).

In line with your comment, we recognize the necessity for a more comprehensive analysis that accounts for temporal variations, including diurnal shifts, in the study of nitric oxide dynamics in estuaries. Additional research, potentially involving continuous monitoring at various sites, would be invaluable in deciphering these complex interactions and understanding how they might influence the distribution and nitrogen cycling in the estuary. We will discuss temporal effects and elaborate on the need for further research on this in our revised discussion

*Lines 383-384: You talk very little about photolysis in your discussion so I would remove it here.*

We agree to the Reviewer's comment. We will remove photolysis in our conclusion as this is not a major finding in our study.

*Lines 385-397: I would move this paragraph on potential temporal effects into your discussion section (see my previous comment). Then summarize it in your conclusions.*

We will follow the Reviewer's suggestion to move the paragraph into the discussion session and summarize it in our conclusion.

**Technical corrections**

*Line 24: Faulty parallelism: replace "and affecting" with "and affects"*

We will edit the text to improve parallel structure.

*Line 48: replace "Its estuarine part stretches" with "Its estuaries stretch"*

We will edit the text for conciseness. We will replace it to "Its estuary".

*Line 83: change to "within 20 minutes OF sampling"*

We will edit the grammatical error.

*Line 93/eqn. (1): It's confusing to have the letter "x" as the multiplication sign here because you also have an x variable. Use the mathematical symbol you use below or just take them out.*

For consistency, we will use the multiplier symbol $\times$ all throughout the manuscript.

*Line 120/eqn. (10): Write $e^{0.0447T}$ not exp.*

We will edit the text to reflect the correction pointed out by the Reviewer.

*Line 124/eqn. (12): $p_{NO}$ and $K_H$ are quantity symbols - italicize here as you did above.*

Thank you for your attention to detail. We will italicize the quantity symbols.

*Figures 2, 3, and 5: I would put the y axis labels (salinity, temperature, etc.) on the left side with the y axis ticks - it's confusing to have them on the opposite side of the plot. You can move the subplot labels ("a", "b") to the upper left corner.*

Thank you for the comment. We understand the importance of clarity in presenting scientific data. We will edit the Figures to reflect the comment of the Reviewer.

*Line 158: Add salinity units.*

I initially added the practical salinity unit (psu) as a unit for reporting the salinity from sensors, but I learned that this is a common mistake and is strongly discouraged:

"It is important to emphasize that Practical Salinities do not have units. This fact, confusing to non-specialists, is related to technical issues that prevented an absolute definition when PSS-78 was constructed. Sometimes this lack of units is awkwardly handled by appending the acronym PSU (Practical Salinity Units) to the numerical value, although doing so is formally incorrect and strongly discouraged."

See Pawlowicz, R. (2013) Key Physical Variables in the Ocean: Temperature, Salinity, and Density. *Nature Education Knowledge* 4(4):13 Available at https://www.nature.com/scitable/knowledge/library/key-physical-variables-in-the-ocean-temperature-102805293/

Therefore in our revised manuscript, we will retain the text excluding units for salinity.

**Figure 6: This is a really nice compilation plot to put your measurements in context. Instead of saying the NO fluxes are x10$^{-17}$, just report in units of fmol cm$^{-2}$ s$^{-1}$.**

Similar to my response in Line 232: We agree that it would have been better to use the shorter name. However, for the sake of intercomparability with previous research, we decided that we will use scientific notation in reporting the flux values.

*Table S3: Table S3: instead of superscripts "a", "b" and "c" corresponding to different significance levels, use \*, \*\*, and \*\*\*, which is the convention.*

We will edit the superscripts and use *,**,and *** to signify different significant levels.

*Figure 7: Use \*, \*\*, and \*\*\* instead of a, b, c superscripts.*

We will edit the superscripts and use *,**, and *** to signify different significant levels.

*Lines 332-333: I agree with the ammonium limitation idea but rephrase this and the following sentences to improve clarity and flow.*

*Line 344: Here and elsewhere: "were" not "are", since most of your results are reported in past tense.*

We will ensure to check again the grammar and results are reported in the past tense. However we will use the present tense to report trivial information.

*Line 348: Remove clause "when the nitrification proceeds" – unnecessary.*

We will remove the unnecessary clause.

*Line 350: Remove "therefore" - the support for this statement comes later in this paragraph, not from the preceding one.*

We will remove "therefore" in the revised manuscript

**Line 355: "correlations" should be plural.**

We will correct the grammar errors in the revised manuscript.

**References:**

Deek, A., Dähnke, K., Beusekom, J. van, Meyer, S., Voss, M., and Emeis, K.: N2 fluxes in sediments of the Elbe Estuary and adjacent coastal zones, Mar. Ecol. Prog. Ser., 493, 9–21, https://doi.org/10.3354/meps10514, 2013.

Liu, T., Xia, X., Liu, S., Mou, X., and Qiu, Y.: Acceleration of Denitrification in Turbid Rivers Due to Denitrification Occurring on Suspended Sediment in Oxic Waters, Environ. Sci. Technol., 47, 4053–4061, https://doi.org/10.1021/es304504m, 2013.

Olasehinde, E. F., Takeda, K., and Sakugawa, H.: Photochemical Production and Consumption Mechanisms of Nitric Oxide in Seawater, Environ Sci Technol, 44, 8403–8408, https://doi.org/10.1021/es101426x, 2010.

Schreiber, F., Stief, P., Kuypers, M. M. M., and Beer, D. de: Nitric oxide turnover in permeable river sediment, Limnol. Oceanogr., 59, 1310–1320, https://doi.org/10.4319/lo.2014.59.4.1310, 2014.

Schroeder, F., Klages, D., and Knauth, H.-D.: Contributions of sediments to the nitrogen budget of the Elbe estuary, Int. Ver. für Theor. Angew. Limnol.: Verhandlungen, 24, 3063–3066, https://doi.org/10.1080/03680770.1989.11899231, 1991.

Schulz, G., Sanders, T., Beusekom, J. E. E. van, Voynova, Y. G., Schöl, A., and Dähnke, K.: Suspended particulate matter drives the spatial segregation of nitrogen turnover along the hyper-turbid Ems estuary, Biogeosciences, 19, 2007–2024, https://doi.org/10.5194/bg-19-2007-2022, 2022.

Sørensen, J.: Occurrence of nitric and nitrous oxides in a coastal marine sediment., Appl. Environ. Microbiol., 36, 809–13, https://doi.org/10.1128/aem.36.6.809-813.1978, 1978.

Xia, X., Liu, T., Yang, Z., Michalski, G., Liu, S., Jia, Z., and Zhang, S.: Enhanced nitrogen loss from rivers through coupled nitrification-denitrification caused by suspended sediment, Sci. Total Environ., 579, 47–59, https://doi.org/10.1016/j.scitotenv.2016.10.181, 2017.

Zafiriou, O. C. and McFarland, M.: Nitric oxide from nitrite photolysis in the central equatorial Pacific, J Geophys Res, 86, 3173, https://doi.org/10.1029/jc086ic04p03173, 1981.

---

## Author Response (AR1)

**General Remark from the Editor**

Dear Mr. Ingeniero,

We have received three external evaluations of your manuscript. While all reviewers acknowledge the importance and quality of the data reported in this work, they also raise significant concerns regarding the motivation and implications of the work (e.g., reviewer #1), the presence of heavy speculation in certain sections of the discussion (e.g., reviewer #2), and the conclusion regarding the primary source of NO (e.g., reviewer #3). Additional specific comments and suggestions have been provided by the reviewers, and I believe the manuscript would benefit from addressing these issues.

I invite you to submit a revised manuscript that carefully incorporates comments and suggestions from all reviewers. Please update the author's responses when submitting the revisions.

Best regards,
Associate Editor

**Response Letter**

Dear Dr. Yuan Shen and Anonymous Reviewers,

I wish to express my sincere gratitude for the time and effort you have invested in providing detailed and constructive feedbacks on our manuscript. Your insightful comments and suggestions have been invaluable in refining our work, significantly contributing to its improvement, and preparing it to be more suitable for publication in Biogeosciences. We have taken careful note of the concerns raised by the Reviewers and have meticulously addressed each of them in our revised manuscript. In our revised submission, we have implemented the following changes:

1. In our previous submitted manuscript, we briefly enumerated the areas where dissolved NO concentrations were already measured. This supported our argument that limited studies are done on NO in the marine environment. In our revised manuscript, we added sentences that provide more context on the importance of measuring NO concentration and estimating sea-to-air flux densities.

2. We enhanced the clarity of our discussion section. We acknowledge the reviewer's point that the discussion section requires strengthening, particularly in establishing a robust causal link between the observed correlations and our discussion/conclusions. In the revised manuscript, we enhanced the clarity of the manuscript, ensuring that the role of NO as an intermediate in the nitrogen cycle is comprehensively explained and clearly articulated. We also provided references to substantiate the use of ratios we included in our discussion. For example, it has been well-established that a linear relationship between $\Delta N_2O$ and AOU indicates the occurrence of $N_2O$ production from nitrification (Yoshinari, 1976; Nevison et al., 2003; Schulz et al., 2023b). Furthermore, to assist readers unfamiliar with the complexities of the role of NO in the nitrogen cycle, we included chemical equations (R2 to R4). By providing these, the readers will better appreciate the various correlations we reported between the different dissolved inorganic nitrogen substrates and NO and $N_2O$. Despite these limitations, we view this study as an initial step in laying the groundwork for future research.

3. We improved our conclusion (and our abstract). We corrected our conclusion that the nitrifier-denitrification process is the primary source of NO in the Hamburg Port Area. Moreover, we also did not exclude the possibility of the anammox process without other evidence to rule it out.

4. We revised the Figures to follow Reviewer 3's comment on placing the axis titles on the right and using asterisks ($^*, ^{**}, ^{***}$) instead of superscript letters ([a],[b],[c]) to indicate the statistical significance level.

We hope that the modifications made to our manuscript have thoroughly addressed the issues highlighted by the Associate Editor and the Reviewers. In response to their valuable feedback, we have meticulously prepared a point-by-point response to ensure that each concern has been carefully considered and resolved. Our revisions include comprehensive updates to the text, revision of Figures, and inclusion of new references, all aimed at enhancing the clarity, depth, and impact of our work. We are confident that these revisions have significantly improved our manuscript, making it a more robust and valuable contribution to the field. We appreciate the opportunity to refine our work based on the insightful feedback provided and look forward to any further suggestions you may have.

Note that Reviewer comments are written in bold italics and our answers are kept in plain font.

Sincerely,

Riel Carlo O. Ingeniero
On behalf of all Authors

**1. Response to Reviewer 1 (RC1)**

*The authors present a recent effort of NO measurement in the Lower Elbe Estuary and the Hamburg Port Area, filling research blanks of this trace gas in coastal areas and estuaries. This manuscript is well-organized, with nice figures. It does provide an important picture of estuarine NO, an active trace gas difficult to measure, showing the distribution, flux, and potential production mechanisms of NO in the study region.*

*However, I have two major concerns here (also see specific comments below):*

1. *This paper is a good case study, but, as a manuscript expected to be published in bg, the text is lacking in the laying out of the scientific issues as well as extrapolation. For example, in the introduction there is a lack of elicitation of the gaps for the current research, and in the discussion, there is a lack of implications of the conclusions for other research in the field (i.e., what is the new knowledge compared to other published NO studies).*

2. *The whole discussion section and the present of implications is still weak, e.g., the main conclusions are mainly drawn through correlations, but without sufficient explanation and logic connection between correlation and their conclusion. This problem is particularly evident in section 4.4.*

*In the present version, I think there are still some gaps away from the publication level, and a major revision would be recommended.*

We thank Reviewer 1 for dedicating her/his/their time and effort to offer constructive feedback, which is instrumental in enhancing our manuscript. We acknowledge the reviewer's feedback to expound on the gaps in research on nitric oxide in the marine environment in our Introduction section. We revised our manuscript to mention these gaps in our paper.

We acknowledge the reviewer's point that the discussion section requires strengthening, particularly in establishing a robust causal link between the observed correlations and our discussion/conclusions. Our approach was to interpret the available data in order to explain the patterns of NO distribution in the Elbe Estuary. In our revised manuscript, we enhanced the clarity of the manuscript, ensuring that the role of NO as an intermediate in the nitrogen cycle is comprehensively explained and clearly articulated. We addressed points raised by the reviewers to enhance the discussion section.

The reviewer noted that we have an insufficient explanation of the correlation analysis on nitrogen nutrients, NO, $N_2O$, excess $N_2O$ ($\Delta N_2O$), and apparent oxygen utilization (AOU). We recognize that relying on correlation alone may not adequately illustrate the complexities of NO cycling. We provided references to substantiate the use of the said ratios in our discussion. For example, it has been well-established that a linear relationship between $\Delta N_2O$ and AOU indicates the occurrence of $N_2O$ production from nitrification (Yoshinari, 1976; Nevison et al., 2003; Schulz et al., 2023b).

We have addressed the concerns highlighted by the reviewer and detailed the changes we intend to implement in the revised manuscript to address the reviewer's critiques. Reviewer comments are presented in bold italics, while our responses are in plain font.

**Specific comments:**

**Introduction**

*Lines 36-39 This is just a list of past study areas, and the authors should have devoted some space to specifying the major scientific conclusions and advances made by these studies in the marine environment NO.*

*Line 40 What is the research gap of NO? Where might the behavior of estuarine NO differ from that of the study areas described above, or what is the scientific importance of studying estuarine NO? These should be briefly described in the Intro section.*

We appreciate the reviewer's constructive comments on the Introduction section of our manuscript. We recognize the importance of providing a clear scientific context and the specific research gaps our study addresses. Our paper indeed presents a novel case study on the measurement of dissolved NO concentration on the interface between the riverine environment and coastal seas in a well-studied estuarine system in Europe— the Elbe Estuary.

In the Introduction section, we briefly enumerated the areas where dissolved NO concentrations were already measured. It supports our argument that limited studies are done on NO in the marine environment. We modified the paragraph from lines 34 to 39 and added paragraphs that provided context on the importance of measuring NO concentrations and estimating sea-to-air flux densities:

"These studies performed at different periods have indicated that both open and coastal seas are a source of atmospheric NO with fluxes ranging from 0.70 (Anifowose and Sakugawa, 2017) to as high as $45.00 \times 10\text{-}17$ mol cm-2 s-1 (Gong et al., 2023). Global estimates for oceanic NO emissions are still lacking, and studies on the temporal (i.e., diurnal, seasonal, interannual) and spatial variability of NO emissions are not available. To address these gaps, expanded measurements of NO distribution in the open ocean and coastal waters are essential to enhance our understanding and provide a more accurate assessment of sea-to-air flux densities.

A recent paper by Gong et al. (2023) argued that DIN plays an important role in NO distribution– a high level of dissolved inorganic nitrogen (DIN) establishes the necessary conditions for NO production. Other studies (e.g., Olasehinde et al., 2010; Anifowose et al., 2015; Anifowose and Sakugawa, 2017; Ayeni et al., 2021) also observed a positive correlation between NO concentrations or photoproduction rates with dissolved $NO_2^-$ concentrations. To our understanding, dissolved NO measurements and the magnitude of flux density in estuaries, which have relatively high DIN concentrations (Howarth et al., 2011), have not yet been reported."
**[Lines 44 – 55]**

**Method**

*Line 51 Define Elbe-km here.*

The definition of Elbe-km was moved from line 62 (Figure caption) to the main text. For better coherence, we moved the definition after the sentence "Originating from the Karkonosze Mountains in the northern region of the Czech Republic, the Elbe River basin is the fourth largest catchment area (148,268 $km^2$) in Central Europe (Amann et al., 2012) with average long-term freshwater runoff of about 720 $m^3 s^{-1}$ (Kerner, 2007)."
**[Lines 64 – 65]**

*Line 79 Method uncertainty and detect limit should be presented here.*

We included the uncertainty (i.e., the average standard error of 1.28%) and added citations to previous
publications for the methods' detection limit (Schulz et al., 2023; Brase et al., 2017). **[Lines 96 to 97]**

*Line 83 The text here says that triplicate NO samples are measured. But I don't see the error bars in the*
*figures. Uncertainty of NO flux density estimate also needs to be added.*

We revised the Figures and added the error bars in the NO concentration distribution and estimated NO flux
densities:

[Figure]

*Line 128-129 and Fig. S2. I was surprised by the range of data in the figure, which, given the error bars, can*
*range from − 5 to 15 μg/m³. I'm a bit curious whether this range of error is primarily from (a) limitations of*
*the detection method, (b) spatial heterogeneity, or (c) temporal variability. If it's from (a), the authors'*
*averaging method may be reasonable, and if it's from (b) and (c), how large are the potential calculation*
*errors? It looks like it might have (up to) an order of magnitude impact on the flux calculations.*

We acknowledge the Reviewer's concerns regarding the precision of our NO flux calculations. For the same
reason, we have clearly stated and emphasized in the manuscript that the calculated NO flux represents a rough
estimate. Ideally, measuring atmospheric NO concentrations directly onboard the research vessel would enhance
accuracy, as *in situ* measurements reduce potential errors in calculating flux.

Nonetheless, due to the lack of necessary additional onboard instrumentation (i.e., NO analyzer dedicated to
atmospheric measurement), we have followed a methodology similar to that used by Tian et al. (2019a),
published in Biogeosciences. They also used the average atmospheric NO concentrations (2.13 ppb) in their
study area for estimating flux density in the Bohai Sea. While their study just noted personal communication as
the source of the average atmospheric NO concentration, we provided the source of our data (i.e., atmospheric
NO measurement by the Hamburg Institute for Hygiene and Environment).

The atmospheric NO concentration was measured using the chemiluminescence method and follows the DIN
(Deutsches Institut für Normung e.V.) EN 1411 standard. The DIN is the German national organization for
standardization and is the German ISO member body. Calibration and quality assurance on measurement data
are discussed on their website (https://luft.hamburg.de/allgemeine-informationen/kalibrierung-und-
qualitaetssicherung-598742). In summary, they ensure the following:

▪     Use of Suitability-Tested Devices: Only devices that have passed suitability tests are employed.

- Regular Checks and Calibrations: Gas measuring devices are checked every 25 hours, and manual calibrations are performed quarterly or post-repair, using traceable standards to monitor and adjust for deviations and long-term drift.
- Traceability: Calibration standards are biennially compared with national and European reference laboratories to ensure alignment with European standards.
- Participation in Round Robin Tests: Annual nationwide and regional tests are conducted to synchronize standards and test instruments across federal states.
- Regular Maintenance: Comprehensive maintenance schedules are followed at all measuring stations in compliance with EN standards, with more extensive tests being less frequent but more intensive.
- Validation of Measurement Data: Data is manually reviewed daily, monthly, and annually to confirm its plausibility based on technical, meteorological, and empirical factors.

To improve the accuracy of our study, we selected all seven background monitoring stations located near the Hamburg Port Area. These designated monitoring stations measure background concentration levels of air pollutants and are typically far enough from emission point sources. We think that all the seven background stations near the Elbe Estuary reflect the ambient atmospheric NO concentrations over the Elbe Estuary. Moreover, to further minimize error, we specifically selected data from the period coinciding with our study. We did not include nighttime atmospheric NO measurements, typically lower due to reduced vehicular and industrial emissions at night. We used the average NO value at the seven background monitoring stations to provide a conservative estimate of the atmospheric NO concentration in the Elbe Estuary during the study period. If we look at the average values at each time point, it is near the average concentration of 4.3 µg m$^{-3}$ that we used to calculate the flux density. Notably, measurements outside the typical rush hours are close to this average concentration value. Here is the statistic of the hourly NO measurement (µg m$^{-3}$):

**Minimum: 2.00**
**Maximum: 8.25**
**Standard deviation: 1.76**
**Median: 3.86**

The Figure S2 caption should have been clearer that the error bars or whiskers in the scatter plot represent the standard deviation of the values measured at the "background" monitoring stations for each time point and not the minimum and maximum NO concentration values typical for box and whisker plots. We edited the caption to indicate that the error bars represent the standard deviation:

**Figure S2: Average hourly atmospheric NO concentration (µg m$^{-3}$) measured in seven background monitoring stations near the Elbe Estuary (AltonaElbhang, Billbrook, FinkenwerderAirbus, FinkenwerderWest, HafenKlGrasbrook, Veddel, and Wilhelmsburg) in Hamburg representative of the time of sampling. Note that the error bars represent the standard deviation. Shown in the red dashed line is the average concentration of 4.3 µg m$^{-3}$. These data were obtained from https://luft.hamburg.de/ (last accessed on 2 May 2023).**

Atmospheric NO concentration may vary spatially and temporally as NO$_x$ can be emitted from vehicles and ships. You would notice that high variability at each time point is more pronounced from around 6:00 to 8:00 AM, which may be attributed to the morning rush hour.

**Section 4.1**

*The discussion in this section was a bit weak. I really like the summary of NO in Figure 6, but there wasn't much discussion of it in the main text. For example, why is it that estuaries with higher nutrient instead have lower NO concentrations than open ocean/nearshore? This study' NO is already supersaturated but still on the lower end of all the studies, what is causing the high concentrations (potentially supersaturated several times over) on the other sites?*

*Will some of the correlation patterns in this work appear in whole compile data set? How important are*
*estuarine/oceanic NO emissions relative to terrestrial/human systems based on currently available data?*
*Etc… This may require more work to sort out, but I believe it may expand the scientific value of this paper to*
*be more than just like a case study.*

Thank you for your feedback. We acknowledge the need to improve the discussion and provide a more
comprehensive analysis of our results. We resolved this in our revised manuscript.

The question of why the Elbe Estuary, with relatively higher nutrients, has a lower NO concentration is still
unresolved and requires further investigation. Nevertheless, this observation is an important finding of our study.
This observation challenges the assumption that higher concentrations of nitrogen nutrients automatically lead
to increased dissolved NO concentration. In our manuscript, subsequent to our discussion highlighting the Elbe
Estuary's relatively higher nutrient levels yet lower NO concentrations than other study sites, we delve into the
conflicting findings concerning the relationship between NO distribution and nitrogen-containing nutrients.
**[Lines 322 to 324]**

We also reported the hypothesis of Ayeni et al. (2021) regarding these conflicting relationships: "…Likewise,
Ayeni et al. (2021) also noted that some rivers in Japan with higher $NO_2^-$ concentrations had lower rates of
photoproduction of NO and vice versa, attributing these imbalances to nitrogen cycling processes (nitrification,
denitrification, and anammox), which could produce or consume NO, or the photochemical transformation of
organic nitrogen from dissolved organic matter producing $NO_2^-$ to form NO in areas with low $NO_2^-$." **[Lines**
**332 to 335]**

The high reactivity of nitric oxide (NO) as a radical initiates various consumption mechanisms which may
influence its concentration in the Elbe Estuary. Zafiriou et al. (1979) reported that there is no evidence of
interaction between NO and metals under marine conditions, though NO is known to react with metals yielding
nitrosyl (M-NO) or iso-nitrosyl (M-ON) metal complexes (Ford and Lorkovic, 2002; Richter-Addo et al., 2002).
We are not certain whether reaction with transition metals is a sink in the marine environment, particularly in
coastal and estuarine environment as this has not been explored. Additionally, organisms (algae, phytoplanktons)
can both consume and produce NO. A recent paper by (Bange et al., 2024) noted that the consumption processes
for NO in the sea(water) are still unresolved.

While current literature suggests that coastal areas could potentially act as significant sources of emission to the
atmosphere, this may vary temporally and spatially across the studied sites. Up to now, the majority of the
literature reports positive sea-to-air flux, indicating emissions as a major sink; however, regional exceptions,
such as one measurement in the Shandong Peninsula (Gong et al., 2023), indicate that generalizations should be
made cautiously. **[Lines 259 – 267]**

Regarding the Reviewers' comment on the importance of estuarine/oceanic NO emissions relative to
terrestrial/anthropogenic emissions, papers from Bouwman et al. (2002a), Bouwman et al. (2002b), and Stehfest
and Bouwman (2006) provide global estimates of NO emissions from soils. They have sufficient data to have
global estimate of terrestrial emissions. While we can provide these data on terrestrial emissions, it does not help
in the discussion since there is still no estimate for NO emissions from coasts, estuaries, and the open ocean. We
decided to exclude terrestrial data in this paper to maintain the focus of the paper. It is our hope that we get more
NO measurement data from the marine environment to provide reliable estimate and compare with terrestrial
data.

**Section 4.2**

*Because salinity is also an indicator of mixing, the negative correlation with salinity noted here is likely to*
*represent "mixing" for NO (i.e., mixing affects both NO and salinity), not "salinity and freshwater input*
*influencing NO concentrations" (i.e., salinity/freshwater itself influences NO).*

We appreciate the reviewer's comment regarding the role of mixing in the observed negative correlation between salinity and NO concentrations. We recognize that mixing indeed plays an important role in the distribution of biogeochemical parameters in the Elbe Estuary. Indeed, Dähnke et al. (2008) noted that conservative mixing behavior could be observed in the Elbe Estuary irrespective of the season.

However, our intention in this section is to emphasize the significance of riverine/freshwater inputs as a primary source of higher NO concentrations. We supported our argument with two studies: one documenting relatively higher surface dissolved NO in the southern Bohai Sea due to the Yellow River's outflow ascribing it to high DIN input (Gong et al., 2023), and another study (Ayeni et al., 2021) noting a NO concentration gradient in the Kurose River, with downstream sections influenced by anthropogenic activities.

**Section 4.4**

***The source/sink of NO is so complex that I would suggest that the authors include a suitable concept fig in an attachment or in the main text to allow more readers to easy follow the processes you describe.***

We provided a brief text on known sources and sinks of NO:

"The major sources of atmospheric NOx are emissions from fossil fuel combustion and soils (Jaeglé et al., 2005). Until now, little is known about the distribution as well as the production and consumption processes of NO in the marine environment. Two known primary sources of NO in the ocean are NO photolysis from nitrite and NO production from phytoplankton, macroalgae, and the microbial nitrogen cycle. Bange et al. (2024) noted that the consumption mechanisms of NO in the marine environment are still unresolved." **[Lines 27 – 31]**

We understand the importance of ensuring clarity for readers and a broader audience. We provided simplified reaction (R2 to R4) of nitrification, denitrification, and anammox so readers can follow the complex nitrogen cycle processes discussed. **[Lines 356 – 362]**

**Section 4.4.1**

***Lines 322-323 Why this statement make sense? Nitrification only contribute minor part of AOU. Some explanations or references are needed.***

We thank the reviewer for this comment and the opportunity to clarify this statement. Indeed, nitrification only contributes a minor part to the AOU. We understand the previous text could be enhanced for clarity, and as such, we edited the text to ensure that readers understand the text better.

However, it is established that a significant linear correlation between excess $N_2O$ ($\Delta N_2O$) and AOU indicates the occurrence of nitrification. We revised the text and provide references to support this argument. **[Lines 373 – 380]**

***Lines 324-325 I can't follow these sentences. Many ratios (e.g., $N_2O/NH_4^+$, $NO_2^-/O_2$ ...) appear in the correlation diagram. What do these ratios represent? Some background should be provided.***

We understand the need to enhance clarity for readers. We revised the text for readers to understand these ratios. We provided a simplified reaction steps for nitrification (R2), denitrification (R3), and anammox (R4) for readers to understand how these ratios might be related to the different nitrogen cycling processes. For instance, by providing chemical reaction R2 (i.e. the nitrification process), it would be easier to pinpoint that $N_2O$ is a product, $NH_4^+$ is a reactant, and $NO_2^-$ and $NO_3^-$ can be oxidized from NO.

***Lines 326-327 How "a significant positive linear relationship exists between $N_2O$ and $NO_3^-$" is linked to "These findings point to NO production via nitrification"? I can't find the logic connection.***

We revised the text to clarify the link between $N_2O$ and $NO_3^-$ in nitrification. To establish a logical connection
between these statements, it's important to understand the following:

- Nitrous oxide ($N_2O$) is a known byproduct of nitrification and an intermediate of the denitrification
  processes.
- Nitrification is a microbial process where ammonia ($NH_3$) is oxidized to nitrate ($NO_3^-$), and it can also
  lead to the production of nitrite ($NO_2^-$).
- During nitrification, obligatory intermediates (Caranto and Lancaster, 2017), nitric oxide (NO) and
  hydroxylamine ($NH_2OH$) can be produced. NO can further yield $N_2O$.

The significant positive linear relationship between $N_2O$ and $NO_3^-$, may suggest that as the concentration of
nitrate increases, so does the concentration of $N_2O$. If this relationship is found to be significant within the
context of the study, it is possible that the processes leading to the production of $NO_3^-$ (like nitrification) are also
associated with the production of $N_2O$ (see Schulz et al., 2023). Hence, if $N_2O$ levels are rising with $NO_3^-$ levels,
it could be indicative of active nitrification, during which NO is produced as an intermediate. The logic is that if
$N_2O$ is increasing with $NO_3^-$ and we know that $N_2O$ can be a byproduct of nitrification (which also produces
$NO_3^-$) then an increase in both could point to nitrification as the source process, and thus, the production of NO
as part of that process.

***Line 331 What "observed trends" refer to?***

We revised the manuscript to enhance clarity for the reader and avoid unspecific phrasings.

***Line 334 Authors discuss here that nitrification is the SINK of NO. I am a little confused because the whole***
***section discusses about nitrification as SOURCE of NO.***

Note that while NO can be produced in the nitrification process as an obligatory intermediate (Caranto and
Lancaster, 2017), it can also be consumed in further oxidation steps. Shown below is **Figure 1** from Caranto et
al. (2017):

[Figure]

[Figure]

**Figure 1: Schematic diagram comparing the prevailing view on the nitrification process and the model proposed by Caranto**
**and Lancaster (2017) that shows nitric oxide is an additional obligate intermediate in the nitrification process (From "Nitric**

**oxide is an obligate bacterial nitrification intermediate produced by hydroxylamine oxidoreductase," by J.D. Caranto and**
**K.M. Lancaster, 2017).**

In the nitrification process, ammonia ($NH_4^+$) undergoes oxidation to form hydroxylamine ($NH_2OH$), which can
further yield NO and then form $N_2O$, $NO_2^-$, or $NO_3^-$. Another Reviewer agreed with the idea of the $NH_4^+$
limitation in the coastal/brackish and limnic zones leading to the observed significant inverse relationship
between NO and $NO_2^-$ and NO vs $NO_2^-/O_2$ ratio. If $NH_4^+$ is not limited or has a continuous supply in the reaction,
one would see a direct relationship between NO and $NO_2^-$. When $NH_4^+$ is limited, NO will be consumed in the
process, decreasing its concentration while increasing the product $NO_2^-$, $NO_3^-$, or $N_2O$.

For clarity, we added, the chemical reaction equations (R2 to R4), which provides a general overview of the
nitrogen cycle involving NO. **[Lines 357 – 362]**

*Section 4.4.2*

*This entire section suffers from a problem like that of section 4.4.1, in that a large amount of the text simply*
*suggests the correlation without explaining it, making the logical chain of support for the author's argument*
*incomplete. For example, almost all of the text in lines 350-365.*

We understand the need to enhance the manuscript by providing a thorough explanation of the nitrogen cycle
processes. We revised and restructured the manuscript for clarity. **[Lines 396 – 417]**

*Other notes:*

*Table S2: Why NO flux density (mol $m^{-2}$ $s^{-1}$) have a different unit with $N_2O$ flux density ($\mu$mol $L^{-1}$ $d^{-1}$)? It*
*also differs from unit in the main text and figure 5 and 6.*

Thank you for the attention to detail. We apologize for the oversight. We have corrected the unit of NO flux
density to mol $cm^{-2}$ $s^{-1}$ in Table S2, and the unit of $N_2O$ flux density to $\mu$mol $m^{-2}$ $d^{-1}$. These are standard flux
density units established in prior publications. For easier comparability and consistency with previous
publications, we have used the units $\mu$mol $m^{-2}$ $d^{-1}$ for $N_2O$ flux density and mol $cm^{-2}$ $s^{-1}$ for NO flux density.

*Why don't you add NO to the correlation plots of the main text and attachments? I don't see NO in Figure 7*
*and Figures S4-S6? And if space permits, I suggest you place Fig. S4 (after adding NO) and Fig. S7 into*
*main text.*

We followed the suggestion of the Reviewer to add correlation plots of NO **[Figure 8, Line 392 – 394]** to the
main text. We removed NO in Figure 7 and Figures S4-S6 because we have made separate correlation plots of
NO vs other parameters.

**2. Response to Reviewer 2**

*This is an interesting paper providing new data on NO distribution and fluxes and with potential for*
*improving our insight in the complex biogeochemistry of N-transformation in estuarine/riverine*
*environments. The analytical procedures for data acquisition are explained in detail and the quality of the*
*data seems very robust.*

*Next to presenting the estuarine profile of NO concentrations and fluxes together with other physico-chemical*
*parameters (Temp, Sal, $O_2$, nuts, Chl..) authors proceed with discussing possible processes steering the*
*observed distributions. This is done exclusively based on regression analyses. I found this part of the paper*
*based on a lot of speculation, forcibly as no other tools permitting process identification and process rate*
*assessment were applied. This weakens somewhat the strength of the paper which therefore rests mostly on*

*the quality of the analytical part. Especially N, O isotopic composition measurements of nutrients could*
*possibly confirm/infirm occurrence of nitrification/denitrification and resolve impact of both processes. I can*
*understand such an approach was not possible in the present context, but isotopic data for the Elbe have been*
*published by others (Dähnke et al.), and some thoughts on how these fit with the present observations might*
*have been a useful addition to the paper. Can authors comment on this?*

We are grateful to the Reviewer for dedicating his/her/their time and effort to provide constructive feedback,
which is instrumental in enhancing the depth and clarity of our manuscript. We are heartened by the positive
evaluation of our analytical procedures and the robustness of our data.

Indeed, to explain the NO distribution observed during our campaign in the absence of additional data, we
employed regression analysis to assess the relationship of NO with various dissolved nitrogen substrates.
Regression analysis allows us to determine the degree to which NO concentrations vary with changes in the
levels of these parameters, providing insights into potential underlying biogeochemical and microbial processes.

Our analysis revealed significant correlations between NO and other measured parameters. These significant
correlations are suggestive of systemic relationships that may not be immediately apparent without statistical
investigation. By employing regression analysis, we were able to quantify the strength and direction of these
relationships, offering a foundation for hypothesizing about the interactions occurring between NO and these
parameters. This method, while inferential, presents a valuable first step toward understanding complex
environmental interactions, particularly when more direct methods of assessment are not available.

The significant findings from the regression analysis warrant further study. We acknowledge that our approach
could be enriched by incorporating other biogeochemical tools such as measurements of
nitrification/denitrification rates, assays for nitrogen marker genes, and analyses of stable isotopes of nitrogen
and oxygen.

Nevertheless, as the initial measurement of NO in the area, our study could lay the groundwork for future
research. In earlier manuscript drafts prepared for submission to Biogeosciences, we explored including data on
dual stable isotopes of nitrate. Our analysis indicated that mixing or dilution predominantly affects the
coastal/brackish and limnic zones, with nitrogen cycling processes being more pronounced in the Hamburg Port
area. We ultimately decided against including this data to maintain the focus of our manuscript without delving
into the intricacies of dual stable isotopes of $NO_3^-$ in a study not primarily focused on stable isotope
biogeochemistry. We believe this decision helps maintain clarity and focus in our paper.

***Specific comments:***

***Were any data obtained for the tributary rivers Oste, Meden, Stör ?***

No data were obtained for the tributary rivers Oste, Meden, and Stör. However, we still added this on the Map
and Figure legends since there are a few sentences in the manuscript that we referred to Oste and Meden. It might
guide readers not familiar with the study site about the tributaries we mentioned in the manuscript.
▪ An increase in $NO_2^-$ and $NH_4^+$ concentrations was also observed downstream of the maximum turbidity
zone (Dähnke et al., 2022) at the confluence of River Oste and Meden.
▪ Concentrations started to increase slightly above the detection limit at the outflow of the River Meden
near Otterndorf at Elbe-km 710 and 714.
Regarding Stör, we observed the following: A slight decrease in $O_2$ concentration and pH and a slight increase
in chlorophyll a (Fig. 2) and nitrate (Fig. 3) concentrations at the confluence of River Stör. These are minor
changes that noticeably deviate from the general mixing in the Estuary. We did not discuss this in the manuscript
as it deviates from the main focus of the paper. However, future researchers working on rivers and estuaries
might conduct further research on the influence of tributaries on NO dynamics in the Elbe Estuary.  As suggested, we edited the caption describes the tributaries to enhance clarity. We also corrected the typographic error Stor
to Stör in the revised manuscript.
**In section 4.3 photolysis is mentioned as a source of NO but this is very little discussed further. Can it be**
**a significant process in a turbid estuary? Are there data for suspended matter load, vertical light profiles?**

Previous research (Zafiriou and McFarland, 1981; Zafiriou and True, 1979; Gong et al., 2023) has established
that the photolysis of nitrite ($NO_2^-$) constitutes a primary source of nitric oxide (NO) in marine environments.
However, the significance of this process in turbid estuarine systems, such as the Elbe Estuary, remains an open
question.

The literature presents conflicting evidence regarding the influence of nitrite concentrations on the levels of
dissolved NO. Some studies suggest a direct correlation, while others do not find a significant relationship,
indicating the complexity of the factors that control NO levels in dynamic environments. We did not find any
direct relationship between NO and suspended particulate matter.

**Lines 313-315: Can presence/absence of anamox activity be confirmed based solely on information of O2**
**conc.? Could this process possibly proceed inside micro-environments such as aggregates, flocs with low**
**internal O2?**

We agree that the current data does not definitively rule out other processes, such as anammox, and that it is
prudent to consider such as an alternative NO production source. We revised the manuscript to reflect a more
balanced view of potential NO sources, acknowledging the strong correlations observed and how they may also
include anammox. **[Section 4.4.2, Lines 397 – 417]**

*Lines 322-326: As written, the reader gets the impression AOU is solely set by $O_2$ consumption during*
*nitrification. What about respiration?*

In our discussion of the linear relationship between $\Delta N_2O$ and AOU, it appears the original text led to the
misunderstanding of AOU being solely attributed to oxygen consumption during nitrification. We acknowledge
this oversight and clarify that AOU is influenced by a variety of biological and chemical processes in the ocean,
including both nitrification and aerobic respiration. We improved the clarity of the text to present our intention
to note that the significant linear relationship between $\Delta N_2O$ and AOU is usually associated with $N_2O$ production
through nitrification (Schulz et al., 2023a; Brase et al., 2017; Nevison et al., 2003; Walter et al., 2006). **[Lines**
**373 – 380]**

*Lines 335-336: This sentence leaves us asking so what ? Detail the meaning. How does it clarify the foregoing*
*statement?*

The sentence in Line 335-336 reads: "Furthermore, we observed that five sampling sites in the coastal-brackish
zone with $O_2 > 200$ μM had NO concentrations less than the detection limit (Fig. 6)."

This sentence discussed the observation that NO concentration appears to be very low in the coastal-brackish
zone, probably due to the relatively higher oxygen concentration. It is known that NO is reactive with $O_2$. In the
nitrification reaction, NO can be oxidized further to $NO_2^-$ and $NO_3^-$. We removed this sentence and restructured
the Discussion section.

*Lines 361-365: These statements are unclear and the rationale is difficult to follow. Try to clarify.*

We edited these statements and discuss in detail the nitrification process. Another Reviewer agreed that the lack
of correlation between nitrate and apparent oxygen utilization (AOU) in the Hamburg Port area may indicate the presence of denitrification or nitrifier-denitrification or any process that influences nitrate. However, we cannot
rule out one or another. What we can conclude from the data is that there is a process other than nitrite oxidation
that influences nitrate concentration in the Hamburg Port. **[Section 4.4.2, Lines 409 – 417]**

*Lines 366-368: The possible role of suspended particles with low internal O2 is mentioned for the port area.*
*How does this look in the downstream maximum turbidity zone ?*

We added this text:

"We noted that both NO and $N_2O$ concentrations started to increase downstream of the maximum turbidity zone
near the confluence of River Meden and Stör." **[Lines 434 – 436]**

*Lines 400 and further (Conclusions): Will a higher temporal resolution and improved sampling strategy be*
*sufficient to get insight into the dynamic interplay of controlling factors? Would adding stable N, O isotopic*
*methodologies be helpful ?*

Increasing the temporal resolution of our sampling would indeed yield helpful information on whether
seasonality affects NO concentration and sea(water) to air fluxes. To date, no study has done this; doing so will
enhance our understanding of the nitrogen dynamics and the processes of NO production and consumption
within the estuary. Such detailed temporal data could reveal patterns that are not discernible at lower sampling
frequencies including diurnal cycles and episodic events.

Incorporating dual stable isotope techniques and the measurement of process rates across all sampling sites
would significantly strengthen our study. This methodology would allow us to trace the pathways of nitrogen
transformations more precisely and could provide definitive evidence of nitrification and other nitrogen-related
processes. Additionally, the use of molecular or genetic tools to detect marker genes specific to nitrogen-cycling
microbes would offer insights into the microbial contributions to observed nitrogen transformations. These
genetic markers could help us pinpoint the active microbial communities and link them to the biogeochemical
processes we are studying. Overall, integrating these advanced methods in future studies will deepen our analysis
and provide a more comprehensive interpretation of the results. We have incorporated this in the Conclusion;
see **Line 463 – 471.**

**Technical issues:**

*Figure 1: legend should mention tributaries Oste, Meden, Stör.*

We have revised the Figure caption to mention the tributaries.

*Line 336: reference to Fig. 6.. is this correct, or should it be Fig. 5 ?*

Thank you for your attention to detail. We ensured that the revised manuscript properly referenced the Figures
in the text.

*Line 361: Fig 7g should be Fig S7g*

Thank you for your attention to detail. Similar to our response above, we thoroughly checked that the revised
manuscript properly referenced the final Figures in the text.

*AOU is given without unit*

We apologize for the oversight. In our revised manuscript and supplementary file, we provided the unit (μmol
$L^{-1}$) of AOU.

**3. Response to Reviewer 3**

*Summary*

*The manuscript titled "Dissolved Nitric Oxide in the Lower Elbe Estuary" by Ingeniero et al. quantified the*
*fluxes of nitric oxide (NO) in relation to other nitrogen cycle parameters in the Elbe River Estuary and*
*Hamburg Port Area. Using a clever chemiluminescent detection method and flow-through sampling system,*
*the authors measured dissolved NO concentrations in surface waters alongside temperature, salinity, pH,*
*and dissolved oxygen ($O_2$). The authors made concurrent measurements of nitrate, nitrite, and ammonium*
*with an autoanalyzer and nitrous oxide ($N_2O$) with laser spectroscopy. The authors found that NO was*
*supersaturated in the surface layer of both study areas, so they were both a source of NO to the atmosphere.*
*Based on the concurrent [$O_2$] and dissolved inorganic nitrogen measurements, the authors conclude that*
*this NO is likely produced via biological processes (nitrification, denitrification, and nitrifier-*
*denitrification), as opposed to the photolysis of nitrite.*

*General Appraisal*

*In this paper, the authors present the first-ever measurements of NO in the Elbe River system. NO*
*measurements in the literature are scarce because its short lifetime makes analysis difficult, so this paper*
*represents a substantial contribution to our understanding of NO in the marine environment. Furthermore,*
*the authors measure significant NO supersaturation and fluxes in the surface waters of much of the Elbe*
*River, which is important because NO is a contributor to smog, acid rain, and ozone.*

*The major strengths of this paper are the presentation of novel, high-resolution NO measurements and the*
*clear relationships that emerge between NO and other inorganic nitrogen species, [$O_2$], pH, and*
*chlorophyll. The authors present a clean, concise interpretation of these results and the paper is generally*
*straightforward and easy to read.*

*The major weakness of this paper is that the discussion of temporal variability (day/night and seasonal*
*variations) is not linked to the clear boom-and-bust cycle seen in the Hamburg Port area. The authors have*
*locations with peaks of chlorophyll and [$O_2$], and other locations with oxygen and pH minima and $N_2O$ and*
*NO maxima. This implies to me that there are some locations where you captured net production and others*
*where they captured net respiration, which draws down [$O_2$] and creates an ideal environment for $N_2O$ and*
*NO production in sediments or particles. The authors allude to this in the conclusions, but how would day-*
*night temporal variation at each site affects the data? Would blooms in some locations propagate*
*downstream and create pockets of high respiration further downstream? The authors have a paragraph in*
*the conclusions about potential temporal effects, and my suggestion would be to move this paragraph into*
*the discussion and link it more clearly to their results.*

*The paper is generally well-written. There are only a few grammatical errors and clumsy sentences that I*
*note in the technical corrections.*

*My primary concern is about the conclusion (and I believe this is only stated in the abstract) that nitrifier-*
*denitrification is the primary source of NO in the Hamburg Port area. While I agree that the lack of*
*correlation between nitrate and apparent oxygen utilization (AOU) in the Hamburg Port area may indicate*
*the presence of denitrification or nitrifier-denitrification, I don't think you can rule out one or the other. In*
*other words, all you can conclude from this data is that there is a process other than nitrite oxidation that is*
*consuming nitrite. Likewise, if you invoke denitrification and/or nitrifier-denitrification in sediments or*
*particles, I don't think you can rule out the presence of anammox. In fact, instead of ruling out anammox*

*based on water column [O₂], you should list it as a potential alternative source of NO. The strong*
*correlations between NO, nitrite, and ammonium may indeed be a sign of anammox as a source of NO in*
*the Hamburg Port area.  Also, while denitrification and/or nitrifier-denitrification may be present in this*
*zone, the water column [O₂] suggests that the primary source of NO would still be nitrification, and this is*
*supported by the strong correlations in this zone between NO, nitrite, and ammonium.*

We appreciate your recognition of the novel contributions our work makes to the field – the first-ever
measurements of NO in the Elbe River system and the identification of significant NO supersaturation and fluxes
in surface waters. Your acknowledgment of the clarity and readability of the paper is encouraging.

Your critique concerning the discussion of temporal variability and its connection to the observed
biogeochemical cycles within the Hamburg Port area is well-founded. We incorporated your suggestion in our
revised manuscript.

Regarding the primary sources of NO, we acknowledge the Reviewer's concerns about the conclusiveness of
nitrifier-denitrification as the dominant process in the Hamburg Port area. We agree that the current data does
not definitively rule out other processes, such as anammox, and that it is prudent to consider such alternative NO
consumption processes. We revised the manuscript to discussion to have a more balanced view of potential NO
sinks or sources, acknowledging the strong correlations observed and how they may implicate various nitrogen-
transforming processes, including anammox.

Thank you for the helpful and very detailed comments. The detailed suggestions were implemented to enhance
the manuscript's technical quality. We have addressed the concerns highlighted by the reviewer and detailed the
changes we intend to implement in the revised manuscript to address the reviewer's critiques. Reviewer
comments are presented in bold italics, while our responses are in plain font. We look forward to submitting a
comprehensive revised manuscript that addresses the points you've raised.

**Specific comments**

***Line 14: Is the same chemiluminescent optode spot system often used for O₂ (Frey et al., 2023)?***

No. The luminescence measuring oxygen sensors used by Frey et al. (2023) are different from our detection
method. We used a chemiluminescent method for NOx which is typically used for atmospheric monitoring of
NOx. Lutterbeck and Bange (2015) describe the method in detail. In our earlier drafts of the manuscript, we
cited the method paper by Lutterbeck and Bange (2015) in the Abstract for clarity. However, adhering to
standard writing practices, we omitted this citation from the Abstract in the final draft when we submitted the
paper to Biogeosciences. This paper, if published, would be the first application of the method in a coastal and
estuarine environment. We edited the text in the abstract as follows:

"The discrete surface water samples were analyzed using a chemiluminescence NO analyzer connected to a
stripping unit." **[Lines 13 – 14]**

**Line 15: Why not write pM instead of $10^{-12}$ mol/L? You do so later in the manuscript.**

Thank you for your comment. For consistency, we followed your suggestion to use pM.

**Line 20: Based on your discussion, this could be nitrifier-denitrification or denitrification. I don't think**
**you can rule out one or the other based on your data.**

We agree with this comment. While we cannot rule out which exact nitrogen cycling processes could be
present, we think that nitrifier-denitrification or denitrification influences the NO distribution in the Hamburg
Port Area. We have edited the text to reflect a more balanced view. **[Lines 19 – 20]**

**Line 34: What is the lifetime of NO in seawater/water?**

The lifetime of nitric oxide (NO) in seawater or water is relatively short due to its high reactivity. In aquatic
environments, NO can rapidly react with oxygen, metals, and organic compounds. The exact lifetime can vary
depending on several factors, including temperature, pH, and the presence of reactants, but it is typically on the
order of a few seconds to a few minutes (i.e. 3 – 100 s) (Zafiriou and McFarland, 1981; Olasehinde et al.,
2010). We provided these values in the revised manuscript.

**Line 72: The way this equation is written is confusing. Are you multiplying the corrected $O_2$ by 1.12? Or**
**the uncorrected? What are the units of the intercept? Also, does the intercept of 13.41 mean that the**
**detection limit of the oxygen optodes is 13.41 (units?)?**

We edited the $O_2$ correction equation. The revised equation was stated as: $[1.12 \times O_{2(optode\ measurement)}] + 13.41$
($R^2 = 0.97$). The unit is μM. **[Lines 88 – 90]**

**Line 83: Give us some numbers for what this lifetime is**

Please see our response above (-> Line 34). We added these values in the revised manuscript.

**Line 84: So the calibrator is just an NO source, right?**

Yes, this is right. It is a portable calibration source that operates using a compact nitrous oxide ($N_2O$) cartridge,
producing gas output that is traceable to the US National Institute of Standards and Technology (NIST)
standards, as detailed in the study by Birks et al. (2020). **[Lines 101 – 102]**

**Line 90: Why do you need the calibrator in addition to the aqueous NO standard solutions?**

The calibrator is used to adjust the NO analyzer, ensuring its responses are accurate and reliable. This step is
fundamental because it directly affects the instrument's precision and accuracy, ensuring that its readings are
consistent with "true" NO concentrations. Calibration with the calibrator involves adjusting the instrument's
response to known concentrations of NO gas. This process ensures that the instrument's detection and
measurement systems are properly aligned with the actual concentrations, correcting for any drift, sensor
degradation, or other factors that might affect accuracy over time. Meanwhile, the aqueous NO standard
solution is used for method calibration.

*Line 94: This calculation is to convert the mole fraction you measure in the headspace to the dissolved NO*
*concentration, right? Is there a reason to assume that the headspace is at a pressure of 1 atm? I would*
*assume it would be slightly over pressurized... how would that affect your measurements?*

We used the stripping method detailed in Lutterbeck and Bange (2015).  Furthermore, the NO analyzer
operates with atmospheric pressure input and will display an error if it exceeds a certain pressure threshold. A
needle valve was also installed to reduce pressure variations.

*Line 97: Here you use pM. I would stick to this throughout the text.*

Thank you for pointing out the inconsistency. We have revised the manuscript to ensure that 'pM' is
consistently used throughout the manuscript.

*Lines 102-103: In eqn. (2) you assume the barometric/atmospheric pressure is 1 atm. Is this a reasonable*
*assumption at this time of year, in this part of the world?*

The average air pressure in Hamburg during this time is at 1009 hPa, or when converted to atmosphere, is
0.9958 atm which is close to 1 atm.

See https://meteostat.net/en/place/de/hamburg?s=10147&t=2021-07-27/2021-07-29 (last accessed 1 March
2024), which uses weather data from NOAA.

*Line 125: Same comment as above with setting atmospheric pressure to 1 atm.*

Please see our response above for Lines 102-103.

*Lines 129-130: How was this mean value calculated? Mean of all hourly measurements at all monitoring*
*stations over the study period? Given the short lifetime of NO, doesn't it make sense to calculate a*
*mean $c_{EQ}$ on a day-by-day or even shorter basis - or do all of the stations look like figure S2, where the*
*hourly concentrations are all within error of the average?*

You are correct. This is the mean of the average hourly measurement at all monitoring stations over the study
period. We excluded nighttime values as NO concentrations are rather low in the evening due to low emissions
from vehicles. We think this is a conservative estimate of the NO concentration in the Elbe Estuary.

*Lines 172-174: Is the variability of [$O_2$] because of changes in productivity?*

The variability in [$O_2$] levels can indeed be a result of changes in productivity. Note that the measurements were
taken during the daytime when net productivity should be higher. During photosynthesis, phytoplankton
consume $CO_2$ and release $O_2$, which increases the [$O_2$] in the water. The higher the phytoplankton productivity,
the more $O_2$ is produced. Additionally, photosynthesis affects pH levels. As phytoplankton consume $CO_2$, they
can reduce the amount of $CO_2$ in the water, which can cause the water to become less acidic (increase in pH
level). We briefly mentioned this in the Results **[see Lines 193 – 194]**

*Lines 184-185: Report a number for the maximum concentration to give a sense of scale - 200 µM is a lot!*

To improve specificity, we edited the sentence and provide the value of the concentration.

Overall, the DIN concentrations (Fig. 3f) increased from the mouth of the estuary upstream, with the highest
concentrations (201 µM) recorded just before the Hamburg Port area (see also Fig. S3). Further details on the
concentration of the DIN substrates are presented in the next section. **[Lines 202 – 203]**

*Lines 200-201: It looks like the peaks in $N_2O$ correspond to the minima in [$O_2$] - if that's the case, worth*
*pointing out here.*

Yes, this is correct. We edited the sentence to emphasize $N_2O$ production in minimum dissolved $O_2$
concentration **[see Lines 219 – 221].**

*Line 225: You should also mention that the peaks in NO in the Hamburg Port area correspond to the peaks*
*in $N_2O$, $NO_2^-$, and $NH_4^+$!*

This is correct for two peaks in the Hamburg Port Area but not in the maximum NO concentration measured.
However this was already mentioned previously in the manuscript which I moved in **Section 4.5:**

Dissolved oxygen, which was mainly influenced by primary productivity and respiration (see Figs. 2c–e), plays a significant role in the distribution of nitrogen compounds. In this study, we noted significant negative correlations ($p < 0.0001$) between $O_2$ and $NO_2^-$, $NH_4^+$, and $N_2O$ (Fig. S6). Moreover, distinct peaks of $NO_2^-$ (> 4 µM) and $NH_4^+$ (>9.5 µM) were measured at the sampling sites in the Hamburg Port area at Elbe-km 628.04, 628.21, and 623.40, with the lowest O2 concentrations (<150 µM) (Fig. 3). In this sampling locations, relatively higher concentrations of NO (>14 pM) and $N_2O$ (>30 µM) were also measured. At these sampling stations, the $N_2O$ and NO saturations were exceedingly high, reaching values over 360% and 270%, respectively. These high NO and $N_2O$ saturations are notable, as they suggest a significant level of production. **[Lines 419 – 425].**

*Line 232: I would recommend converting these flux values to fM: 0.31-55 fmol cm$^{-2}$ s$^{-1}$.*

We agree that it would have been better to use the shorter name. However, for the sake of inter-comparability with previous research, we decided to use scientific notation in reporting the flux values.

*Line 238: How do your measurements compare to previous measurements in terms of saturation? If 147-274% saturated is at the low end of marine NO measurements, I'm curious what these higher concentrations correspond to. This would imply that the ocean could be a major source of NO to the atmosphere!*

We updated the Figure to include reported saturation values in previous studies (if these are available). **[Lines 268 – 270]**

*Lines 251-269: I would avoid interpreting a relationship that is not statistically significant. This section is mostly literature review anyways.*

While the relationship in our findings is not statistically significant in linear correlation analysis, the general trend remains that at lower salinity values (with higher DIN), NO values are also relatively higher. In this section, we want to emphasize the importance of DIN input from freshwater, particularly ammonium and nitrite on the NO distribution.

*Lines 276-278: This is a really important finding: you have much higher $NO_3^-$, $NO_2^-$, and $NH_4^+$ than previous studies in other rivere and coastal areas, but not higher NO. What is unique to the Elbe river compared to the other rivers cited here?*

The question of why the Elbe Estuary, with relatively higher nutrients, has a lower NO concentration is still unresolved and requires further investigation. Nevertheless, we think that this observation is an important finding of our study. This observation challenges the assumption that higher concentrations of nitrogen nutrients automatically lead to increased dissolved NO concentration. In our manuscript, subsequent to our discussion that highlighted the Elbe Estuary's relatively higher nutrient levels yet lower NO concentrations compared to other study sites, we discussed the conflicting findings concerning the relationship between NO distribution and nitrogen-containing nutrients. **[Lines 322 – 324]**

We also reported the hypothesis of Ayeni et al. (2021) regarding these conflicting relationships: "…Likewise, Ayeni et al. (2021) also noted that some rivers in Japan with higher $NO_2^-$ concentrations had lower rates of photoproduction of NO and vice versa, attributing these imbalances to nitrogen cycling processes (nitrification, denitrification, and anammox), which could produce or consume NO, or the photochemical transformation of organic nitrogen from dissolved organic matter producing $NO_2^-$ to form NO in areas with low $NO_2^-$."

The high reactivity of nitric oxide (NO) as a radical initiates various consumption mechanisms which may influence its concentration in the Elbe Estuary. Zafiriou et al. (1979) reported that there is no evidence of interaction between NO and metals under marine conditions, though NO is known to react with metals yielding nitrosyl (M-NO) or iso-nitrosyl (M-ON) metal complexes (Ford and Lorkovic, 2002; Richter-Addo et
al., 2002). Additionally, biological productivity can both consume and, produce NO, further contributing to its
dynamic cycle in the environment. A recent paper by (Bange et al., 2024) noted that the consumption
processes for NO in the sea(water) are still unresolved.

*Lines 291-292: What about the $DN_2O/NO_3^-$ ratio?*

Thank you for your helpful comment. There is a significant correlation between NO and $\Delta N_2O/NO_3^-$ ratio in
the Hamburg Port area ($R^2$=0.95, p<0.001) and limnic zone ($R^2$=0.72, p<0.001). We provided the $\Delta N_2O/NO_3^-$
ratio plot in the revised Fig. 8k. **[Lines 395 – 398]**

*Lines 299-302: So the overall trend (which is positive) is driven by the Hamburg Port area, and the overall*
*trend masks the negative relationships in the limnic and coastal-brackish zones. This is a good example of*
*an ecological fallacy.*

Indeed, this is accurate, which underlines the rationale for incorporating this information into the discussion. It
is often essential to focus on finer details as opposed to the broader context, as this approach can reveal
features influenced by biogeochemical processes within particular ecological zones that might otherwise be
overlooked.

*Lines 307-308: Add citation: Burlacot et al. (2020).*

We added this important paper discussing algal photosynthesis utilizing NO and producing $N_2O$ in our citation
**[Line 351]**:

"We explored the possibility of NO production from phytoplankton (e.g., Wang et al., 2020; Kim et al., 2006)
as NO may be generally consumed or produced by phytoplankton while they bloom and/or in response to
environmental stress and pollution (Burlacot et al., 2020; Estevez and Puntarulo, 2005; Mallick et al., 2002;
Zhang et al., 2006)."

*Lines 308-310: It's worth pointing out that the Chl. peaks occurred right before the NO peaks.*

We have checked this comment but did not observe obvious pattern between the chlorophyll a and NO peaks.
However, we included this statements in the revised discussion which we think is relevant:

"During eutrophication, increased nutrient availability stimulates algal growth, leading to $O_2$ depletion at night or
daybreak, as algae consume $O_2$ through respiration. As the algal blooms eventually die off and decompose (Goosen
et al., 1995), microbial processes like nitrifier-denitrification and denitrification thrive under low $O_2$ conditions,
potentially releasing NO and $N_2O$. These biological processes are important in shaping the biogeochemical profile of
the estuary, with photosynthesis contributing to peaks in $O_2$ and chlorophyll a during daylight hours and respiration
leading to $O_2$ depletion and potentially creating favorable conditions for $N_2O$ and NO production during nighttime or
in less oxygenated microenvironments such as suspended sediments or particulate matter (Schulz et al., 2022). Future
studies on the influence of primary productivity and respiration on $O_2$ conditions and the NO production or
consumption processes in estuaries are recommended." **[Lines 435 – 443]**
**Lines 313-315: What about the anaerobic process (anammox, denitrification) in the river sediments?**

We understand that anammox and denitrification processes could occur in the river sediments. We provided
references regarding these anaerobic processes in the sediments (Schroeder et al., 1991; Deek et al., 2013).
There are also previous studies that measured NO in sediments (Sørensen, 1978; Schreiber et al., 2014).

However, we are not certain whether NO released from sedimentary processes significantly impacts NO in the
sampled water in the Hamburg Port Area. The overall water depth in the Hamburg Port Area was >15 m, so
NO released from the sediments is unlikely to make it to the surface layer where we took the samples (because
it has a short lifetime in seawater).

***Line 323: You could also look at the relationship of $DN_2O/NO_3^-$ vs. [O2] or $DN_2O/AOU$ vs. [O2] (Nevison et***
***al., 2003).***

Thank you for your helpful comment. Shown is the result of our regression analysis. We noted a significant
positive correlation between $DN_2O/NO_3^-$ vs. [O2] in the limnic zone but a negative correlation (not significant)
in the Hamburg Port Area. Meanwhile, the $DN_2O/AOU$ ratio vs [O2] is negatively correlated in the overall plot
($R^2=-0.4031$, p=0.02) but there is no significant linear relationship in the coastal brackish zone and limnic
zone. There is a significant negative correlation between $DN_2O/AOU$ vs [O2] ratio in the Hamburg Port Area
($R^2=-0.61928$, p=0.03).

```
{'Overall'              }    {'dN2ONitrateRatio'}      0.23955      0.19431
{'Coastal-Brackish Zone'}    {'dN2ONitrateRatio'}      0.23418      0.65515
{'Limnic Zone'          }    {'dN2ONitrateRatio'}      0.66273      0.013565
{'Hamburg Port Area'    }    {'dN2ONitrateRatio'}     -0.47012      0.12301
{'Overall'              }    {'dN2OAOURatio'    }     -0.40311      0.018083
{'Coastal-Brackish Zone'}    {'dN2OAOURatio'    }     -0.27444      0.44287
{'Limnic Zone'          }    {'dN2OAOURatio'    }     -0.16077      0.61768
{'Hamburg Port Area'    }    {'dN2OAOURatio'    }     -0.61928      0.031764
```

***Line 325: In this context, I would actually call $NO_2^-$ a product of nitrification, not a precursor, because***
***$NH_4^+$ oxidation to $NO_2^-$ produces $N_2O$ and NO as a byproduct; $NO_2^-$ oxidation does not.***

You are correct, $NO_2^-$ is a product of nitrification. We amended the text to accurately reflect nitrite as a product
in the nitrification process. Thank you for bringing this to our attention, ensuring the precision of the scientific
content of our manuscript. We also added simplified chemical reaction R2 to R4. **[Lines 357 – 362]**

***Line 329: The limnic zone correlations in Figure S7 look like they're being driven by two points at either***
***extreme of NO, while the rest of the points cluster in the middle. I would avoid over-interpreting these plots.***

Thank you for your pointing this out. We agree that we have to avoid overinterpretation of the result. However,
it is also important to note the significant correlation that exists at p < 0.001.

***Lines 351-352: Elaborate here upon why the lack of a significant relationship between $NO_3^-$ and AOU***
***indicates the presence of denitrification or nitrifier-denitrification.***

We revised the text to note that there could be other processes aside from nitrification (not just
denitrification/nitrifier denitrification) that affected the $NO_3^-$ and AOU relationship in the Hamburg Port Area.
This may also include high respiration/remineralization rates and mixing with water from the port basins,
which might impact the correlation.

"We think that this lack of correlation between AOU vs $NO_3^-$ may be brought by other nitrogen transformation
processes that influence $NO_3^-$ concentration or that affect $NO_2^-$ oxidation, such as nitrifier-denitrification,
denitrification (R3), anammox (R4), and/or primary production. Previous studies reported that the Hamburg

Port area is a hotspot for $N_2O$ production, attributed to nitrification and nitrifier-denitrification processes (Brase et al., 2017). Prior studies confirmed the highest denitrification rates in the sediments (Deek et al., 2013) and the highest nitrification rates in the water column at this section of the Elbe Estuary (Sanders et al., 2018). During this study, we didn't have the tools to distinguish the exact process involved. However, future studies are recommended to utilize dual stable isotope techniques and molecular or genetic tools to detect marker genes specific to nitrogen-cycling microorganisms." **[Lines 410 – 417]**

*Line 363-365: If you imply that denitrification could be occurring in the sediments even though the water column oxygen concentrations are too high, I don't think you can rule out anammox based on water column oxygen concentrations.*

We agree to this, and in the revise manuscript, we did not exclude the possibility of anammox process without other evidence to rule it out. Genetic analysis of nitrogen cycle marker genes could have been helpful in our data analysis. We have deleted the sentence ruling out anammox.

*Line 367: ...or anoxic microsites within particles.*

Thank you for your helpful suggestion. We added this phrase to the revised manuscript and cited relevant publications (Liu et al., 2013; Xia et al., 2017). **[Line 423]**

*Lines 370-371: I'm really interested in this apparent boom and bust cycle in your data. You have locations with peaks of chlorophyll and oxygen, and other locations with oxygen and pH minima and $N_2O$ maxima. This implies to me that there are some locations where you captured net production and others where you captured net respiration, which draws down $O_2$ and creates an ideal environment for $N_2O$ and NO production in sediments or particles. You allude to this in the conclusions, but how do you think day-night temporal variation in each of your sites affects your data? Would blooms in some locations propagate downstream and create pockets of high respiration further downstream?*

Thank you for highlighting these aspects of our dataset. The observed fluctuations in chlorophyll and oxygen concentrations, along with the corresponding variations in pH and $N_2O$ levels across different locations, indeed suggest episodic events of net production and respiration. These biological processes are fundamental in shaping the biogeochemical profile of the estuary, with photosynthesis contributing to peaks in oxygen and chlorophyll during daylight hours, and respiration leading to $O_2$ depletion and potentially creating favorable conditions for $N_2O$ and NO production during nighttime or in less oxygenated microenvironments such as suspended sediments or particulate matter (Schulz et al., 2022).

In line with your comment, we recognize the necessity for a more comprehensive analysis that accounts for temporal variations, including diurnal shifts, in the study of nitric oxide dynamics in estuaries. Additional research, potentially involving continuous monitoring at various sites, would be invaluable in deciphering these complex interactions and understanding how they might influence the distribution and nitrogen cycling in the estuary. We discussed briefly temporal effects and elaborate on the need for further research on this in our revised discussion. **[Lines 463 – 471]**

*Lines 383-384: You talk very little about photolysis in your discussion so I would remove it here.*

We agree to the Reviewer's comment. We removed photolysis in our conclusion as this is not a major finding in our study.

*Lines 385-397: I would move this paragraph on potential temporal effects into your discussion section (see my previous comment). Then summarize it in your conclusions.*

We followed the Reviewer's suggestion to move the paragraph into the discussion session and summarize it in
our conclusion.

**Technical corrections**

*Line 24: Faulty parallelism: replace "and affecting" with "and affects"*

We edited the text to improve parallel structure. **[Line 24]**

*Line 48: replace "Its estuarine part stretches" with "Its estuaries stretch"*

We edited the text for conciseness. We replaced it to "Its estuary". **[Line 66]**

*Line 83: change to "within 20 minutes OF sampling"*

We edited the grammatical error. **[Line 100]**

*Line 93/eqn. (1): It's confusing to have the letter "x" as the multiplication sign here because you also have*
*an x variable. Use the mathematical symbol you use below or just take them out.*

For consistency, we used the multiplier symbol $\times$ all throughout the manuscript.

*Line 120/eqn. (10): Write $e^{0.0447T}$ not exp.*

We edited the text to reflect the correction pointed out by the Reviewer. **[Line 137]**

*Line 124/eqn. (12): $p_{NO}$ and $K_H$ are quantity symbols - italicize here as you did above.*

Thank you for your attention to detail. We italicized the quantity symbols. **[Line 141]**

*Figures 2, 3, and 5: I would put the y axis labels (salinity, temperature, etc.) on the left side with the y axis*
*ticks - it's confusing to have them on the opposite side of the plot. You can move the subplot labels ("a",*
*"b") to the upper left corner.*

Thank you for the comment. We understand the importance of clarity in presenting scientific data. We edited
the Figures to reflect the comment of the Reviewer.

*Line 158: Add salinity units.*

I initially added the practical salinity unit (psu) as a unit for reporting the salinity from sensors, but I learned
that this is a common mistake and is strongly discouraged:

"It is important to emphasize that Practical Salinities do not have units. This fact, confusing to non-specialists,
is related to technical issues that prevented an absolute definition when PSS-78 was constructed. Sometimes
this lack of units is awkwardly handled by appending the acronym PSU (Practical Salinity Units) to the
numerical value, although doing so is formally incorrect and strongly discouraged."

See Pawlowicz, R. (2013) Key Physical Variables in the Ocean: Temperature, Salinity, and Density. *Nature*
*Education Knowledge* 4(4):13 Available at https://www.nature.com/scitable/knowledge/library/key-physical-
variables-in-the-ocean-temperature-102805293/

Therefore in our revised manuscript, we retained the text excluding units for salinity.

**Figure 6: This is a really nice compilation plot to put your measurements in context. Instead of saying**
**the NO fluxes are x10$^{-17}$, just report in units of fmol cm$^{-2}$ s$^{-1}$.**

Similar to my response in Line 232: We agree that it would have been better to use the shorter name. However,
for the sake of intercomparability with previous research, we decided to use scientific notation in reporting the
flux values.

*Table S3: Table S3: instead of superscripts "a", "b" and "c" corresponding to different significance levels,*
*use \*, \*\*, and \*\*\*, which is the convention.*

We edited the superscripts and use \*,\*\*,and \*\*\* to signify different significant levels.

*Figure 7: Use \*, \*\*, and \*\*\* instead of a, b, c superscripts.*

We edited the superscripts and use \*,\*\*, and \*\*\* to signify different significant levels.

*Lines 332-333: I agree with the ammonium limitation idea but rephrase this and the following sentences to*
*improve clarity and flow.*

*Line 344: Here and elsewhere: "were" not "are", since most of your results are reported in past tense.*

We have checked the grammar and results are reported in the past tense. However we used the present tense to
report trivial information/ facts.

*Line 348: Remove clause "when the nitrification proceeds" – unnecessary.*

We revised the discussion and removed the unnecessary clause.

*Line 350: Remove "therefore" - the support for this statement comes later in this paragraph, not from the*
*preceding one.*

We edited the discussion section.

**Line 355: "correlations" should be plural.**

We corrected the grammar errors in the revised manuscript.

**References**

Amann, T., Weiss, A., and Hartmann, J.: Carbon dynamics in the freshwater part of the Elbe estuary, Germany: Implications of improving water quality, Estuar Coast Shelf Sci, 107, 112–121, https://doi.org/10.1016/j.ecss.2012.05.012, 2012.

Anifowose, A. and Sakugawa, H.: Determination of daytime flux of nitric oxide radical (NO•) at an inland sea–atmospheric boundary in Japan, Journal of Aquatic Pollution and Toxicology, 1, 10, 2017.

Anifowose, A. J., Takeda, K., and Sakugawa, H.: Photoformation rate, steady-state concentration and lifetime of nitric oxide radical (NO) in a eutrophic river in Higashi-Hiroshima, Japan, Chemosphere, 119, 302–309, https://doi.org/10.1016/j.chemosphere.2014.06.063, 2015.

Ayeni, T. T., Jadoon, W. A., Adesina, A. O., Sunday, M. O., Anifowose, A. J., Takeda, K., and Sakugawa, H.: Measurements, sources and sinks of photoformed reactive oxygen species in Japanese rivers, Geochem J, 55, 89–102, https://doi.org/10.2343/geochemj.2.0620, 2021.

Bange, H. W., Mongwe, P., Shutler, J. D., Arévalo-Martínez, D. L., Bianchi, D., Lauvset, S. K., Liu, C., Löscher, C. R., Martins, H., Rosentreter, J. A., Schmale, O., Steinhoff, T., Upstill-Goddard, R. C., Wanninkhof, R., Wilson, S. T., and Xie, H.: Advances in understanding of air–sea exchange and cycling of greenhouse gases in the upper ocean, Elem Sci Anth, 12, https://doi.org/10.1525/elementa.2023.00044, 2024.

Bouwman, A., Boumans, L., and Batjes, N.: Modeling global annual $N_2O$ and NO emissions from fertilized fields, Global Biogeochem Cy, 16, 28-1-28–9, https://doi.org/10.1029/2001gb001812, 2002a.

Bouwman, A. F., Boumans, L. J. M., and Batjes, N. H.: Emissions of $N_2O$ and NO from fertilized fields: Summary of available measurement data, Glob. Biogeochem. Cycles, 16, 6-1-6–13, https://doi.org/10.1029/2001gb001811, 2002b.

Brase, L., Bange, H. W., Lendt, R., Sanders, T., and Dähnke, K.: High Resolution Measurements of Nitrous Oxide (N2O) in the Elbe Estuary, Frontiers Mar Sci, 4, 162, https://doi.org/10.3389/fmars.2017.00162, 2017.

Caranto, J. D. and Lancaster, K. M.: Nitric oxide is an obligate bacterial nitrification intermediate produced by hydroxylamine oxidoreductase, Proc National Acad Sci, 114, 8217–8222, https://doi.org/10.1073/pnas.1704504114, 2017.

Dähnke, K., Bahlmann, E., and Emeis, K.: A nitrate sink in estuaries? An assessment by means of stable nitrate isotopes in the Elbe estuary, Limnol. Oceanogr., 53, 1504–1511, https://doi.org/10.4319/lo.2008.53.4.1504, 2008.

Deek, A., Dähnke, K., Beusekom, J. van, Meyer, S., Voss, M., and Emeis, K.: N$_2$ fluxes in
sediments of the Elbe Estuary and adjacent coastal zones, Mar. Ecol. Prog. Ser., 493, 9–21,
https://doi.org/10.3354/meps10514, 2013.

Ford, P. C. and Lorkovic, I. M.: Mechanistic Aspects of the Reactions of Nitric Oxide with
Transition-Metal Complexes, Chem Rev, 102, 993–1018, https://doi.org/10.1021/cr0000271,
2002.

Gong, J.-C., Jin, H., Li, B.-H., Tian, Y., Liu, C.-Y., Li, P.-F., Liu, Q., Ingeniero, R. C. O., and
Yang, G.-P.: Emissions of Nitric Oxide from Photochemical and Microbial Processes in Coastal
Waters of the Yellow and East China Seas, Environmental Science & Technology, 57, 4039–
4049, https://doi.org/10.1021/acs.est.2c08978, 2023.

Kerner, M.: Effects of deepening the Elbe Estuary on sediment regime and water quality, Estuar
Coast Shelf Sci, 75, 492–500, https://doi.org/10.1016/j.ecss.2007.05.033, 2007.

Lancaster, J. R.: A Tutorial on the Diffusibility and Reactivity of Free Nitric Oxide, Nato Sci S
A Lif Sci, 1, 18–30, https://doi.org/10.1006/niox.1996.0112, 1997.

Liu, T., Xia, X., Liu, S., Mou, X., and Qiu, Y.: Acceleration of Denitrification in Turbid Rivers
Due to Denitrification Occurring on Suspended Sediment in Oxic Waters, Environ. Sci.
Technol., 47, 4053–4061, https://doi.org/10.1021/es304504m, 2013.

Lutterbeck, H. E. and Bange, H. W.: An improved method for the determination of dissolved
nitric oxide (NO) in seawater samples, Ocean Sci, 11, 937–946, https://doi.org/10.5194/os-11-
937-2015, 2015.

Lutterbeck, H. E., Arévalo-Martínez, D. L., Löscher, C. R., and Bange, H. W.: Nitric oxide (NO)
in the oxygen minimum zone off Peru, Deep Sea Res Part Ii Top Stud Oceanogr, 156, 148–154,
https://doi.org/10.1016/j.dsr2.2017.12.023, 2018.

Nevison, C., Butler, J. H., and Elkins, J. W.: Global distribution of N2O and the ΔN2O-AOU
yield in the subsurface ocean, Glob. Biogeochem. Cycles, 17, n/a-n/a,
https://doi.org/10.1029/2003gb002068, 2003.

Olasehinde, E. F., Takeda, K., and Sakugawa, H.: Photochemical Production and Consumption
Mechanisms of Nitric Oxide in Seawater, Environ Sci Technol, 44, 8403–8408,
https://doi.org/10.1021/es101426x, 2010.

Richter-Addo, G. B., Legzdins, P., and Burstyn, J.: Introduction: nitric oxide chemistry., Chem.
Rev., 102, 857–60, https://doi.org/10.1021/cr010188k, 2002.

Schreiber, F., Stief, P., Kuypers, M. M. M., and Beer, D. de: Nitric oxide turnover in permeable
river sediment, Limnol. Oceanogr., 59, 1310–1320, https://doi.org/10.4319/lo.2014.59.4.1310,
2014.

Schroeder, F., Klages, D., and Knauth, H.-D.: Contributions of sediments to the nitrogen budget of the Elbe estuary, Int. Ver. für Theor. Angew. Limnol.: Verhandlungen, 24, 3063–3066, https://doi.org/10.1080/03680770.1989.11899231, 1991.

Schulz, G., Sanders, T., Beusekom, J. E. E. van, Voynova, Y. G., Schöl, A., and Dähnke, K.: Suspended particulate matter drives the spatial segregation of nitrogen turnover along the hyper-turbid Ems estuary, Biogeosciences, 19, 2007–2024, https://doi.org/10.5194/bg-19-2007-2022, 2022.

Schulz, G., Sanders, T., Voynova, Y. G., Bange, H. W., and Dähnke, K.: Seasonal variability of nitrous oxide concentrations and emissions along the Elbe estuary, Biogeosciences Discuss, 2023, 1–21, https://doi.org/10.5194/bg-2023-35, 2023a.

Schulz, G., Sanders, T., Voynova, Y. G., Bange, H. W., and Dähnke, K.: Seasonal variability of nitrous oxide concentrations and emissions in a temperate estuary, Biogeosciences, 20, 3229–3247, https://doi.org/10.5194/bg-20-3229-2023, 2023b.

Sørensen, J.: Occurrence of nitric and nitrous oxides in a coastal marine sediment., Appl. Environ. Microbiol., 36, 809–13, https://doi.org/10.1128/aem.36.6.809-813.1978, 1978.

Tian, Y., Xue, C., Liu, C.-Y., Yang, G.-P., Li, P.-F., Feng, W.-H., and Bange, H. W.: Nitric oxide (NO) in the Bohai Sea and the Yellow Sea, Biogeosciences, 16, 4485–4496, https://doi.org/10.5194/bg-16-4485-2019, 2019a.

Tian, Y., Yang, G.-P., Liu, C.-Y., Li, P.-F., Chen, H.-T., and Bange, H. W.: Photoproduction of nitric oxide in seawater, Ocean Sci, 16, 135–148, https://doi.org/10.5194/os-16-135-2020, 2019b.

Tian, Y., Wang, K.-K., Yang, G.-P., Li, P.-F., Liu, C.-Y., Ingeniero, R. C. O., and Bange, H. W.: Continuous Chemiluminescence Measurements of Dissolved Nitric Oxide (NO) and Nitrogen Dioxide (NO 2 ) in the Ocean Surface Layer of the East China Sea, Environ Sci Technol, https://doi.org/10.1021/acs.est.0c06799, 2021.

Tian, Y., Jian, H.-M., Liu, C.-Y., Gong, J.-C., Li, P.-F., and Yang, G.-P.: Distribution and influencing factors of atmospheric nitrogen oxides (NOx) over the east coast of China in spring: Indication of the sea as a sink of the atmospheric NOx, Mar. Pollut. Bull., 200, 116095, https://doi.org/10.1016/j.marpolbul.2024.116095, 2024.

Walter, S., Bange, H. W., Breitenbach, U., and Wallace, D. W. R.: Nitrous oxide in the North Atlantic Ocean, Biogeosciences, 3, 607–619, https://doi.org/10.5194/bg-3-607-2006, 2006.

Williams, E. J., Hutchinson, G. L., and Fehsenfeld, F. C.: NOx And N2O Emissions From Soil, Glob. Biogeochem. Cycles, 6, 351–388, https://doi.org/10.1029/92gb02124, 1992.

Xia, X., Liu, T., Yang, Z., Michalski, G., Liu, S., Jia, Z., and Zhang, S.: Enhanced nitrogen loss
from rivers through coupled nitrification-denitrification caused by suspended sediment, Sci.
Total Environ., 579, 47–59, https://doi.org/10.1016/j.scitotenv.2016.10.181, 2017.

Yoshinari, T.: Nitrous oxide in the sea, Mar. Chem., 4, 189–202, https://doi.org/10.1016/0304-
4203(76)90007-4, 1976.

Zafiriou, O. C. and McFarland, M.: Nitric oxide from nitrite photolysis in the central equatorial
Pacific, J Geophys Res, 86, 3173, https://doi.org/10.1029/jc086ic04p03173, 1981.

Zafiriou, O. C. and True, M. B.: Nitrite photolysis in seawater by sunlight, Mar Chem, 8, 9–32,
https://doi.org/10.1016/0304-4203(79)90029-x, 1979.

---

## Referee Report (RR1)

*General appraisal*

In their revised manuscript and author comments, Ingeniero et al. addressed my major concerns about the paper (namely, the assumption that nitrifier-denitrification was the only reductive process that may be occurring in their study site). While the authors do not spend a lot of time on the implications of their study for global biogeochemical cycles, the measurements are novel and provide another piece of the puzzle of marine NO cycling.

My main criticism of the revised manuscript is that it should be streamlined and revised for clarity. As it is, the discussion is a bit convoluted and difficult to read — especially section 4.3 (see below).

Also, at this stage of publication the data should be deposited in a repository with an associated DOI. Not enough to say it "will be made available."

*Specific comments*

Lines 130-131: Why not just use the GSW MATLAB toolbox to calculate density?

Lines 292-303: I would drop these two paragraphs and just say, "Nonetheless, salinity alone is insufficient to explain the uneven distribution of NO at our study site, indicating that other parameters influence NO concentrations along the Elbe estuary." The salinity gradient tells you about mixing but not about the sources of NO, so I think it's sufficient in this section simply to point out that the weak negative correlation between NO and salinity indicates that higher NO concentrations in the Hamburg Port area mix out as your move towards the North Sea.

Line 304/Section 4.3: This section still needs to be streamlined and clarified. Is the main point just that high DIN doesn't necessarily lead to high NO? Or that there isn't much evidence for NO photoproduction in your study area?

Line 336/Table 1: Here, is $N_2O$ just the concentration or $\Delta N_2O$? Figure 7 is $\Delta N_2O$…

Line 360: Specify that this reaction is for ammonia-oxidizing bacteria; the exact pathway and enzymology for archaeal nitrification is still a matter of debate. Also, use the commonly accepted abbreviations for each enzyme to make this figure easier to read. E.g., amo instead of ammonium monooxygenase.

Lines 460-462: Wait, I thought you had a whole section on how your study challenges the assumption that higher concentrations of nitrogen nutrients automatically lead to increased dissolved NO concentration?

*Technical corrections*

Line 46: global estimates OF oceanic NO emissions

Line 423: should be "these sampling locations"

---

## Author Response (AR2)

Dear Mr. Ingeniero,

Thank you for submitting the revised version of your manuscript. Two of the three previous referees have reviewed the revisions and provided overall positive feedback. However, there are still concerns regarding the clarity of the discussion (particularly in section 4.3) and the availability of data. Please review the comments thoroughly and make the necessary revisions.

Additionally, when resubmitting, please include a point-by-point response to the Reviewers' comments.

Best regards,
Yuan Shen

Dear Dr. Yuan Shen,

Thank you very much for your time and effort in reviewing our manuscript. We appreciate the constructive feedback provided by the referees and your guidance in enhancing the quality of our work.

We have thoroughly reviewed the comments and made the necessary minor revisions to address these issues. Additionally, we included a detailed point-by-point response to the Reviewers' comments in our resubmission. Our responses are in blue font and in italics.

Thank you once again for your invaluable feedback and support.

Best regards,

Riel Carlo O. Ingeniero
On behalf of the Authors

**Reviewer 1:**

I'm glad the author addressed my previous concerns. And I found the quality of the manuscript has been improved. I think the current form is now acceptable for publication.
Only some minor suggestions left:

*Thank you very much for your time and effort in reviewing our paper. Your valuable suggestions and comments have significantly enhanced the quality of our manuscript, making it ready for publication. We greatly appreciate your detailed feedback and thoughtful recommendations, which have contributed to the improvement and refinement of our work.*

**Main text:**

Fig. 7, Fig S4 and S6 The authors replied that they have made separate correlation plots of NO vs other parameters. But I can't find NO vs salinity plot, NO vs. oxygen plot, and NO vs. AOU plot in other places (e.g., Fig. 8) as I commented on the 1st version manuscript. I would still suggest that NO be added to these figures, as NO is at the center of the discussion in this study.

*Response: We have added NO in these plots.*

Line 45: ranging from $0.70 \times 10^{-17}$?

*Response: This was intended to mean as 0.70 to 45.00 ($\times 10^{-17}$ mol cm$^{-2}$ s$^{-1}$). We have followed the Reviewer's suggestion and edited the text for clarity*

Line 360 R2 ammonium monooxygenase should be ammonia monooxygenase.

*Response: We have followed another Reviewer's suggestion to use the standard abbreviation for the enzymes.*

Line 473 Ludwig Prandtl should be Italic.

*Response: Thank you for your attention to detail. We have italicized Ludwig Prandtl following the editorial guideline of Biogeosciences on the name of research vessels.*

Supplementary information (SI):
Line 5 I see that the authors have revised the address in the text, keep consistent in the SI.

*Response: We updated the address of our research institution in both the manuscript and supplementary information*

Line 50 Table S2 Chlorophyll a (mg L−1) the ")" is incorrectly up-scripted.

*Response: We have corrected the typographic error and changed mg L$^{-1}$ to µg L$^{-1}$.*

Units of Chlorophyll a appeared as (µg L−1) (e.g., Fig. 2) or (mg L−1) (e.g., Table S2) through text and SI. Check and make them consistent.

*Response: I am really sorry for the oversight. It should definitely be µg L$^{-1}$. We have revised this in the manuscript and supplementary information and checked for consistency.*

Ensure that revisions/changes in supplementary information are "accepted" in word/latex/etc. for final publication.

*Response: We have followed the Reviewer's suggestion to ensure that revisions or changes are accepted.*

**Reviewer 2:**

General appraisal

In their revised manuscript and author comments, Ingeniero et al. addressed my major concerns about the paper (namely, the assumption that nitrifier-denitrification was the only reductive process that may be occurring in their study site). While the authors do not spend a lot of time on the implications of their study for global biogeochemical cycles, the measurements are novel and provide another piece of the puzzle of marine NO cycling.

My main criticism of the revised manuscript is that it should be streamlined and revised for clarity. As it is, the discussion is a bit convoluted and difficult to read — especially section 4.3 (see below).

Also, at this stage of publication, the data should be deposited in a repository with an associated DOI. Not enough to say it "will be made available."

*Response: Thank you for your valuable input and for helping to make our manuscript ready for publication. We appreciate the effort and time you have dedicated to reviewing our paper.*

*We acknowledge that our discussion on the implications for the global biogeochemical cycle is not extensive. We tried to avoid overstating or exaggerating the implications of our findings. Nevertheless, we think that our work serves as a solid foundation and will support future research on nitric oxide measurements in estuarine and coastal systems.*

*We have revised the data availability section. The FerryBox data are readily available at the Coastal Observing System for Northern and Arctic Seas (COSYNA) data portal (https://codm.hzg.de/codm/). We have also uploaded our data to another open-access data repository Zenodo: https://doi.org/10.5281/zenodo.11548798.*

**Specific comments**

Lines 130-131: Why not just use the GSW MATLAB toolbox to calculate density?

*Response: We acknowledge that there are multiple methods to calculate seawater density from temperature and salinity. Unfortunately, the main author was not familiar with the GSW MATLAB toolbox at the time of writing the manuscript. Additionally, the seawater density MATLAB function is straightforward and convenient for our purpose. Other papers have used the same tool in calculating seawater density:*

*Barker, S., & Knorr, G. (2023). A systematic role for extreme ocean-atmosphere oscillations in the development of glacial conditions since the Mid Pleistocene Transition. Paleoceanography and Paleoclimatology, 38, e2023PA004690. https://doi.org/10.1029/2023PA004690*

292-303: I would drop these two paragraphs and just say, "Nonetheless, salinity alone is insufficient to explain the uneven distribution of NO at our study site, indicating that other parameters influence NO concentrations along the Elbe estuary." The salinity gradient tells you about mixing but not about the sources of NO, so I think it's sufficient in this section simply to point out that the weak negative correlation between NO and salinity indicates that higher NO concentrations in the Hamburg Port area mix out as you move towards the North Sea.

*Response: We followed the Reviewers' comments to remove the two paragraphs and edited the last concluding sentence.*

Line 304/Section 4.3: This section still needs to be streamlined and clarified. Is the main point just that high DIN doesn't necessarily lead to high NO? Or that there isn't much evidence for NO photoproduction in your study area?

*Response: We began this section by providing context on the primary sources of NO in marine environments, particularly in the open ocean—NO photoproduction and biological production. Previous research (Zafiriou and McFarland, 1981; Zafiriou and True, 1979; Gong et al., 2023) established that the photolysis of nitrite ($NO_2^-$) is a primary source of NO in marine environments.*

*The main point in Section 4.3 is to highlight that high dissolved inorganic nitrogen (DIN) concentrations do not necessarily lead to high NO concentrations. Our observations in the Elbe Estuary demonstrated that despite high DIN levels, NO concentrations were not correspondingly elevated compared to other study sites. We have decided to remove the sentences about turbidity and suspended matter, which were added in response to another Reviewer's comment, as this part was highly speculative.*

*We hypothesize that microbial nitrogen cycling processes might have a greater influence on the NO concentrations observed in the Elbe Estuary than NO photoproduction. This section provides a good transition to the subsequent sections, where we discuss the role of microbial nitrogen cycling processes in detail, which we aim to highlight in this study.*

Line 336/Table 1: Here, is N2O just the concentration or ΔN2O? Figure 7 is ΔN2O…

*Response: Table 1 and Figure 7 are referring to different results. Table 1 discusses the correlation analysis between NO and different nitrogen components, while Figure 7 presents AOU vs ΔN2O, which is helpful in our discussion of the nitrification process.*

Line 360: Specify that this reaction is for ammonia-oxidizing bacteria; the exact pathway and enzymology for archaeal nitrification is still a matter of debate. Also, use the commonly accepted abbreviations for each enzyme to make this figure easier to read. E.g., amo instead of ammonium monooxygenase.

*Response: We followed the Reviewer's comment that the reaction shown is for ammonia-oxidizing bacteria. Initially, we used the full names of the enzymes to aid readers who may not be specialists in the nitrogen cycle. However, we have now revised the text accordingly and used commonly accepted abbreviations for each enzyme.*

Lines 460-462: Wait, I thought you had a whole section on how your study challenges the assumption that higher concentrations of nitrogen nutrients automatically lead to increased dissolved NO concentration?

*Response: Yes, the study does challenge that assumption. It highlights that site-specific conditions, such as microbial nitrogen cycling, should also be taken into account. We noted that despite the high nutrient concentration in the Elbe Estuary compared to other study sites, the dissolved NO concentration did not correspond to a higher concentration in the Elbe Estuary. It is crucial to consider these site-specific conditions rather than assuming a direct correlation between nitrogen nutrient concentration and dissolved NO concentration. For clarity, we have removed the sentence.*

Technical corrections

Line 46: global estimates OF oceanic NO emissions
*Response: We have edited the text as suggested by the Reviewer.*

Line 423: should be "these sampling locations"
*Response: We have edited the text as suggested by the Reviewer.*